# Segmented filamentous bacteria undergo a structural transition at their adhesive tip during unicellular to filament development

Ana Raquel Cruz [1], Benedikt H. Wimmer [2], Teck Hui Teo[1,8], Gérard Pehau-Arnaudet[3], Jean-Marie Winter[4], Anastasia D. Gazi [3], Pierre Lafaye [5], Sébastien Brier [6], Agnès Legrand [1], Anna Dubrovsky-Gaupp[2], Marion Bérard [7], Gabriel Aymé [5], Anna Sartori-Rupp [4], Ohad Medalia[2] & Pamela Schnupf [1] ✉

Segmented filamentous bacteria (SFB) are intestinal commensals that promote immune system development and pathogen protection through intimate attachment to the ileal epithelium. Attachment occurs via the tip of unicellular teardrop-shaped SFB, called intracellular offsprings (IOs), before outgrowth into filaments. To characterize this critical stage of the SFB life cycle, we imaged SFB using cryo-electron microscopy and tomography. IOs were surrounded by a repetitive surface (S)-layer that became replaced by disordered hair-like structures specifically at the IO tip. During outgrowth into filaments, the S-layer was exchanged for a morphologically distinct repetitive hair-like layer. The bacterial structures and morphological transition were conserved across SFB from mouse and rat origin, while growth of mouse-SFB under non-attachment conditions in a heterologous host affected the SFB tip length and relative proportion of the tip stages. Moreover, a major Th17 and B cell antigen, while being a ubiquitous cell wall-associated protein, was immunologically accessible only at the filament tip, underscoring the unique properties of the tip structure. This study identifies an IO-specific S-layer and reveals a conserved developmental transition of the SFB tip surface including the transient appearance of structures consistent in location and timing with being involved in host cell attachment.

The intimate interactions between the intestinal microbiota and its host play a crucial role in shaping host physiology including the maturation of the immune system[1]. Segmented Filamentous Bacteria (SFB) are ubiquitous commensals in vertebrates but most extensively studied in mice where they strongly stimulate both innate and adaptive immunity[2–7] and provide protection against intestinal and respiratory pathogens[3,8,9]. However, these *Clostridium*-related monoderm bacteria can also be harmful by exacerbating symptoms severity in a number of disease models[10]. Despite their immunomodulatory roles, SFB remain poorly studied

[1]Laboratory of Host-Microbiota Interaction, Université Paris Cité, INSERM UMR S1151, CNRS UMR S8253, Institut Necker Enfants Malades, 75015 Paris, France. [2]Department of Biochemistry, University of Zurich, 8057 Zurich, Switzerland. [3]Ultrastructural BioImaging, Institut Pasteur, Université Paris Cité, CNRS UMR 3528, 75015 Paris, France. [4]NanoImaging Core Facility, Institut Pasteur, Université Paris Cité, F-75015 Paris, France. [5]Antibody Engineering Platform, Institut Pasteur, Université Paris Cité, CNRS UMR 3528, C2RT, 75015 Paris, France. [6]Biological NMR and HDX-MS Platform, Institut Pasteur, Université Paris Cité, CNRS UMR 3528, 75015 Paris, France. [7]Animalerie Centrale, Institut Pasteur, Université Paris Cité, 75015 Paris, France. [8]Present address: A∗STAR Infectious Diseases Labs, Agency for Science, Technology and Research, 138648 Singapore, Republic of Singapore. ✉e-mail: pamela.schnupf@inserm.fr

due to technical constraints such as limited in vitro culturing[11] and a lack of genetic tools.

Unlike most intestinal commensals, SFB are part of the more restricted mucosa-associated microbiota[1]. To establish their niche, teardrop-shaped unicellular SFB, called intracellular offsprings (IOs), attach to ileal epithelial cells and cells overlaying Peyer's patches[12]. IOs attach using their tip, an unusual hook-like structure, and elongate to form segmented filaments that can reach over 100 μm in length[11,12]. Filament differentiation leads to the formation of two new IOs per bacterial segment and eventual IO release at the distal end[12,13].

The host-specific interaction[14,15] mediated by the SFB tip leads to actin re-organization at the attachment site without evidence of SFB penetration beyond the terminal web or apparent plasma membrane disruption[16–18]. Endocytic vesicle formation near the attachment site[18] results in the uptake of a major SFB antigen of the B cell[19] and T helper 17 (Th17) response[2,3,7,18], a hallmark host response to SFB colonization[2,3,7,18]. However, genome analyses of mouse- and rat-SFB have provided little insights regarding the composition of the SFB surface[20–23]. The typical cell wall binding motifs and domains present in most Clostridia are largely absent[20–23]. This includes surface layer homology (SLH) domains characteristic of some, but not all, surface layer (S-layer) proteins from monoderm bacteria[24]. Instead, likely surface-anchoring domains in SFB include a domain of unknown function (DUF4214) identified in multiple predicted surface proteins[20–23]. Some genes potentially involved in polysaccharide capsule formation were also found[20–23] but scanning and transmission electron microscopy have not identified bacterial supramolecular structures such as an S-layer or capsule in SFB[17,25].

To obtain a more detailed view of the SFB surface, we imaged purified IOs and filaments from mouse- and rat-SFB in a hydrated state using a combination of cryogenic electron microscopy (cryo-EM) and tomography (cryo-ET). We identified a number of intracellular structures but mainly focused on the SFB surface at the bacterial tip for which we describe a morphological transition during the outgrowth of unicellular IOs into filaments. This transition includes the replacement of an IO-specific S-layer by a repetitive hair-like layer, the appearance of tip structures, and a unique accessibility of a major Th17 and B cell antigen at the filament tip.

## Results

### Unicellular SFB are surrounded by an S-layer
We aimed to visualize the cellular structures that compose SFB in their native state and assess their conservation in SFB from different hosts. For this, we used cryo-EM and cryo-ET to image whole cells of SFB isolated from monocolonized mice (mouse-SFB) and rats (rat-SFB)[26,27]. We mainly focused on the SFB tip structure (Fig. 1a), which has a diameter of approximately $168.8 \pm 28.9$ nm (n = 10, Supplementary Data 1), and therefore can be studied in toto. A total of 130 mouse-SFB (85 IOs and 45 filaments) and 68 rat-SFB (48 IOs and 20 filaments) were imaged (Supplementary Data 1). The SFB surface was characterized at both the bacterium's tip and distal end (Fig. 1a–c). Bacterial tip imaging included part of the SFB cell body that, together with the separate imaging of the IO distal end, was considered to represent the overall bacterial surface beyond the tip.

Exterior to the SFB membrane, a region of low electron density was followed by a more electron-dense region although this region was not always well defined (Fig. 1d–g). These two layers likely constitute the inner wall (IW) and outer wall (OW) zones of the cell wall (CW), as previously described for other monoderm bacteria such as *Bacillus subtilis*[28].

Exterior to the cell wall, mouse-SFB IOs (Fig. 1d, e) and rat-SFB IOs (Fig. 1f, g) exhibited another electron-dense layer at both the tip (Fig. 1d/f(ii), h, j) and the distal end (Fig. 1e/g(ii), i, j). In cross-section views, this outer layer appeared to be composed of repetitive crescent-like elements and regularly-spaced globular densities (Fig. 1d–g(ii), j)

that showed resemblance to the S-layer of *Caulobacter crescentus*, *Clostridioides difficile*, and *Clostridium thermocellum*[29–31]. These data are consistent with the coverage of the entire IO surface by an S-layer, similarly to what is described for other bacteria[32].

The distance between the repetitive globular subunits was measured manually, by Fourier spectra analysis, and by plot profile tracing on the side view of tomograms from the IO tip (Fig. 1d/f(iii–vi), k). The three methods consistently estimated the globular subunit spacing at around 8 nm. Spacing was similar for mouse-SFB ($8 \pm 1$ nm, n = 50) and rat-SFB ($8 \pm 1$ nm, n = 50) ($p = 0.4109$, Fig. 1k) and was within the range of the typical center-to-center S-layer spacings (2.5–35 nm) of other bacteria[33].

Top views of tomograms from the IO tip suggest that the electron dense subunits are arranged in rows (Fig. 1l and Supplementary Fig. 1a–d). Additionally, an apparent hexagonal organization was detected (Supplementary Fig. 1e, f) and most clearly seen in a tomogram of a rat-SFB IO with a broken tip phenotype whereby the S-layer showed abrupt discontinuity and was absent at the tip end (Supplementary Fig. 1f). The lack of a clear symmetry seen in the top view of SFB tomograms may be explained by the need to accommodate differences in SFB tip curvature. In the top views of intact SFB tips, the arrangement of S-layer subunits revealed a twisted appearance at the tip (Supplementary Fig. 1a).

Identification of the S-layer symmetry will require a more detailed analysis beyond what whole SFB cryo-ET can provide. In comparison, in the relatively closely-related species *C. difficile*, a p2-symmetric S-layer was observed[30], while the morphologically more similar *C. crescentus* is covered by a p6-symmetrical S-layer[29]. Together, these data reveal the presence of a repetitive outermost layer with the typical S-layer center-to-center spacings that surrounds IOs and is conserved in SFB from various hosts.

### Identification of intracellular and extracellular SFB features
During the SFB surface characterization, several intracellular and extracellular structures were identified (Fig. 2a–f and Supplementary movies 1–4). These structures were characterized at the SFB tip and proximal cell body (Fig. 2a–f and Supplementary Fig. 2a–e) although some of these features were also found at the SFB distal end (Supplementary Fig. 3a, band Supplementary Data 1).

The presence of flagella at the concave side of SFB IOs, as previously described[34], was confirmed for both mouse- and rat-SFB (Fig. 2a, e/f(i) and Supplementary Fig. 2a–d(i)). A structure typical of a chemosensory array, predicted to be present by genome analysis[20,21], was found localized at the convex side of the cell body, opposite to where flagella are anchored, in IOs from both hosts (Fig. 2b, e/f(iii), and Supplementary Fig. 2a–d(ii)).

Vesicles with a median diameter of 51 nm for mouse-SFB (average diameter of $54 \pm 15$ nm, n = 15) and 41 nm for rat-SFB (average diameter of $62 \pm 53$ nm, n = 6) (Supplementary Table 1) were identified inside the tip of IOs and filaments (Fig. 2a, e/f(ii), Supplementary Fig. 2a(v), b(iii), c/d(vi), and Supplementary Data 1).

We also identified electron-dense lines parallel to the cell wall at the SFB tip, here designated as tracks (Fig. 2a and Supplementary Fig. 2a/b(v), c(vi), d(vii)). Tracks were commonly identified at the mouse-SFB tip but rarely at the rat-SFB tip (Supplementary Data 1 and Supplementary Fig. 2d(vii)), possibly due to the shorter tip length of rat-SFB compared to that of mouse-SFB (Supplementary Fig. 4). Given that our cryo-ET dataset mainly focused on the tip structure, these intracellular features were only characterized in this region even though they could also be found at the distal end (Supplementary Fig. 3a and Supplementary Data 1).

Tracks were designated as short (Fig. 2a and Supplementary Fig. 2a/b(v), c(vi), d(vii)) when their distance could be measured and as long when we could not fully visualize them in the field of view imaged (Supplementary Fig. 5a). While these features were commonly present

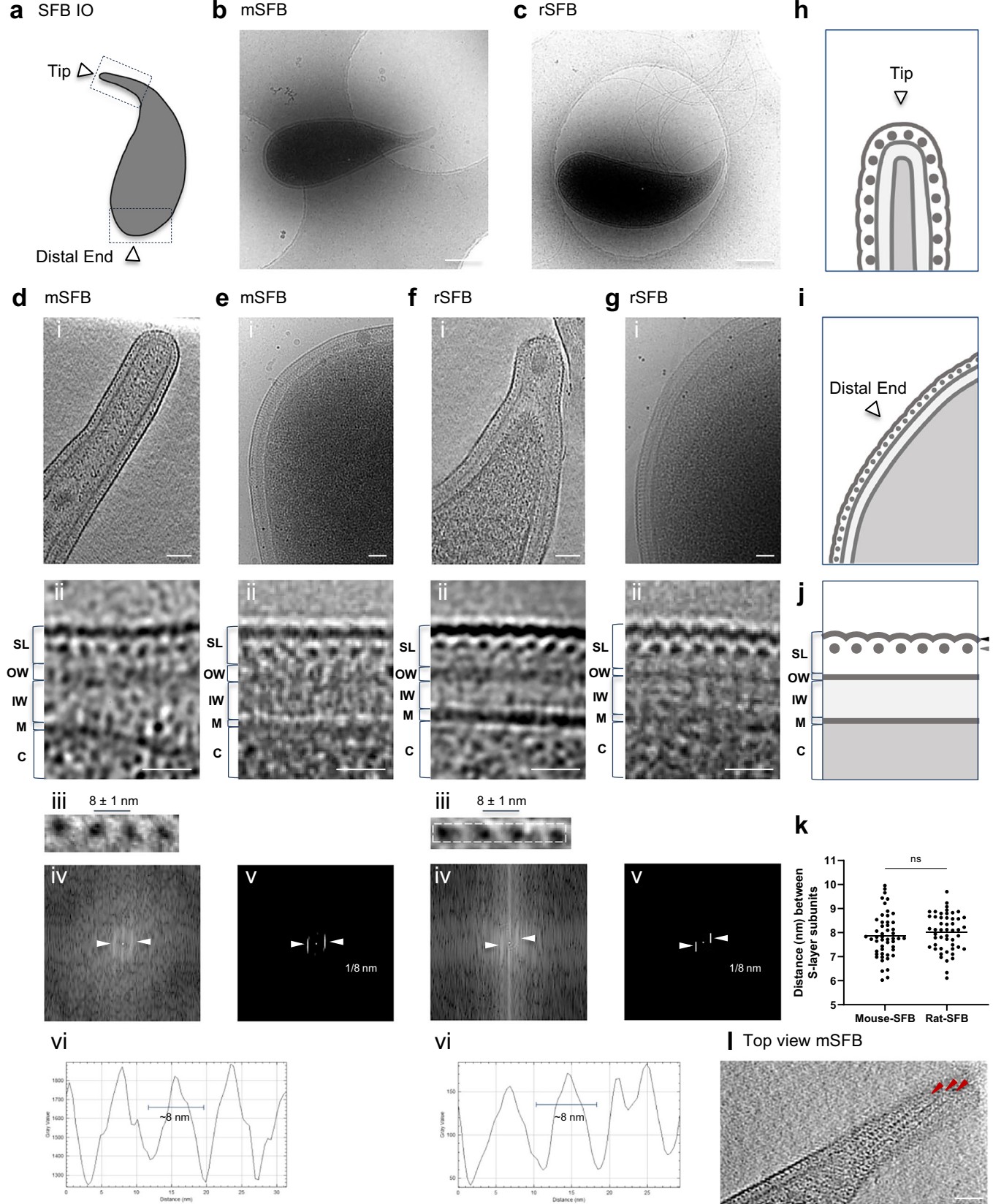

on both sides of the tip, we also observed them only on one side (Supplementary Fig. 5b). Short and long tracks could be found in the same bacterium (Supplementary Fig. 5c). Tracks were only identified in SFB that showed evidence of cytoplasmic membrane retraction (Supplementary Fig. 5d(i)) and were present in the majority of SFB also

harboring intracellular vesicles (61%, n = 18) (Supplementary Fig. 5d(ii)). The length and thickness of short tracks was on average 137 ± 56 nm (n = 15) and 58 ± 30 nm (n = 10), respectively (Supplementary Fig. 5e(i/ii)). The length and thickness could not be estimated for long tracks due to the limited region imaged (Supplementary Fig. 5e(i/ii)). Nevertheless,

**Fig. 1 | Identification of an S-layer in SFB IOs. a** Schematic representation of a Segmented Filamentous Bacteria (SFB) intracellular offspring (IO). The IO tip and distal end are indicated by white arrow heads. Projection image of: **b** a mouse-SFB IO and **c** a rat-SFB IO. Tomographic slice showing the tip of: **d**(i) a mouse-SFB IO (EMD-52655) and **f**(i) a rat-SFB IO (EMD-52684). **e/g**(i) Projection images of the distal end of: **e**(i) a mouse-SFB IO and **g**(i) a rat-SFB IO. Close-ups of the layers present in SFB poles, shown in (**d**–**g**(i)), respectively. **d/f**(iii), Close-ups of S-layer subunits from (**d/f**(i)). The average spacing and standard deviation between S-layer subunits is shown. The region from (**f**(iii)) used for repetitive pattern analysis is delimited by a dashed line. **d/f**(iv), Fast Fourier transform, **d/f**(v), Maximum intensity peaks (indicated by white arrow heads) and their corresponding frequency, and **d/f**(vi), Plot profiles are shown for: **d**(iii) mouse-SFB and **f**(iii) rat-SFB S-layer subunits. Schematic representation of: **h** the tip, **i** the distal end and **j** the layers found in S-layer-containing IOs. The regions where crescent-like and globular subunits of the S-layer can be found are indicated by black and gray arrow heads, respectively. **k** Distance between S-layer subunits in mouse-SFB and rat-SFB. Ten measurements were performed for each of the 5 SFB selected from 2 independent experiments (n = 50 measurements). The mean is shown and the statistical significance was assessed using an unpaired two-sided t-test (ns not significant). Source data are provided as a Source Data file. **l** Top view of the organization of S-layer subunits in rows seen in tomograms from the tip of the mouse-SFB IO shown in (**d**) (EMD-52655). Red arrow heads show S-layer subunits arranged in rows. SL surface layer (S-layer); OW outer wall zone; IW inner wall zone; M membrane; C cytoplasm. mSFB mouse-SFB; rSFB rat-SFB. Scale bars: **b**, **c**: 500 nm; **d**–**g**(i), **l**: 50 nm; **d**–**g**(ii): 20 nm.

given their similar location and width (Supplementary Fig. 5e(iii)), we hypothesize that both short and long tracks may play a role in maintaining the SFB shape or in anchoring the cytoplasmic membrane within the elongated tip and beyond.

Two types of intracellular filament-like structures were also identified in both mouse- and rat-SFB IOs (Supplementary Data 1). One type consisted of multiple filaments clustered together with a thickness of approximately only $26 \pm 12$ nm (n = 23), which made it challenging to follow them in every tomographic slice (Fig. 2c, e/f(iv)), while the other filaments resembled more plate-like structures with a thickness of approximately $63 \pm 20$ nm (n = 6) (Fig. 2d, e/f(v)). The difference between the estimated thickness of these two intracellular features was statistically significant (Supplementary Fig. 3c(i), $p = 0.0110$). The width of individual clustered filaments and plate-like structures was also significantly different (Supplementary Fig. 3c(ii), $p = 0.0055$). Clustered filaments had a width of approximately 9 nm (mouse-SFB, $9 \pm 1$ nm, n = 23; rat-SFB, $9 \pm 1$ nm, n = 9) while the width of plate-like structures was 4/6 nm (mouse-SFB, $4 \pm 1$ nm, n = 6; rat-SFB, $6 \pm 1$ nm, n = 4) (Fig. 2c, d and Supplementary Fig. 2a(ii, iii), b–d(iii, iv), and Supplementary Table 1). Clustered filaments and plate-like structures were at least 50 and 100 nm in length, respectively. Due to the technical limitations stemming from sample thickness and from the limited field of view, their full length could not always be determined.

Clustered filaments were found both in the center of the cell body or parallel to the membrane in the concave side of IOs along the curvature of the cell (Supplementary Fig. 2a/b(iii)). These structures showed a potential resemblance to the cytoskeletal filaments formed in *C. crescentus* by the CTP synthase[35], an enzyme also encoded in the genome of mouse- and rat-SFB[20–23]. In *C. crescentus*, these filaments are ~500 nm in length, play a role in the regulation of bacterial curvature and, in initial developmental stages, they are found in the center of the cell body but their position changes to the cell periphery parallel to the inner membrane in later developmental stages[35]. Conversely, plate-like structures were less common (Supplementary Data 1) and were found in the center of the IOs cell body, but also at the SFB distal end (Supplementary Fig. 3b), either isolated or in groups of 2–3 plate-like structures, showing a maximum of 200 nm in length when their full length could be measured (Supplementary Fig. 2a(ii), b(iv)). To our knowledge, these plate-like structures do not resemble any previously characterized structures.

Lastly, two types of extracellular vesicles (eVs) were identified: undecorated (Supplementary Fig. 2e) and decorated (Supplementary Fig. 2a(iv), b(iii), c/d(v)). Morphologically similar undecorated eVs have been identified for the monoderm gut commensal *Bifidobacterium longum*[36] and decorated vesicles are commonly found in diderm bacteria[37]. In our samples, eVs were found in close proximity (Supplementary Fig. 2a(iv), b(iii)) and in direct contact with the SFB tip (Supplementary Fig. 2e(iii)). Additionally, both undecorated and decorated vesicles were identified near SFB that possess both intact and broken tip phenotypes (Supplementary Fig. 2a(iv), b(iii) and Supplementary Data 1). Overall, the average diameter of eVs was

$85 \pm 40$ nm (n = 15) in mouse-SFB and $103 \pm 54$ nm (n = 26) in rat-SFB samples (Supplementary Table 1), placing them within the range of Gram-positive eVs[38]. While there is some evidence that similar vesicles can also be found in vivo, on SFB IOs attached to the epithelium of cows[39], we are currently unable to determine if the eVs are SFB-derived since their molecular composition remains unknown. Overall, whole-cell cryo-ET thereby enabled the identification of expected, predicted, as well as unexpected cellular features of mouse- and rat-SFB.

## The SFB tip undergoes a morphological developmental transition

We next refocused on the surface characterization of the SFB tip (Fig. 3a–n and Supplementary movies 5–8) as this structure mediates adhesion to the host epithelium and is therefore the primary location for potential attachment-related structures. SFB were grouped into stages based on the surface structures identified at the tip. In mouse-SFB IOs of less than 3.2 μm in length (n = 55), 24% were fully covered by an S-layer (stage 1 IOs, Fig. 3a, f). In the remaining IOs below 3.2 μm, the tip showed varying degrees of hair-like structures with a disordered appearance seen from both side (Fig. 3b, c, f and Supplementary Fig. 6a) and top views (Supplementary Fig. 7a). These structures were found only at the tip edge in 20% of IOs (stage 2 IOs, Fig. 3b, f), while they were consistently present further down the tip in 18% of IOs (stage 3 IOs, Fig. 3c, f and Supplementary Fig. 6a).

In 20% of the IOs below 3.2 μm, discontinuities were observed in the bacterial S-layer without disordered hair-like structures, leading to the broken tip phenotype and preventing us from identifying the tip stage (Unidentified: UD, Fig. 3f and Supplementary Fig. 8a). This broken phenotype was only found at the tip, suggesting increased fragility of this region, and was predominantly found in IOs (Supplementary Fig. 8b, c), potentially due to the transition from an S-layer to disordered hair-like structures.

The distinctly repetitive S-layer subunits could no longer be seen in the regions occupied by disordered hair-like structures (Fig. 3c and Supplementary Fig. 6a). However, the S-layer remained present in stages 2 and 3 IOs further down the tip towards the cell body (Fig. 3b(iii), g) as well as in the distal end, where disordered hair-like structures were never observed (Fig. 3j, and Supplementary Fig. 9a, b). Through tracking of the disordered hair-like structures in tomograms, these structures were found to be in contact, or close proximity, to the electron dense line corresponding to the outermost region of the cell wall (Fig. 3c, g and Supplementary Fig. 6a) and to be $34 \pm 6$ nm in length (n = 250) (Fig. 3n) and $1.3 \pm 0.4$ nm in width (n = 250) (Source Data). Together, these results reveal a structural transition from an S-layer to disordered hair-like structures specifically at the tip of IOs.

As IOs transitioned to filaments, the bacterial surface became covered by a morphologically distinct outer layer, here designated as hair-like layer (HLL). This outer layer was found in 18% of IOs below 3.2 μm (Fig. 3f and Supplementary Fig. 6b, d) and in 100% of bacteria above this length (Fig. 3d, e, h, i, l, m, and Supplementary Fig. 8b, c). It is composed of an array of ordered hair-like structures whose

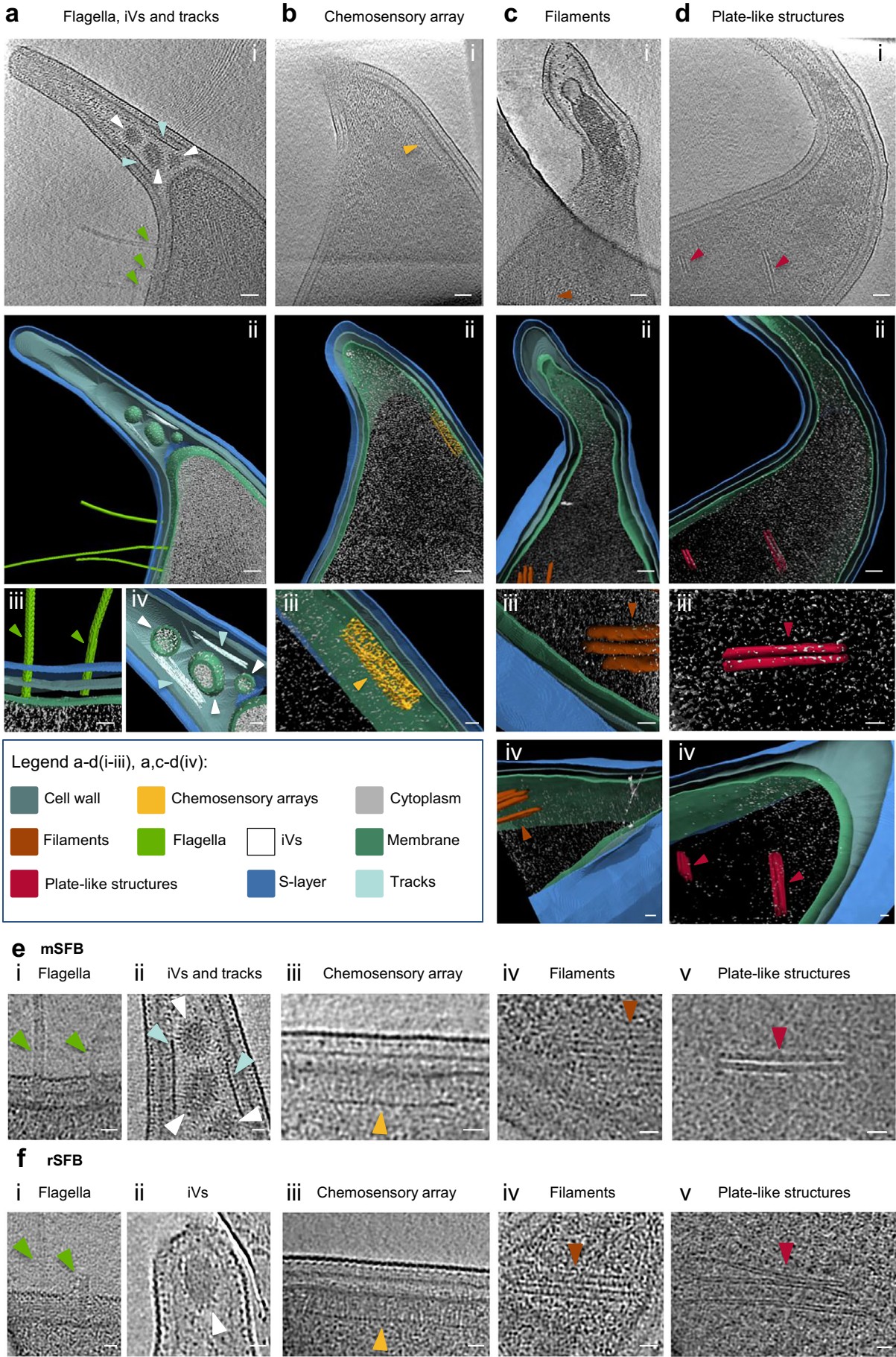

**Fig. 2 | Description of intracellular and surface SFB features identified at the SFB tip and proximal region of the cell body. a–e** Representative tomographic slices from mouse-SFB IOs containing: **a**, **e**(i–ii) flagella, intracellular vesicles (iVs) and tracks, **b**, **e**(iii) chemosensory array, **c**, **e**(iv) filaments and **d**, **e**(v) plate-like structures. Representatives of the identified features are shown by arrow heads of the colors indicated in the legend. **a–d**(i), Tomographic slices from reconstructed tomograms EMD-52655 (Supplementary movie 1), EMD-52667 (Supplementary movie 2), EMD-52668 (Supplementary movie 3) and EMD-52669 (Supplementary movie 4) from SFB IOs with a length of: 2.7, 2.3, 2.0, 2.7, and 2.5 μm. **a–d**(ii), Segmentation rendering of the SFB tip and the identified intracellular and extra-cellular features present in the tomographic slices **a–d**(i). **a–d**(iii) and **a**(iv), Close-

ups of: **a**(iii) flagellum, **a**(iv) intracellular vesicles and tracks, **b**(iii) chemosensory array, **c**(iii) filaments and **d**(iii) stacks from segmentation rendering shown in **a**−**d**(ii). **c/d**(iv), Close-ups from the segmentation rendering shown in **c/d**(ii) to show the thickness in z of **c**(iv) filaments and **d**(iv) plate-like structures. **e/f**(i–v), Close-ups of the **e** mouse-SFB and **f** rat-SFB features represented in (**a**−**d**)(iii) and **a**(iv) from the **e** mouse-SFB tomographic slices shown in **a**−**d**(i) and **f** from rat-SFB tomograms. The **f**(i) flagella and **f**(iv) filaments were isolated from the tomogram EMD-52689. The **f**(ii) intracellular vesicles, **f**(iii) chemosensory array and **f**(v) plate-like structures were isolated from tomograms EMD-52684, EMD-52690 and EMD-52691, respectively. Scale bars: **a**−**d**(i, ii): 50 nm; **a**−**d**(iii), **a/c/d**(iv), **e**, **f**: 20 nm.

arrangement is challenging to identify from the tip's top view (Supplementary Fig. 7b). The HLL was found at the SFB tip end (stage 5, Fig. 3e and Supplementary Fig. 6d(i–ii)), tip side (stage 4 and 5, Fig. 3h, i and Supplementary Fig. 6b–d(iii)) and distal end (stage 4 and 5, Fig. 3k, Supplementary Fig. 9c, d). In stage 4 SFB, the end of the tip possessed disordered hair-like structures (Fig. 3d and Supplementary Fig. 6b, c), which, however, were significantly smaller ($p < 0.0001$) than those of stage 3 ($28 \pm 5$ nm, $n = 250$, Fig. 3n). The cell body of stage 4 SFB was fully covered by the HLL. We only identified one mouse-SFB IO (3.3 μm in length) out of the 14 stage 4 SFB imaged (7%) that showed all three features: HLL, disordered hair-like structures and the likely S-layer, which has neither HLL nor disordered hair-like structures (Supplementary Fig. 10a). This IO may therefore represent a rare transitional stage between stages 3 and 4.

In stage 5 SFB, the disordered hair-like structures were no longer present at the tip (Fig. 3e, i, k, and Supplementary Fig. 6d). Conservation of the HLL at the SFB distal end was confirmed in stages 4 and 5 SFB of variable lengths (Supplementary Data 1), including filaments of over 30 μm (Supplementary Fig. 9d). As stages 4 and 5 SFB were significantly longer than the SFB stages 1–3 ($p$ values from $p = 0.0040$ to $p < 0.0001$, Fig. 3l, m), stages 4 and 5 correspond to more advanced growth stages.

The filament-associated HLL was exterior to the cell wall and could be divided into an inner and outer part, separated by an electron dense layer. This electron dense layer was in a similar location to where the crescent-like elements of the IO-associated S-layer were in stages 1–3 SFB (Fig. 3g–i). Through manual tracing, individual electron dense hair-like subunits, comprising inner and outer parts of the HLL, were found to be $1.4 \pm 0.4$ nm ($n = 250$) in width (Source Data) and, at $28 \pm 4$ nm ($n = 250$) in length, to be significantly shorter than the disordered hair-like structures of stage 3 ($p < 0.0001$, Fig. 3n). Together, these results reveal a morphological transition of the SFB surface from an S-layer, surrounding small IOs, to a filament-associated hair-like layer characterized by ordered elongated subunits with the transient appearance of longer disordered hair-like structures occurring only at the SFB tip. Given the location of the disordered hair-like structures and their presence in unicellular SFB, we hypothesize that these structures may play a role in SFB attachment to the host.

We have also explored a potential association between the different mouse-SFB stages and the intracellular and extracellular features identified. All flagellated IOs belonged to an early developmental stage (stages 1–3), and the proportion of early- and late-stages (stages 4 and 5) in flagellated and non-flagellated IOs was significantly different (Supplementary Fig. 11a, b). Furthermore, stage 1 mouse-SFB comprised 37% of the flagellated IOs ($n = 11$) and only 12% of the non-flagellated IOs ($n = 74$). However, this change in stage proportions was not statistically significant (Supplementary Fig. 11a, b). Similarly, the chemosensory array was only found in SFB IOs and may be characteristic of early SFB stages since it was identified in five S-layer-covered IOs and in only one HLL-covered IO (Supplementary Data 1). However, the limited field of view of our tomograms prevented us from properly assessing a potential stage-association not only for the chemosensory array but also for other cytosolic features, including

filaments and plate-like structures, which, however, were also only found in SFB IOs (Supplementary Data 1).

Conversely, SFB tracks were found in both IOs and filaments. Long tracks were associated with late-stage SFB (Supplementary Fig. 5f) and tracks were found in significantly longer SFB when either any tracks or only long tracks were analyzed (Supplementary Fig. 5g). Overall, these results provide preliminary evidence for potential associations between intracellular and surface features and the here identified developmental stages.

## The developmental transition is conserved in SFB from different hosts

We next aimed to determine whether the morphological transition of the SFB outer layer was conserved across SFB from a different host by characterizing the outer layer of rat-SFB during the outgrowth of IOs into filaments. As observed for the mouse-SFB tip, features of the early development stages (stages 1 and 3), including the S-layer and the replacement of this layer by disordered hair-like structures at the tip, were only found in IOs (Fig. 4a, b). In addition, the IO-associated S-layer and the absence of disordered hair-like structures were conserved in rat-SFB at the distal end of stages 1 and 3 IOs (Fig. 4e and Supplementary Fig. 9e, f). Similarly, the later developmental stages (stages 4 and 5), characteristic also of longer rat-SFB ($p$ values from $p = 0.0023$ to $p < 0.0001$, Fig. 4g and Supplementary Fig. 8d), possessed the filament-associated HLL at the tip (Fig. 4c, d) and at the distal end (Fig. 4f and Supplementary Fig. 9g). As noted for mouse-SFB, it was challenging to identify a clear arrangement of the HLL subunits from the top view of the rat-SFB tip. An exception was a rare rat-SFB IO showing a potential transitional stage between stages 3 and 4 (Supplementary Fig. 10b and Supplementary movie 9), where arrangements resembling rows could be seen (Supplementary Fig. 7c). Nevertheless, some differences were noted. There was a significant increase ($p < 0.0001$) in the proportion of IOs (74% of rat-SFB IOs vs 14% of mouse-SFB IOs) that could not be assigned to a tip stage (UD, Supplementary Fig. 8e–g). Of these, 31% also showed considerable S-layer discontinuities at the distal end (Supplementary Fig. 9h and Supplementary Data 1). As the purification conditions were the same for both mouse and rat-SFB, this may therefore be due to an increased fragility of the rat-SFB S-layer. Overall, our results confirm the conservation of the main developmental transitions at the tip and the distal end of rat-SFB, including the S-layer, the disordered hair-like structures at the tip, and the HLL surrounding SFB as they grow out into filaments.

## The filament-associated hair-like layer is composed of repetitive subunits

We next aimed to obtain a more precise characterization of the hair-like layer by focussing on the inner (I-HLL) and outer (O-HLL) parts of this filament-associated layer. The I-HLL from stages 4 and 5 SFB was in a similar location to the IO-associated S-layer globular subunits from stages 1 to 3 IOs (Figs. 3g–i, and 5a, b), whereas the O-HLL extended beyond the IO-associated crescent-like elements of the S-layer (Fig. 5c–e). While the outer and inner subunits appeared to be

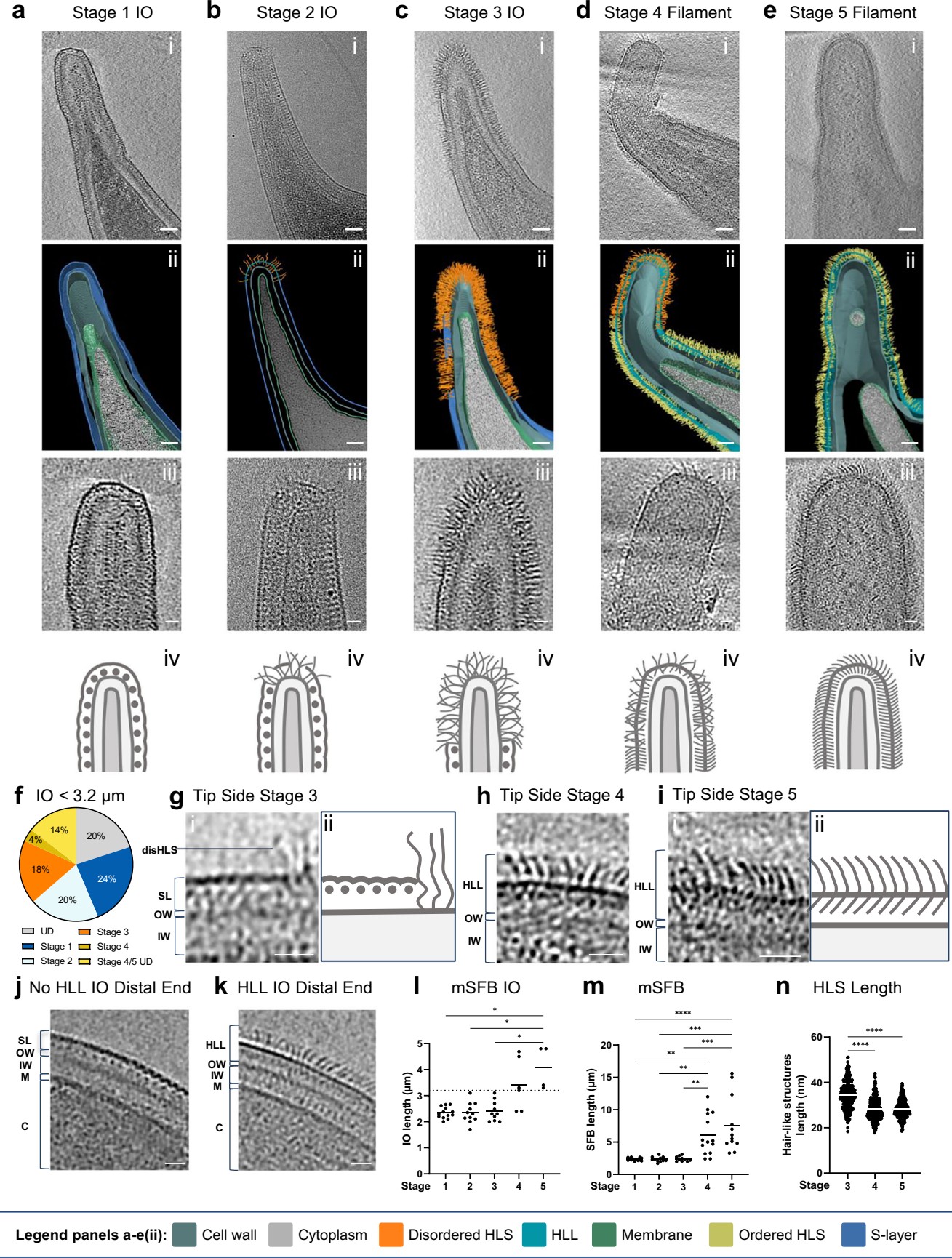

**Fig. 3 | Identification of a morphological developmental transition at the tip of mouse-SFB. a–e** Morphological stages identified at the tip of mouse-SFB. **a/c/d/e**(i), Representative tomographic slices from reconstructed tomograms showing SFB tip of stages: 1 (EMD-52685, Supplementary movie 5), 3 (EMD-52676, Supplementary movie 6), 4 (EMD-52677, Supplementary movie 7) and 5 (EMD-52678, Supplementary movie 8). **b**(i) Representative projection image of stage 2. SFB selected as representative of Stages 1–5 are 2.2, 2.3, 3.1, 12.0, and 15.6 μm in length, respectively. **a–e**(ii), Segmentation rendering of mouse-SFB tip from stages 1–5. The color legend is displayed at the bottom of the figure. The coloring of the hair-like structures (HLS) denotes their qualitative appearance. **a–e**(iii), close-ups of the mouse-SFB tip from stages 1–5. **a–e**(iv), Schematic representation of the mouse-SFB tip from stages 1–5. **f** Pie chart showing the percentage of IOs below 3.2 μm in length identified in each tip stage shown in (**a–d**). IOs for which the tip stage could not be identified (UD: unidentified) and for which the distinction between tip stage 4 and 5 could not be made are also shown. Close-ups of side of the mouse-SFB tip from: **g**(i) stage 3, **h** stage 4 and **i**(i) stage 5. **g/i**(ii) schematic representation of the layers present at the side of the mouse-SFB tip assigned to: **g**(ii) stage 3 and **i**(ii) stage 5. Representative tomographic slices of the distal end of a mouse-SFB IO in which a hair-like layer (HLL) was **j** absent (EMD-52696) and **k** present (EMD-52679). The length of the selected IOs was 2.4 and 2.3 μm, respectively. DisHLS disordered hair-like structures; HLL hair-like layer; SL surface layer (S-layer); OW outer wall zone; IW inner wall zone; M membrane; C cytoplasm. **l/m**, Length of: **l** mouse-SFB (mSFB) IOs (n = 44) and **m** total mouse-SFB (n = 60) assigned to each stage and obtained from 7 independent experiments. A dashed line separates small (<3.2 μm) and outgrowing IOs (>3.2 μm) (**l**). **n** Length of hair-like structures (HLS) from the tip of SFB assigned to stages 3–5. Hair-like structures length was measured from tomograms of 5 bacteria per stage from 2 independent experiments. For each bacterium, 10 structures were measured from 5 tomographic slices (n = 250 measurements). For **l–n** individual measurements and the corresponding mean are shown. The statistical significance was assessed using a Kruskal–Wallis test followed by a Dunn's test correction for multiple comparisons (l: stage 1 vs. 5, p = 0.0158; stage 2 vs. 5, p = 0.0160; stage 3 vs. 5, p = 0.0288; m: stage 1 vs. 4, p = 0.0010; stage 2 vs. 4, p = 0.0016; stage 3 vs. 4, p = 0.0040; stage 1 vs. 5, p < 0.0001; stage 2 vs. 5, p = 0.0002; stage 3 vs. 5, p = 0.0005; n: p < 0.0001). Source data are provided as a Source Data file for (**f, l–n**). Scale bars: **a–e**(i, ii): 50 nm; **a–e**(iii), **g–i, j, k**: 20 nm.

---

aligned, and may therefore be connected, an electron dense layer separated these subunits. The I-HLL and O-HLL were often challenging to identify in the same Z plane (Fig. 5a–d). However, we could simultaneously observe hair-like structures both in the exterior and in the interior of the HLL electron dense line for both mouse-SFB (Fig. 3h, i) and rat-SFB (Fig. 4f).

The distance between the individual subunits of the inner and outer HLL was estimated by Fourier spectra analysis and plot profile tracing on the side view of tomograms of the SFB filament tips and combined with manual measurements (Fig. 5a–d). The subunits in both the inner and outer layer were found to be repetitive and approximately 5 nm apart (n = 50), both in mouse and rat-SFB (Fig. 5a–d, f). This distance between subunits was significantly smaller (p < 0.0001) than the approximately 8 nm separating the globular subunits of the S-layer (Fig. 5f). These results support the replacement of the IO-associated S-layer with a morphologically distinct hair-like layer composed of more tightly arranged repetitive units as SFB transition from IOs to filaments.

**The major Th17 antigen is accessible at the tip of SFB filaments**
The hair-like layer described is, to our knowledge, morphologically distinct from any surface structure previously identified by cryo-EM/ET and its molecular composition remains unknown. The major Th17 antigen (here referred to as Th17Ag)[7], which was recently shown to also be a major B cell antigen, eliciting IgA and IgG responses in the blood and IgA responses in the intestinal lumen[19], was previously found to surround attached SFB of the mouse-SFB-NYU strain[18]. This makes the Th17Ag a potential candidate for a component of the hair-like structures constituting the HLL. To assess the localization of this SFB factor in more detail, we recombinantly expressed and purified the mouse-SFB-NL homolog (AID45212) (Supplementary Fig. 12a, b) and generated a Th17Ag-specific nanobody (VHH). The Th17Ag was localized at the mouse-SFB-NL surface both by immunogold labeling followed by cryo-EM and by immunofluorescence followed by confocal microscopy (Fig. 6a–h, Supplementary Fig. 13a, b and Supplementary Fig. 14a–g).

Labeling with the Th17Ag-specific VHH fused to the Fc region of human IgG1 (VHH-Fc) (Supplementary Fig. 12c) revealed distinct labeling patterns in SFB filaments and IOs. For filaments, Th17Ag labeling was seen in 82% of the filaments imaged (n = 17) and, for 93% of these, was strikingly restricted to the filament tip (n = 13) (Fig. 6a, f and Supplementary Data 2). The Th17Ag labeling was seen in regions containing disordered hair-like structures (Fig. 6e(i)) but not the HLL (Fig. 6e(ii)). For IOs, Th17Ag labeling was observed in only 30% of the IOs imaged (n = 27) (Fig. 6f) and, in 75% of those, occurred at the tip but also on the remaining cell body, including at the distal end (Fig. 6c). In agreement, IOs showed labeling with the VHH-Fc targeting the Th17Ag using immunofluorescence (Fig. 6g). In contrast with the pattern observed in filaments, labeling of IOs was, albeit weakly, observed in regions containing the HLL (Supplementary Fig. 15a, b) but was not always present in regions including disordered hair-like structures (Supplementary Fig 15c). Additionally, as the Th17Ag was detected in the IOs distal end (Fig. 6c and Supplementary Fig. 14c) and in other regions where an S-layer was present (Supplementary Fig. 15c), it is unlikely that the Th17Ag is a component of the disordered hair-like structures.

While labeling was detected across the different IO stages (Supplementary Data 2), late-stage IOs (stages 4/5, Supplementary Fig. 15a, b) were present at a significantly higher proportion in the labeled population (36%, n = 11) compared to the unlabeled population (0%, n = 17) (Supplementary Fig. 16a(i-iii)). The same trend was identified for stage 3 IOs (Supplementary Fig. 14c and Supplementary Fig. 15c) among early SFB stages (stages 1 to 3) with 71% of labeled IOs (n = 7) vs 18% of unlabeled IOs (n = 17) (Supplementary Fig. 16a(iv)).

The tip-restricted labeling of SFB filaments, which could also be seen by immunofluorescence (Fig. 6a, f, g and Supplementary Fig. 14a, e, f), contrasts with earlier reports showing labeling of the Th17Ag surrounding SFB in histological sections[18]. As histological processing involves sample permeabilization, we treated PFA-fixed SFB with 0.5% triton X-100 at 95 °C for 5 min, which disrupts the outermost layer without compromising cell shape integrity. This treatment largely prevented all labeling with the VHH-Fc (Fig. 6f and Supplementary Fig. 14b) but using the rabbit polyclonal antibody raised against the Th17Ag[7] recapitulated the published findings of extensive labeling surrounding 100% of the filaments (n = 7), including the tip structure (Fig. 6b, f, h). In addition, the rabbit polyclonal antibodies showed strong staining of the bacterial surface in 86% of IOs imaged (n = 7) (Fig. 6d, f and Supplementary Data 2). Under non-denaturing conditions, the rabbit polyclonal antibody displayed similar staining patterns as the VHH-Fc (Supplementary Fig. 14a, c, e, f). Epitope mapping of the VHH, following assessment of binding affinity (Supplementary Fig. 17a and Supplementary Table 2), identified three target regions of the Th17Ag towards the protein C-terminus (Fig. 6i). The nearly nonexistent immunolabelling with the VHH-Fc under denaturing conditions (Fig. 6f and Supplementary Fig. 14b, g) is therefore likely due to the disruption of the conformational epitope recognized by VHH (Fig. 6i and Supplementary Fig. 17b, c). These results identify the Th17Ag as an abundant component of the SFB surface for both IOs and filaments whose accessibility is largely restricted to the tip in filaments and moderately accessible over the full surface in a subset of IOs. The Th17Ag, which contains a domain of unknown function also found in SFB proteins annotated as peptidoglycan hydrolases[20–23], is therefore likely a cell wall-associated protein whose accessibility in filaments is prevented by the HLL. For IOs, as only a small subset, enriched in late-

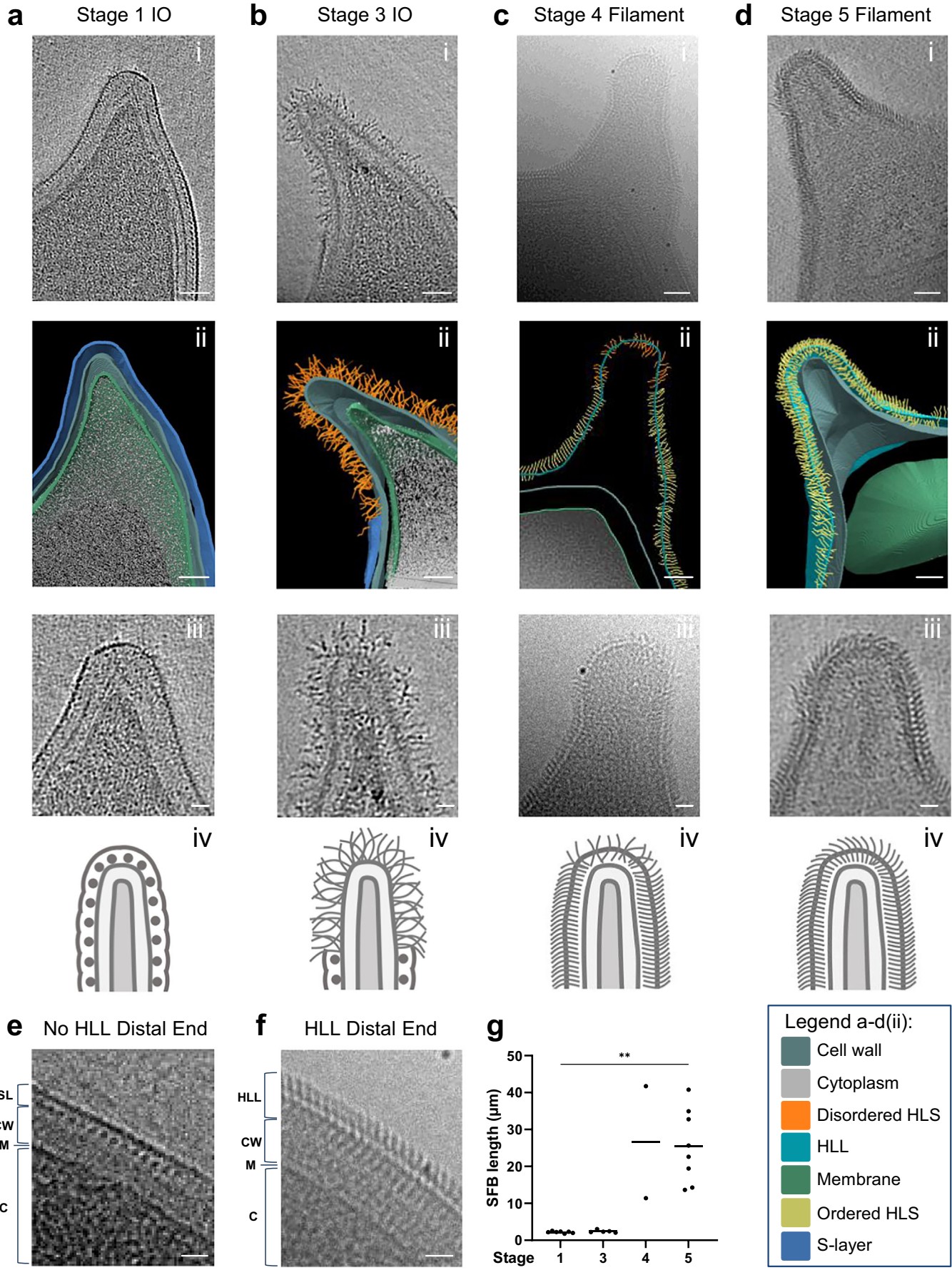

**Fig. 4 | Conservation of the morphological developmental transition at the tip in rat-SFB. a–d** Rat-SFB morphological stages equivalent to those identified at the tip of mouse-SFB. **a/b/d(i)** Representative tomographic slices from reconstructed tomograms showing SFB tip of stages: 1 (EMD-52695), 3 (EMD-52688) and 5 (EMD-52699). **c(i)** Representative projection images of the SFB tip of stage 4. The selected SFB from stages 1, 3, 4, and 5 have a length of 2.2, 3.0, 41.7 and 34.9 μm, respectively. **a–d(ii)** Segmentation rendering of rat-SFB tip from stages 1, 3, 4, and 5. The color legend is displayed at the bottom of the figure. The coloring of the hair-like structures (HLS) denotes their qualitative appearance. **a–d(iii)** Close-ups of the SFB tip. **a–d(iv)** Schematics of the SFB tip. Representative projection images of the distal end of rat-SFB in which a hair-like layer (HLL) was **e** absent and **f** present. SL Surface layer (S-layer); HLL hair-like layer; HLS hair-like structures; CW cell wall (composed of inner and outer wall zone); M membrane; C cytoplasm. **g** Length of SFB assigned to each tip stage ($n = 32$) obtained from 5 independent experiments. Individual measurements were plotted, the mean is shown and the statistical significance was assessed using the Kruskal–Wallis test followed by a Dunn's test correction for multiple comparisons (stage 1 vs 5, $p = 0.0023$). Source data are provided as a Source Data file. Scale bars: **a–d(i,ii)**: 50 nm; **a–d(iii)**, **e**, **f**: 20 nm.

stage IOs, was labeled, we hypothesize that there may be intermediate accessibility stages (Supplementary Fig. 15a, b) prior to the impermeability of the HLL seen in filaments. Overall, our results point towards an increased exposure of the cell wall specifically at the SFB tip.

## SFB growth in a heterologous host affects adhesive tip morphology

We next evaluated the tip stages of mouse-SFB in a context where adhesion was not supported. Attachment, but not growth, of SFB has been shown to be host-specific for rat-SFB under monocolonization conditions in mice[14]. Similarly, mouse-SFB did not attach to the ileal epithelium in rats, while mouse-SFB could still grow robustly in the rat intestinal lumen (Supplementary Fig. 18). When mouse-SFB were grown in rats, all five developmental stages were identified as well as the association of stages 4 and 5 with the outgrowth into filaments (Fig. 7a–f). However, we observed a decrease in tip length for mouse-SFB, both in IOs ($575.6 \pm 146.8$ nm in mice, $n = 24$, vs $467.4 \pm 95.0$ nm in rats, $n = 36$, $p = 0.0010$) and filaments ($625.0 \pm 138.4$ nm in mice, $n = 39$, vs $439.2 \pm 193.4$ nm in rats, $n = 10$, $p = 0.0087$) (Fig. 7g, h, k, l) when SFB of equivalent lengths (IOs, $p = 0.1717$; filaments, $p = 0.2939$) were compared (Fig. 7i, j). Additionally, stage 2 IOs were present at a significantly higher proportion when mouse-SFB were grown in rats compared to when grown in mice (61% vs. 32%, $p = 0.0147$) (Fig. 7m–n). These findings support a role of attachment in the elongation of the tip and in the potential transition of SFB from stage 2 to stage 3, the stage where the disordered hair-like structures are most prominent.

## Discussion

In this study, we characterized the SFB surface by cryo-EM and cryo-ET. We describe two morphologically distinct outer layers that are conserved in SFB from different rodent hosts. The first possesses the characteristics of a bacterial S-layer with morphological similarity to the S-layer of *C. crescentus* and related Clostridia[29–31]. The second, here designated as hair-like layer (HLL), is composed of regularly spaced subunits that resemble hair-like structures (HLS) but, to our knowledge, possesses no clear resemblance to any other outermost layer previously imaged.

The S-layer is only seen in small unicellular SFB, characteristic of early developmental stages, while the HLL is present in longer IOs and in all SFB filaments, characteristic of later developmental stages. The globular subunits of the S-layer can no longer be seen when the HLL is present and the subunits spacing is significantly different between the S-layer and the HLL.

Our results suggest that the transition to the HLL occurs rapidly as it was challenging to identify transitional stages in which both layers were concomitantly present at the SFB surface. The HLL may constitute an atypical surface layer as it is repetitive and exterior to the cell wall. The potential replacement of one surface layer with another is reminiscent of the transition previously described for *Bacillus anthracis*[40]. In *B. anthracis*, an exponential phase-associated S-layer composed of the surface array protein (Sap) is replaced by a stationary phase-associated S-layer composed of the extracellular antigen 1 (EA1).

S-layers can play a role in host adhesion in bacteria with symmetric poles[32]. However, since in SFB it is the tip that mediates epithelial cell attachment, it is unlikely that the S-layer or the HLL are

involved in adhesion but rather may function in bacterial protection and/or maintenance of cell wall integrity. Immunogold labeling of the Th17Ag supports this hypothesis. The Th17Ag was previously found to surround SFB filaments in intestinal sections and to be taken up from the filament tip through the formation of host endocytic vesicles[18]. We obtained similar labeling with permeabilization but without permeabilization this protein was surface-exposed only at the filament tip for the majority of the filaments imaged. Similar to filaments, strong labeling was observed in IOs under permeabilization conditions. However, only approximately a quarter of non-permeabilized IOs showed labeling and labeling was not restricted to a specific region of the bacterial surface. These results suggest that the HLL in SFB filaments prevents access to this potential cell wall-anchored protein while the S-layer appears to have a similar function for IOs. We hypothesize that the increased surface exposure in a subset of IOs may be due to changes in cell wall accessibility during the S-layer to HLL transition. These findings provide further evidence of a developmental transition occurring at the SFB tip.

Additional surface structures identified here are the tip-specific disordered HLS. Since these structures were morphologically different from the subunits of the HLL and only located at the bacterial tip, we hypothesize that these structures may also be functionally different. The disordered HLS found at the tip of IOs otherwise surrounded by an S-layer were longer and morphologically distinct compared to the ordered HLS found at the tip and surrounding long IOs and filaments, suggesting that the disordered and ordered HLS may have a different molecular composition. For type IV pili of *Thermus thermophilus*, the morphological differences observed using cryo-EM were due to distinct protein compositions leading to functional differences[41]. Nevertheless, since no differences were found between the length of disordered and ordered HLS found in stage 4 SFB, experimental evidence at the molecular level is needed to better understand the observed variations in HLS length.

Since attachment is mediated by the SFB tip[17], the tip-restricted location of disordered HLS suggests that they may be involved in SFB attachment to host cells. Furthermore, their emergence prior to bacterial elongation and the increase in the proportion of IOs without fully developed disordered HLS in a heterologous host, where attachment is not supported, further support this hypothesis. Growth of mouse-SFB in rats also revealed that attachment is not required for the developmental transitions identified at the bacterial surface. This is consistent with the developmental transitions that occur during the life-cycle of the environmental bacterium *C. crescentus*, which attaches to abiotic surfaces using a stalk. In this species, adhesion is not necessary but the transition from the motile to sessile stage is promoted by surface contact[42]. Notably, the presence of a higher proportion of IOs without fully developed disordered HLS suggests a delay in SFB tip development when attachment is not supported. These results point towards the influence of internal and external signals in the regulation of the disordered HLS formation.

The tip length of mouse-SFB was also reduced when colonizing rats. A longer tip may provide better anchoring to the epithelium and higher access to host-derived nutrients. These may be critical for SFB development given that these bacteria lack most genes involved in the biosynthesis of nucleotides, amino acids, vitamins and cofactors[20–23].

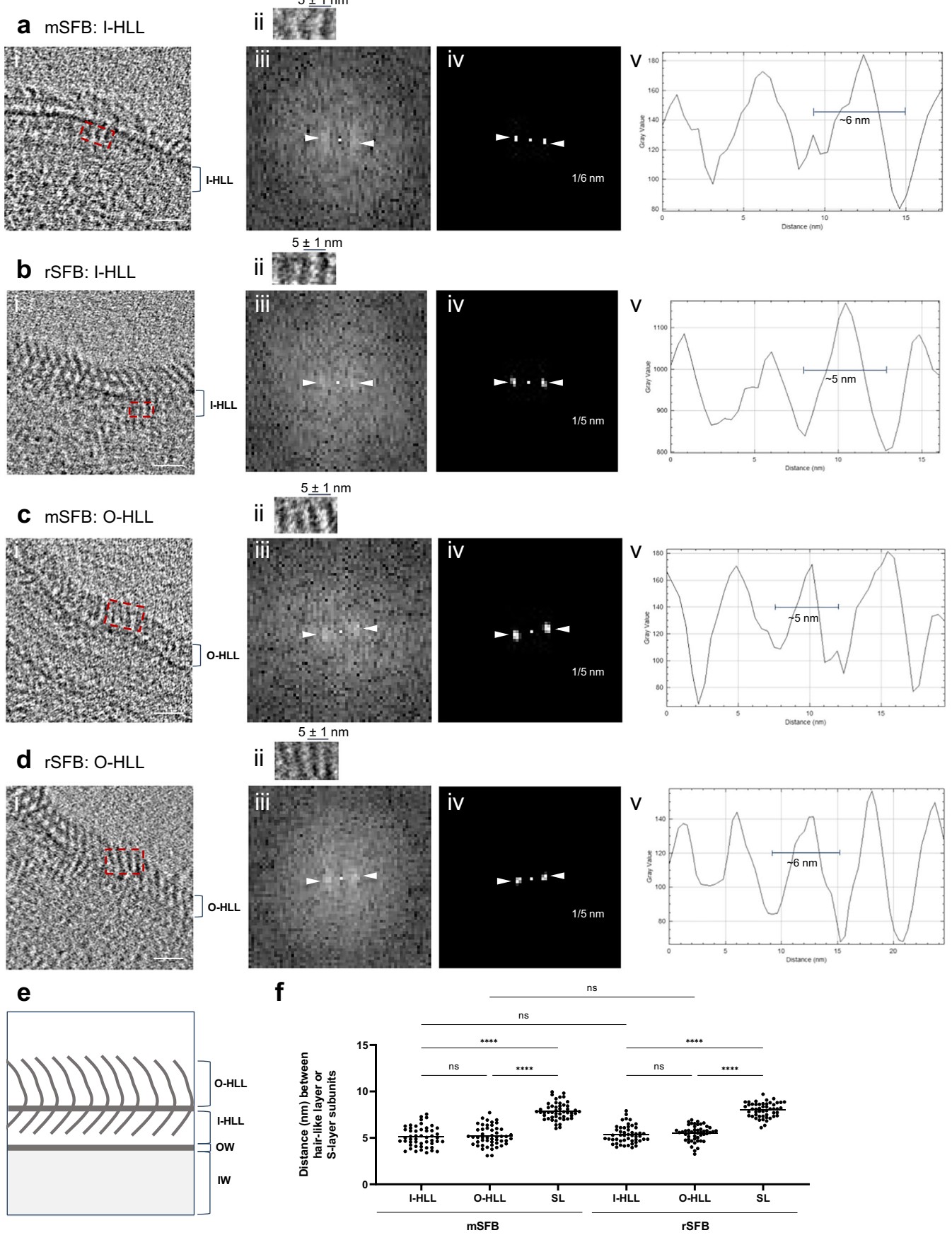

**Fig. 5 | Characterization of the filament-associated hair-like layer.** Close-ups of the tip side of: **a/c**(i) mouse-SFB and **b/d**(i) rat-SFB from reconstructed tomograms EMD-52677 and EMD-52699, respectively. The regions used for hair-like layer (HLL) analysis are delimited by red rectangles. Regions containing: **a/b**(i) the inner part of the hair-like layer (I-HLL) and **c/d**(i) the outer part of the hair-like layer (O-HLL) shown in (**a–d**(ii)). The average spacing and standard deviation between HLL subunits is shown. **a–d**(iii), Fast Fourier transform, **a–d**(iv) Maximum intensity peaks (indicated by white arrow heads) and their corresponding frequency, and **a–d**(v) Plot profiles are shown for: **a**(i) mouse-SFB I-HLL, **b**(i) rat-SFB I-HLL, **c**(i) mouse-SFB O-HLL, **d**(i) rat-SFB O-HLL. **e** Schematic representation of inner and outer parts of the HLL. OW outer wall zone; IW inner wall zone. **f** Distance between HLL and Surface layer (S-layer, SL) subunits in mouse-SFB and rat-SFB. Ten measurements were performed for each of the 5 SFB from 2 independent experiments (n = 50 measurements). The distance between S-layer subunits are also shown in Fig. 1k. The mean is shown and the statistical significance was assessed using the One-way ANOVA (****: $p < 0.0001$, ns: not significant). The most important comparisons are shown. Source data are provided as a Source Data file. mSFB mouse-SFB; rSFB rat-SFB; OW outer wall zone; IW inner wall zone. Scale bars: **a–d**(i): 20 nm.

Attachment-related cues may therefore play a role in the regulation of tip elongation, similar to what has been observed for the stalk length in *C. crescentus*[43]. Nevertheless, the tip of rat-SFB was shorter than the tip of mouse-SFB when the characterization was performed in their respective hosts. As the genetic background of the host can influence the colonization of microbiota members[44], we cannot exclude the possibility that the host environment also influences the development of morphological characteristics associated with the establishment of a niche, such as tip elongation.

Limitations of our work are related to the thickness of our samples and the size of the field of view as these prevented a more detailed characterization of the intracellular features identified and their association with the five SFB stages described. Future studies using cryo-FIB[45] and montage cryo-ET schemes[46] to increase the field of view without compromising tomogram resolution could overcome this limitation. The characterization of the SFB S-layer symmetry was also not conclusive. In-depth in situ symmetry identification could be performed for the S-layer covering the long stalk of *C. crescentus*[29], which is morphologically similar to the SFB tip. However, the strong SFB tip curvature, and smaller center-to-center spacing of the SFB S-layer (~8 nm) in comparison to the S-layer from *C. crescentus* (20-22 nm[29,47]), may have contributed to the difficulties encountered in the identification of the lattice symmetry for SFB.

Overall, this study identifies previously undescribed surface structures and other intracellular features, providing critical information on the SFB life-cycle. We describe a developmental structural change of the SFB surface from an S-layer to a hair-like layer that is associated with the SFB transition from the unicellular to filamentous stage and identify tip-restricted disordered HLS potentially involved in host attachment. Furthermore, we show a restricted surface accessibility of the Th17Ag specifically at the SFB filament tip, identifying the SFB tip as a particularly immunologically vulnerable site where SFB cell wall-associated proteins are accessible. Given that the major immunostimulatory properties of SFB are dependent on adhesion to host cells and that surface-exposed supramolecular structures generally play critical roles in host-bacterial interactions, future studies will focus on unraveling the molecular details of the surface structures identified to obtain further insights on how the SFB-host cell interaction is established.

## Methods
### SFB monocolonization
Germ-free C57BL/6J mice were bred and housed either at the animal facility of Institut Pasteur (IP) (agreement number 75-15-01) or at the animal facility of SFR Necker (agreement number A75-15-34), in conditions that answer to regulatory requirements (i.e., approved by the French Ministry of Agriculture). The light/dark cycle was 10/14, the temperature was controlled between 20 and 24 °C, and the relative humidity around 50%. Mice were selected based on availability and regardless of their sex. Germ-free male F344 rats between 5 and 9 weeks-old were obtained from the Plateforme Anaxem de l'Institut Micalis (INRAE, Jouy-en-Josas) and housed at the IP animal facility. Animal sex was not considered in the study design since SFB colonization was the only parameter of interest and no other host parameter was recorded. Monocolonization with mouse-SFB-NL[21,26,27] was established for 7-13-week-old germ-free mice. Monocolonization with

mouse-SFB-NL or rat-SFB-Yit[20,48] was established for 7-17-week-old germ-free rats. To achieve SFB monocolonization, all rodents were kept under axenic conditions and gavaged with previously frozen fecal pellets of rodents monocolonized with the SFB strain of interest resuspended in water. Monocolonization was verified by Gram staining (Sigma Aldrich) and culture-based techniques. All rodents were sacrificed between 7- and 18-days post gavage for further SFB purification.

### SFB purification from rodent intestinal content
SFB were purified from the intestinal content of monocolonized mice and rats (one or two rodents per biological replicate) as previously described[34] with some modifications. Briefly, all liquid reagents were pre-equilibrated overnight in a hypoxic chamber at 2% oxygen. Intestines were dissected in hypoxic conditions and contents homogenized in 20 mL of PBS 1X by gentle pipetting. The intestinal suspension was layered in one volume of 50% nycodenz (Proteogenix) and spun down at 4000 x *g* for 15 min in a swinging bucket at room temperature. The interphase was collected, homogenized, layered in ½ volume of 30% nycodenz and spun down in the conditions above. Pellets were discarded and the remaining sample was washed in PBS 1× and spun down at 100 x *g* for 5 min. Pellets were again discarded, and supernatants spun down at 4000 x *g* for 20 min. When necessary, samples were passed through a 5 µm filter (Cytiva) to obtain an IO-only fraction and concentrated at 4000 x *g* for 30 min. Reverse filtration was used to recover the filament-enriched fraction. The pellet was resuspended in PBS 1× and the suspension layered on 1 volume of 15% nycodenz and spun at 4000 x *g* for 15 min. The pellet was recovered, washed in PBS 1×, and the suspension was spun down as described above. Pellets corresponding to SFB fractions were resuspended in approximately 200 µL of PBS 1× and diluted from 1:2 to 1:10 depending on the purified SFB density verified by Gram staining. SFB suspensions were kept at 4 °C for a maximum of 24 h prior to preparation of cryo-EM samples.

### SFB preparation for Cryo-electron microscopy and tomography
Gold particles of 5 or 10 nm (ProteinA-gold, Aurion) were added to SFB samples in a 1:20 or 1:40 dilution, respectively, and gently homogenized. Four microliters of this suspension were deposited onto Lacey S166-3 mesh 300 copper grids (Agar Scientific) or Quantifoil R 2/1 or R2/2 200 mesh copper grids previously glow-discharged at 2 mA for 1 min using an ELMO glow discharge system (Cordouan Technologies). 2 µL of PBS 1× were deposited on the grid and back blotting was performed for 2–6 s using an automatic plunge freezer (EMGP, Leica). During blotting, the temperature and humidity of the chamber were set to 10 °C and 98%, respectively. The samples were then cryofixed at −180 °C in liquid ethane using an automatic plunge freezer (EMGP, Leica) and stored in liquid nitrogen until imaging. Grids containing frozen-hydrated SFB were subsequently clipped into autogrids (Thermo Fisher Scientific: TFS) whenever subsequent imaging was performed using a 300 kV Titan Krios electron microscope.

### Cryo-electron microscopy and tomography
Initial screening of the samples was performed using a 200 kV Tecnai F20 (TFS) equipped with a direct detector Falcon II (TFS) at the IP Ultrastructural BioImaging Platform. Projection images of the tip and

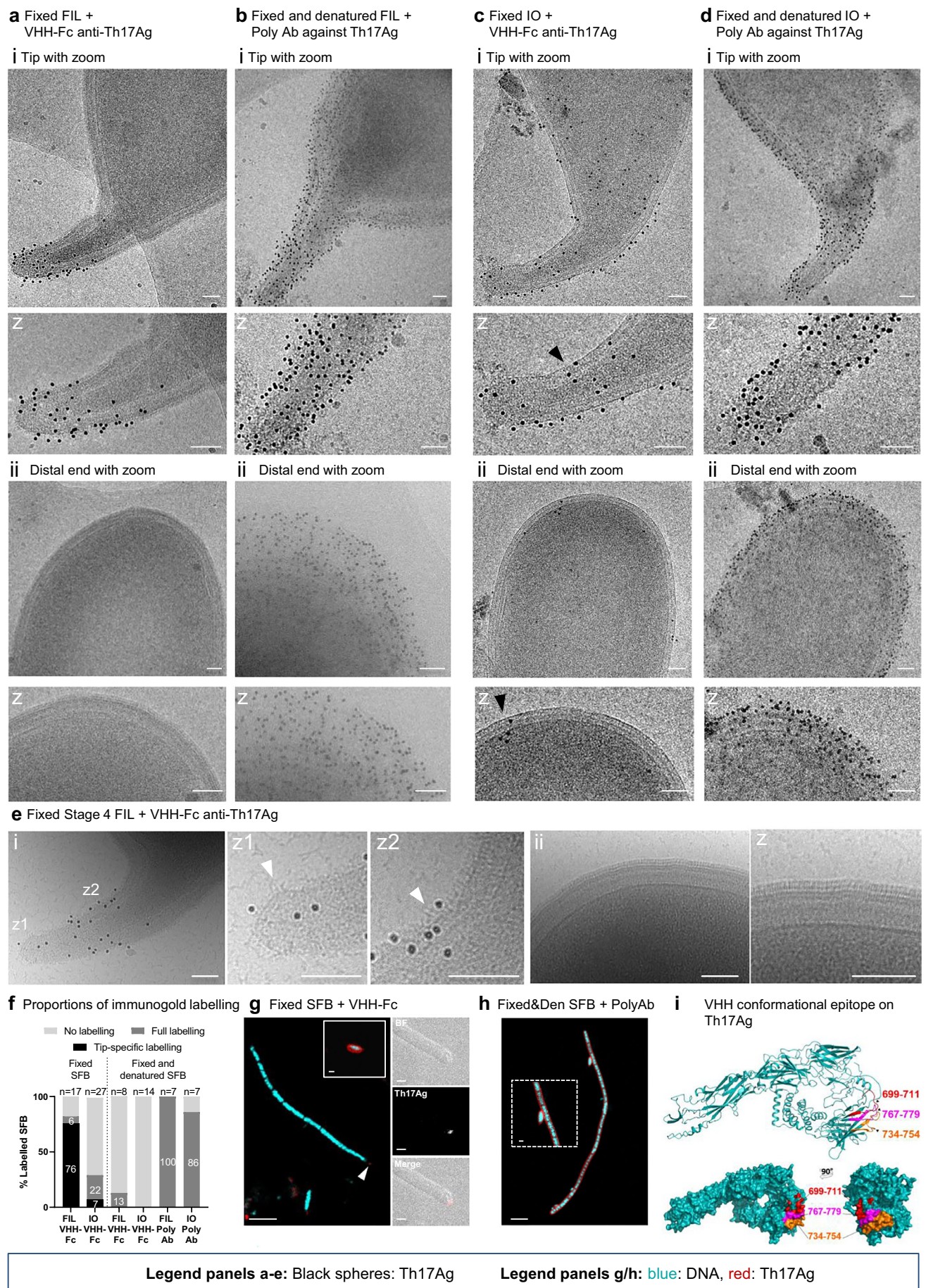

**a** Fixed FIL + VHH-Fc anti-Th17Ag
**i** Tip with zoom
z
**ii** Distal end with zoom
z

**b** Fixed and denatured FIL + Poly Ab against Th17Ag
**i** Tip with zoom
z
**ii** Distal end with zoom
z

**c** Fixed IO + VHH-Fc anti-Th17Ag
**i** Tip with zoom
z
**ii** Distal end with zoom
z

**d** Fixed and denatured IO + Poly Ab against Th17Ag
**i** Tip with zoom
z
**ii** Distal end with zoom
z

**e** Fixed Stage 4 FIL + VHH-Fc anti-Th17Ag
i   z1   z2   ii   z

**f** Proportions of immunogold labelling

No labelling   Full labelling
Tip-specific labelling

**g** Fixed SFB + VHH-Fc

**h** Fixed&Den SFB + PolyAb

**i** VHH conformational epitope on Th17Ag
699-711
767-779
734-754

**Legend panels a-e:** Black spheres: Th17Ag   **Legend panels g/h:** blue: DNA, red: Th17Ag

**Fig. 6 | Localization of the Th17Ag at the SFB surface.** Projection images from purified SFB **a**, **b**, **e** filaments (FIL) and **c**, **d** IOs stained using immunogold labeling with gold-conjugated protein A. Fixed SFB **a**, **e** filaments and **c** IOs were labeled with an anti-Th17Ag nanobody fused to the Fc region of human IgG1 (VHH-Fc). Fixed and denatured (Den) SFB **b** filaments and **d** IOs were labeled with a rabbit polyclonal antibody (Poly Ab) against Th17Ag. Imaging was performed at the SFB **a**–**e**(i) tip and **a**–**e**(ii) distal end. Close-ups (z) of the SFB tip and distal end are shown under or next to each panel. Examples of labeled regions without visible hair-like structures are shown by black arrow heads. **e** Examples of labeled regions with hair-like structures are shown by white arrow heads in a representative stage 4 filament. All projection images were acquired with a Tecnai F20 electron microscope equipped with a Falcon 2 camera with the exception of those included in (**e**). **f** Assessment of immunogold labeling for SFB imaged in the conditions described for **a**–**d** and after labeling of fixed and denatured SFB with a VHH-Fc anti-Th17Ag. The percentage of SFB labeled and labeled specifically at the tip are indicated on the corresponding bar. SFB were considered labeled if co-localization with at least 20 gold particles (black spheres) was observed, or at least 10 gold particles if labeling was restricted to the SFB tip. The number of SFB imaged per condition is included in the graph. Individual data points corresponding to the percentages shown are in the Supplementary Data 2 file. Source data are provided as a Source Data file. Immunofluorescence images of **g** fixed SFB incubated with a biotinylated VHH-Fc anti-Th17Ag and Streptavidin-Alexa568 and **h** fixed and denatured SFB incubated with a rabbit polyclonal antibody (Ab) against the Th17Ag and a secondary antibody anti-rabbit conjugated with Alexa 568. All SFB were additionally labeled with DAPI. An image of a labeled IO was included as an insert delimited by a white line (**g**). Close-ups of the tip of an SFB filament showing Th17Ag labeling (white arrow head) were included next to the main panel. Images showing the signal from bright field (BF), Alexa-568 and corresponding merged image are shown. A close-up of the IO shown in (**h**) was included as an insert delimited by a dashed white line. For all immunogold and immunofluorescence experiments, labeling was performed in two distinct days using two biological replicates for each experiment. **i** Localization of the nanobody binding epitope on the structure of the Th17Ag predicted by AlphaFold[59]. Structure prediction with surface representations of the Th17Ag highlighting the protein regions (in red, orange and magenta) that form the conformational epitope recognized by the nanobody. Epitope location was determined by Hydrogen/Deuterium eXchange-Mass Spectrometry (HDX-MS). The Th17Ag sequence used to generate the AlphaFold model is provided in the Source Data file. Scale bars: **a**–**e**: 100 nm; **g**, **h**: 5 μm for main panel and 1 μm for inserts.

distal end of SFB were acquired using the software EPU (TFS) at a pixel size of 2.01 or 4.14 Å and are listed in Supplementary Data 1. Representative images of the identified phenotypes (Supplementary Data 1) were acquired with a 300 kV TITAN Krios electron microscope equipped with a Falcon 4i direct electron detector and a Selectris X energy filter (10 eV slit width), here designated as TITAN Krios IP, unless otherwise stated. Imaging was performed at the IP NanoImaging Core facility. Projection images of the tip and distal end of SFB were acquired with −10 μm defocus at a calibrated pixel size of 1.59 or 2.01 Å. Data acquisition was controlled by TOMO5 (version 5.14, 5.17 or 5.22, TFS). The magnification used to obtain a full SFB in the field of view to determine SFB length varied according to the bacterial size (from less than 2 μm to more than 65 μm in length). Projection images were used to classify tip stages and to measure SFB length, tip length and vesicle diameter.

To obtain a 3D view and enable the measurement of the SFB structures observed, tilt series were acquired mainly at the SFB tip at high magnifications. We used either the TITAN Krios IP or a Titan Krios microscope (TFS) operated at 300 kV equipped with a post-column energy filter (20 eV slit width, Gatan) and a K2 direct electron detector (Gatan) at the University of Zurich, here designated as TITAN Krios UZH. For the latter microscope, data acquisition was controlled by SerialEM 3.8[49]. The tilt series acquisition parameters used for the TITAN Krios IP (a) and the TITAN Krios UZH (b) were set as follows. Tilt series were acquired at a calibrated pixel size of 1.59 Å (a), 1.75 Å (b), 1.9 Å (a) or 2.21 Å (b) (Supplementary Data 1). Two types of tilt-series acquisition schemes were used: a dose-symmetric scheme[50] starting at 0° from −51° to +51° with a 3° increment (a) and a bidirectional scheme starting at −30° from −60° to +60° with a 3° increment (b). The projection dose was calibrated for each sample to reach a total dose of approximately 120 to 140 e⁻/A² and the defocus used was between −2 and −6 μm.

### Cryo-EM data processing and visualization

For each tilt series acquired, frame alignment was performed using either MotionCor2 1.6.4 or the Alignframes program from the IMOD software package version 4.11 (RRID:SCR_003297)[51]. IMOD was also used for tilt alignment using fiducial markers and tomogram reconstruction at a binning factor of 2 or 4 with an antialiasing filter using dose weighting and a SIRT-like filter with 10-20 iterations. For tomograms selected as representative (binned by a factor of 4, Supplementary Data 1), fiducials were erased with the IMOD findbeads3d tool. These tomograms were further subjected to topaz denoising[52] using a default 10 Å/px pre-trained model, for visualization purposes. A

maximum of 10 tomographic slices were summed using the IMOD slicer tool to increase contrast and shown as representative tomographic slices of the tomograms containing features of interest. Tomographic slices may have been rotated along the x and y axis for presentation and comparison purposes.

For representative tomograms showing tip stages, intracellular and extracellular features, IMOD models were manually prepared for the membrane, cell wall and outer layer and for the features to be highlighted in each model. Linear interpolation was performed for all models except for vesicles (in which spherical interpolation was performed), flagella, filament-like structures and tracks (for which no interpolation was performed). IMOD meshes were prepared for all features except tracks. Upon volume meshing, the cap and tube tool were used for filament-like structures and vesicles and for flagella, respectively, to ensure the features were represented as closed models. IMOD models for the three layers (membrane, cell wall and outer layer), vesicle membranes, intracellular filaments and flagella were used as segmentation masks in UCSF ChimeraX version 1.7 (RRID:SCR_015872)[53] to prepare volume masks using the slab tool with the width measured for each feature. The outer layer corresponded to the electron dense layer seen in the S-layer and in the hair-like layer. The pad tool was used to represent the tomogram densities from the cytosol, the interior of extracellular vesicles, and the chemosensory array while the coating of extracellular vesicles was defined using the slab tool. Tracks and hair-like structures were represented as IMOD contours, and their thickness adjusted using the atom and stickRadius commands. Final segmentation models were clipped, rotated and colored to show the features of interest.

Projection images selected as representative were binned 4 times and opened in FIJI version 2.14.0 (RRID:SCR_002285)[54]. Contrast was adjusted and a Gaussian low-pass filter with 0.5 to 1 px radius was applied using Fiji's unsharped mask tool, for visualization purposes. No image treatment was applied to images acquired using the 200 kV Tecnai F20 (TFS). For two representative images of morphological stages, IMOD models were prepared for the membrane, cell wall, outer layer and hair-like structures. IMOD models were opened as contours in Chimera X and their thickness adjusted using the atom and stickRadius commands. The cytoplasm of each original projection image was isolated using the imodmop command and included in the segmentation model prepared. A cytoplasm mask was defined and the segmentation models colored to show the features of interest.

All schematics were prepared using PowerPoint Version 2501.

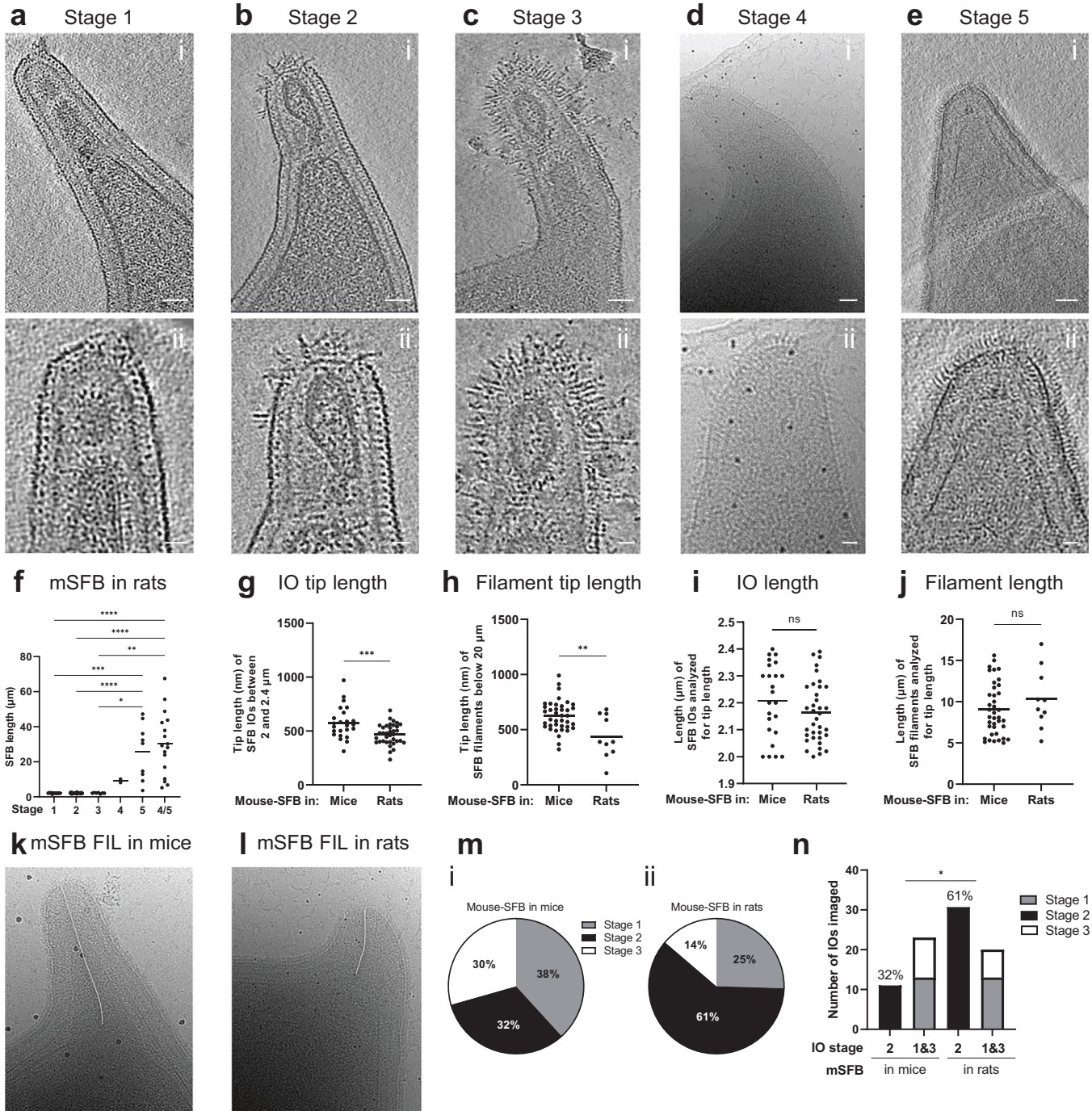

**Fig. 7 | Effect of SFB growth in a heterologous host on tip morphology.**
**a–e** Mouse-SFB morphological stages, when SFB was grown in rats, equivalent to those identified at the tip of mouse-SFB colonizing mice. **a–c/e**(i), Representative tomographic slices from reconstructed tomograms showing SFB tip of stages: 1 (EMD-52700), 2 (EMD-52701), 3 (EMD-52702) and 5 (EMD-52703). **d**(i) Representative projection image showing SFB tip of stage 4. The selected representative SFB from stages 1–5 have a length of 2.1, 2.3, 1.6, 8.2, and 3.7 μm. **a–e**(ii) Close-ups of the SFB tip shown in (**a–e**(i)). **f** Length of mouse-SFB grown in rats (n = 79) from 4 independent experiments assigned to each tip stage, including filaments where no distinction could be made between stage 4 and 5. The statistical significance was assessed using the Kruskal–Wallis test followed by a Dunn's test correction for multiple comparisons (stage 1 vs. 5, p = 0.0006; stage 2 vs. 5, p < 0.0001; stage 3 vs. 5. p = 0.0181, stage 1 or 2 vs. stage 4/5, p < 0.0001; stage 3 vs. 4/5, p = 0.0027). Length of the tip of mouse-SFB **g** IOs and **h** filaments analysed. Length of corresponding SFB **i** IOs and **j** filaments used for tip length analysis of SFB grown in mice

and rats. IOs between 2 and 2.4 μm of length were included in the analysis (n = 27 IOs in mice, n = 38 IOs in rats). Filaments below 20 μm length were included in the analysis (n = 39 filaments in mice and n = 10 filaments in rats). The statistical significance was assessed using an unpaired two-sided t-test for (**g**, **i**) (**g**: p = 0.0004, **i**: ns: not significant) and using a two-sided Mann–Whitney U test for (**h**, **j**) (**h**: p = 0.0087, **j**: ns: not significant). For **f–j** individual measurements and the corresponding mean are shown. Representative projection images of mouse-SFB (mSFB) filaments (FIL) grown in **k** mice and **l** rats. The tip length measurement is shown by a white line. **m** Proportions of IOs assigned to Stages 1-3 when grown (**m**(i)) in mice and (**m**(ii)) in rats. **n** Comparison of the proportions of Stage 2 IOs of mouse-SFB grown in mice (n = 33) and in rats (n = 51). The statistical significance was assessed using a two-sided Fisher's exact test (Stage 2 in mice vs in rats, p = 0.0147). Source data are provided as a Source Data file for (**f–j**, **m**, **n**). The same data were used to prepare (**m**, **n**). Scale bars: **a–e**(i), **k**, **l**: 50 nm; **a–e**(ii): 20 nm.

## Cryo-EM data analysis

To measure SFB length, projection images containing a full SFB in the field of view were opened in IMOD or in FIJI and a contour/line was traced from the SFB distal end to the edge of the tip. Tip length was measured by tracing a contour/line from the base to the tip edge, as shown in Fig. 7k-l. To measure vesicle diameters, projection images or tomograms (binned 4 times) acquired in a 300 kV electron microscope were open in IMOD, a circle shape was used to establish the contour of each vesicle at its maximum diameter and its area was measured using the imodinfo tool. Vesicle diameter was then deduced from the circle area formula ($\pi r^2$).

Measurements of intracellular filaments, hair-like structures length and width and distance between hair-like structures and S-layer subunits from SFB tip tomograms (binned 4 times) were performed manually in IMOD by defining individual contours and using the imodinfo tool. SFB tip diameter was measured as described above in a tomographic slice from the center of the volume. At least five tomograms were selected for the measurements of each feature/stage with the exception of plate-like structures since these were only present in three tomograms for mouse-SFB-NL and two tomograms for rat-SFB-Yit. Ten measurements were performed to estimate the width of tracks, intracellular filaments, and plate-like structures identified in each of the tomograms selected. The average of the measurements performed for each tomogram was used for statistical analysis. Tip diameter and length and vesicle diameter was measured once per tomogram/feature. Ten measurements were performed for each filament-like structure and for the distance between S-layer and hair-like layer subunits. Ten measurements in five different tomographic slices were performed to estimate hair-like structures length and width. Individual measurement values and their corresponding mean were plotted while mean and standard deviation are indicated in the text. The distance between S-layer and hair-like layer subunits was also confirmed by selecting regions from tomograms binned by a factor of 2 that contained clear S-layer or HLL repeats and by using a Fast Fourier transform in Fiji to assess repetitiveness. Plot profiles of the selected regions were obtained and the distances between two peaks were measured.

## Histological analysis

Tissue from the terminal ileum of rodents monocolonized with SFB was fixed in 4% paraformaldehyde (Electron Microscopy Sciences) and frozen in OCT (TFS). Sections of 10 μm thickness were cut using a cryostat (Leica), fixed in 4% paraformaldehyde, stained for 10 min using Giemsa staining (Sigma Aldrich), and mounted using Eukitt Quick-hardening mounting medium (Sigma Aldrich). Imaging was performed at room temperature using an Olympus BX53 microscope equipped with an Olympus cellSens Standard 4.2.1 software. The Olympus UPLFLN 40× objective (0.75 NA) and the UPLFLN 100X objective (1.3 NA) were used.

## Protein expression and purification of the Th17Ag

The Th17Ag gene from mouse-SFB-NL, without the signal peptide sequence, was synthesized with *Escherichia coli* codon optimization and cloned in the plasmid pET151/D-TOPO (GeneArt, TFS). Protein expression of the Th17Ag in the BL21 (DE3) Star *E. coli* strain was performed at 16 °C in Luria-Bertani broth with an overnight induction using 1 mM Isopropyl β-d-1-thiogalactopyranoside (IPTG) (Sigma Aldrich). The recombinantly expressed Th17Ag was produced with a poly-histidine and V5 epitope-containing N-terminal tag. The tagged protein was purified by affinity chromatography followed by size-exclusion using a Cobalt 5 mL HiTrap TALON crude column (Cytiva) and a Superdex 200 Increase 10/300 GL column (Cytiva), respectively. Protein purification was performed using an AKTA-FPLC purification system equipped with a UNICORN 5.31 following the instructions of the manufacturer. On-tube digestion using a GST-tagged Tobacco Etch

Virus (TEV) protease (PX P1108, Proteogenix) was performed to cleave the tag from purified Th17Ag using 1 TEV unit per 6 μg of purified protein. Removal of the remaining tags and TEV was achieved through re-purification by affinity chromatography using a Cobalt 5 mL HiTrap TALON crude column (Cytiva) and a 5 mL GSTrap HP column (Cytiva). Protein-containing fractions were analyzed by sodium dodecyl sulfate–polyacrylamide gel electrophoresis (SDS-PAGE) and western blot using a mouse monoclonal anti-polyHistidine antibody (Sigma Aldrich, 1:5000, catalog number: A7058, RRID:AB_258326) and a horse anti-mouse IgG conjugated with horseradish peroxidase (HRP) (Cell Signaling, 1:20000, catalog number: #7076). Images of the Coomassie-stained gels and of the protein blots were acquired using a ChemiDoc (BioRad). The tagged protein was used for subsequent experiments unless otherwise stated.

## ELISA with polyclonal antibody against the Th17Ag

Purified Th17Ag was coated onto Nunc MaxiSorp flat-bottom plates (TFS) at a concentration of 10 μg/mL at 4 °C overnight. Coated plates were washed four times with 0.05% tween 20 (Sigma Aldrich) in PBS 1X (TFS) and incubated for 1 h at room temperature in blocking buffer (0.05% tween 20 in PBS 1× with 1% bovine serum albumin, BSA, Sigma Aldrich). Rabbit polyclonal antibodies produced against the Th17Ag from mouse-SFB-NYU (1:100 of a 1:1 mix containing two rabbit anti-3340 antibodies)[7] and a goat anti-rabbit IgG (whole molecule) conjugated with alkaline phosphatase (Sigma Aldrich, 1:10 000, catalog number: A3687, RRID: AB_258103) were consecutively added to each well in blocking buffer for 2 h and 1 h, respectively. Four washing steps were performed after each incubation. A solution of 4.5 M NaCl (Sigma Aldrich), 0.1 M Tris-HCl (TFS) and 1 mg/mL 4-nitrophenyl phosphate disodium salt (Sigma Aldrich) was used to reveal binding. The absorbance was measured at 415 nm using an iMark Microplate Absorbance Reader (BioRad) equipped with an MPM6 software (Version 6.3, BioRad). The polyclonal antibody anti-Th17Ag used was kindly provided by Dr. Dan R Littman.

## Nanobody library and phage selection

An alpaca (*Lama pacos*) named Picchu was immunized three times by subcutaneous injection with SFB purified from the intestinal content of monocolonized mice ($1 \times 10^9$ to $4 \times 10^{10}$ SFB genomes) and fixed with 4% paraformaldehyde (PFA). Two extra booster immunizations were performed 3 months later with SFB purified from the SFB-TC7 cell co-culture ($3 \times 10^7$ to $2 \times 10^9$ SFB genomes) described by Schnupf et al.[11] and fixed with 4% PFA. SFB were mixed with complete Freund's adjuvant for the first immunization and with incomplete Freund's adjuvant for subsequent injections. Approximately 250 mL of alpaca blood were collected 6 days after the last immunization. Peripheral blood lymphocytes were isolated by Ficoll (Sigma Aldrich) density gradient centrifugation and stored at −80 °C until further use. Total RNA was extracted from the lymphocytes and converted to cDNA as described by Lafaye et al.[55]. DNA sequences encoding the nanobodies, naturally produced by the immunized alpaca (VHH, variable domain of heavy chain), were amplified by PCR from cDNA using 8 pairs of primers (Supplementary Table 4), and cloned into the pHEN6 phagemid vector. The pHEN6 vector features a pUC backbone for ampicillin selection and high-copy replication in *E. coli* and includes the f1 phage origin for single-stranded DNA production. The ligation mixture was then electroporated into *E. coli* TG1 to obtain a VHH library of approximately $10^8$ different clones. Phage-nanobodies from the VHH library were produced using the helper phage M13K7 (New England Biolabs) and Th17Ag-binding VHH were selected by phage display[56]. Briefly, $10^{11}$ Phage-VHH were incubated with purified Th17Ag previously coated on Nunc-Immunotubes (TFS) for 1 h at room temperature. Three rounds of panning were performed with increasingly lower concentrations of coated Th17Ag, starting at 10 μg/mL, and increasingly higher numbers of extensive washes. Th17Ag-binding phage-VHH were eluted in 100 mM triethylamine (TEA, Sigma Aldrich)

and used to infect *E. coli* TG1. Individual colonies were used to produce phage-VHH in 96-wells plates, which were then tested by phage-ELISA for binding to the Th17Ag previously coated onto 96-wells Nunc Max-iSorp flat-bottom plates (TFS) using an anti-M13 bacteriophage mouse monoclonal antibody conjugated with HRP (Abcam, 1:2500, catalog number: ab305291). The sequence of positive clones was determined by Sanger sequencing using the M13rev-29 primer (5' CAG GAA ACA GCT ATG ACC 3') (Eurofins).

### Nanobody expression and selection
Positive VHH were expressed from pHEN6 vectors with an N-terminal *pelB* signal sequence and a C-terminal c-Myc and poly-histidine-containing tag. Periplasmic expression was performed in *E. coli* TG1 after overnight induction at 30 °C with 1 mM IPTG (Euromedex). Bacterial pellets were resuspended in PBS 1× containing 300 mM NaCl, 1 mg/mL polymyxin B sulfate (Sigma Aldrich) and Complete, EDTA-free Protease Inhibitor Cocktail (Roche) and incubated at 4 °C for 1 h with vigorous shaking (300 rpm). Periplasmic extracts obtained by centrifugation and used to purify the produced nanobodies by affinity chromatography using a Cobalt 5 mL HiTrap TALON crude column (Cytiva) followed by size exclusion chromatography on a Superdex 75 pg column (Cytiva). An AKTA-Start purification system equipped with a UNICORN start 3.1 was used according to the manufacturer instructions. Purified VHH were concentrated and stored in PBS 1× at −20 °C until further use.

Binding of purified VHH (10 μg/mL) to the Th17Ag was assessed by ELISA using a mouse monoclonal antibody anti-c-Myc (Bio-Techne, 1:500, catalog number: #NB600-302) and a goat polyclonal anti-mouse IgG (whole molecule) conjugated with alkaline phosphatase (Sigma Aldrich, 1:10 000, catalog number: A9316, RRID: AB_258446).

To enable large scale VHH production, a positive VHH, with c-Myc and 6xHis tag followed by a stop codon, was subcloned into the vector pFuse, which was transfected into Expi293 cells using the Expi293 Expression System Kit (TFS, catalog number: A14527), according to the manufacturer's instructions. Expi293 cells were grown at 37 °C for 5 days and VHH-Fc were purified from their supernatants by affinity chromatography as described above. Purified VHH were concentrated and stored in PBS 1× at −20 °C until further use.

### Expression of nanobodies fused to Fc region
The coding region of the selected VHH was subcloned into the vector pFuse[57] in order to allow the expression of dimeric VHH-Fc fusion proteins. VHH-Fc were expressed in Expi293 cells as described above, and purified using a HiTrap protein A HP (Cytiva) and an AKTA-Start purification system, according to the manufacturer instructions. Purified VHH were concentrated and stored in PBS 1× at −20 °C until further use.

Binding of purified VHH-Fc (10 μg/mL) to the Th17Ag was assessed by ELISA using a mouse monoclonal antibody anti-human IgG1 Fc conjugated with HRP (TFS, 1:1000, catalog number: A-10648, RRID: AB_2534051) and Nunc MaxiSorp flat-bottom plates (TFS) coated with 10 μg/mL of purified protein. 3,3′,5,5′-Tetramethylbenzidine (TMB, Sigma Aldrich) was used to reveal binding following the manufacturer's instructions. The absorbance was measured at 450 nm.

VHH-Fc was biotinylated using EZ-Link Sulfo NHS-Biotin (TFS) according to the manufacturer guidelines for subsequent use in immunofluorescence.

### Immunofluorescence and immunogold labeling of the Th17Ag
Purified SFB from monocolonized mice (IOs only and filament-enriched fractions) were fixed in 4% PFA at room temperature for 30 min and kept in 1% PFA at 4 °C overnight. SFB samples were washed with PBS 1x and either left untreated (fixed SFB) or incubated in PBS 1X (TFS) with 0.5% triton X-100 (Sigma Aldrich) at 95 °C for 5 min followed by a PBS 1× wash (fixed and denatured SFB). Immunolabelling was performed in 1.5 mL tubes using PBS 1× with 1% BSA (Sigma Aldrich) as

blocking buffer and PBS 1× as washing buffer. After 1 h in blocking buffer, SFB were incubated either with rabbit polyclonal (1:100) or with VHH-Fc (10 μg/mL) in blocking buffer for 2 h at room temperature. For immunogold labeling, Protein A coupled to 10 nm gold particles (Aurion, 1:20) was added for 1 h in blocking buffer. For immunofluorescence, streptavidin coupled to Alexa 568 (1:200, TFS, catalog number S11226, RRID: AB_2315774) and a goat polyclonal anti-rabbit IgG (H + L) conjugated with Alexa Fluor 568 (1:500, TFS, catalog number: A-11036, RRID: AB_10563566) was used for detection of labeling with the biotinylated VHH-Fc and the polyclonal antibody, respectively. DAPI (4 μg/mL, TFS) was added together with the fluorescently coupled antibody/streptavidin to make the identification of SFB easier. Two washing steps were performed after each incubation.

For immunogold labeling, samples were diluted from 1:4 to 1:2 and cryo-EM grids were prepared immediately after staining, as described above, and imaged using a 200 kV Tecnai F20 (TFS) equipped with a direct detector Falcon II (TFS).

For immunofluorescence, samples were diluted from 1:10 to 1:20, dried onto glass slides and preserved in ProLong Gold antifade medium (TFS) at 4 °C until imaging by confocal microscopy. Imaging was performed at room temperature using an SP8 confocal microscope (Leica) equipped with both PMTs and HyD detection systems and a LAS X acquisition software (version 3.5.7). Slides were imaged at room temperature using a HC PL APO 63x (1.40 NA) objective and z-stacks were acquired at 0.3 μm intervals with scanning at 1024 × 1024 pixels and a zoom factor of 2.5 or at 2048 × 2048 pixels and a zoom factor of 1.25. DAPI and the fluorescent dye Alexa Fluor 568 were excited using 405 nm and 561 nm lasers, respectively.

### Quantification of immunogold labeling
Representative projection images acquired using a 200 kV Tecnai F20 (TFS) were selected for each immunogold labeling condition and the ccderaser function of imod was used to erase the black pixel located at the center of the original images. Counting of gold particles co-localized with SFB cells was performed manually in IMOD. Images with over 10 gold particles in the background were not considered for analysis. Additionally, to avoid considering poorly labeled SFB, SFB were only considered labeled if co-localization with at least 20 gold particles was observed, or at least 10 gold particles if labeling was restricted to the SFB tip.

### VHH binding kinetics
The binding kinetics of the anti-Th17Ag VHH were obtained by Biolayer Interferometry (BLI) using an Octet HTX system (Sartorius) equipped with an Octet Data Acquisition software version 13.0 (Sartorius) at 30 °C. The His-tagged VHH was immobilized onto Ni-NTA sensor tips at 5 μg/mL for 180 s following a baseline in PBS 1× supplemented with tween 20 at 0.1% (PBS/T) for 120 s. The VHH-coated tips were then dipped into different concentrations of the Th17Ag (2264, 905, 226.4, 90.5 and 0 nM) for 600 s to allow association. Subsequently, biosensors were placed in PBS/T for 15 min to initiate dissociation. Raw data were processed using Octet Data Analysis Studio (version 13.0) and the data were fitted to a 1:1 Langmuir binding model.

### Identification of VHH binding epitopes on the Th17Ag
A summary of the Hydrogen/Deuterium eXchange-Mass Spectrometry (HDX-MS) data is provided in Supplementary Table 3[58].The quality of the Th17Ag was assessed by intact mass measurement (measured mass: 113 762.00 Da; expected mass: 113 760.7381 Da; Δm = 1.26 Da 11.1 ppm). The Th17Ag alone (7.1 μM in PBS1×) or in complex with a 1.5× molar excess VHH was equilibrated for 30 min at room temperature. Continuous labeling was initiated by adding 107 μL of deuterated buffer (PBS1×, pD 7.45) to 13 μL of equilibrated protein solution, resulting in a final $D_2O/H_2O$ ratio of 90/10. The experimental $K_D$ value was used to adjust the protein concentrations so that more than 90%

of complex remained during labeling. The exchange reaction was quenched after 10 seconds, 1, 10, 30, 60 and 120 min labeling at room temperature by mixing 20 μL of the labeling reaction with 40 μL of an ice-cold solution of 0.15% formic acid and 6 M guanidinium chloride to reduce the pH to 2.5. A similar procedure was used for the preparation of undeuterated control samples. Quenched samples were snap-frozen in liquid nitrogen and stored at −80 °C until further use. Triplicate labeling experiments were performed for each time point and condition (1 biological and 2 technical replicates).

Quenched samples were injected onto a nanoACQUITY UPLC system (Waters Corporation) equipped with an HDX manager maintained at 0 °C. Approximately 11 pmol of labeled Th17Ag either alone or with 16.5 pmol of VHH were digested using an in-house packed cartridge (2.0 × 20 mm, 63 μL bed volume) of immobilized pepsin beads (TFS) for 2 min at 20 °C. Peptides were directly trapped and desalted onto a C18 trap column (VanGuard BEH 1.7 mm, 2.1 × 5 mm, Waters Corporation) at a flow rate of 100 μL/min (0.15% formic acid, pH 2.5) and separated by a 8 min linear gradient of acetonitrile from 5 to 30%, followed by a 2 min increase from 30 to 40%, at 40 μL/min using an ACQUITY UPLC BEH C18 analytical column (1.7 μm, 1.0 mm × 100 mm, Waters Corporation). After each run, the pepsin column was manually cleaned with two consecutive washes of 1.0% formic acid, 5% acetonitrile, 1.5 M guanidinium chloride. Blank injections were performed after each sample to confirm the absence of carry-over.

Mass spectra were acquired in resolution and positive ion mode ($m/z$ 50-1950) on a Synapt G2-Si HDMS mass spectrometer (Waters Corporation) equipped with a standard ESI source and lock-mass correction. Peptides were identified in undeuterated samples by a combination of data independent acquisition ($MS^E$) and exact mass measurement (below 5.0 ppm mass error) using the same chromatographic conditions as for the deuterated samples. To maximize sequence coverage, the fragmentation of Th17Ag peptide ions was conducted using four distinct collision energy ramps in the trap region of the instrument: Low: 10–30 V; Med: 15–35 V; High: 20–45 V; Mix: 10–45 V.

The initial peptide map of the Th17Ag was generated by database searching in ProteinLynX Global server 3.0 (Waters corporation) using the following processing and workflow parameters: low and elevated intensity thresholds set to 250.0 and 100.0 counts; intensity threshold = 750.0 counts; automatic peptide and fragment tolerance; non-specific primary digest reagent; false discovery rate = 4%. Each fragmentation spectrum was manually inspected for assignment confirmation. The peptide map was refined in DynamX 3.0 (Waters corporation) using the following import PLGS results filters: minimum intensity = 10,000; minimum products per amino acid value = 0.3; minimum PLGS score = 7.0; maximum MH + error = 5.0 ppm. A total of 154 peptides covering 90.6% of the Th17Ag protein sequence were selected and analyzed by HDX-MS.

DynamX 3.0 was used to extract the centroid masses of all peptides selected for HDX-MS analyses. Results are reported as relative deuterium exchange levels (no back exchange correction) expressed in either mass unit or as fractional exchange. Fractional exchange values were calculated by dividing the experimental uptake value by the theoretically maximum number of exchangeable backbone amide hydrogens that could be replaced in each peptide, considering the final excess of deuterium present in the labeling mixture.

To facilitate HDX-MS data interpretation, models of the Th17Ag were generated with Alphafold[59] using the default parameters without energy minimization.

## Statistics and Reproducibility
Data was plotted and statistical analysis was performed using GraphPad Prism 9.10 (GraphPad Software Inc, *: $p < 0.05$, **: $p < 0.01$, ***: $p < 0.001$, ****: $p < 0.0001$). Statistical significance was assessed for individual measurements following a normal distribution using an unpaired t-test or a one-way ANOVA in cases where only two or multiple groups were compared, respectively. Statistical significance was assessed for individual measurements that did not pass a normality test using a Mann-Whitney U test or a Kruskal–Wallis test followed by a Dunn's test in cases where only two or multiple groups were compared, respectively. To compare proportions between two groups, a Fisher's exact test was used. The MEMHDX software (version 0.01) was used to statistically validate HDX-MS datasets (Wald test, false discovery rate of 5%)[60]. Supplementary Data 1 and Supplementary Data 2 detail the number of SFB displaying each phenotype and feature described, and the corresponding number of biological replicates ("Experiment no." column) for the cryo-EM/ET and the immunogold labeling datasets, respectively. The immunofluorescence and histological staining experiments were repeated for two biological replicates showing similar results.

## Ethics statement
Animal experiments were performed in accordance with French and European regulations on the protection of the animals used for scientific purposes (Directive 2010/63 of the European Parliament and French decree of February 1, 2013). Rodent (dap210054 and dap220096) and alpaca (2020-27412) experiments were approved by the IP ethical committee for animal experimentation (Comité d'Ethique en Expérimentation Animale, CETEA, registry number #89) and authorized by the Ministère de l'Enseignement Supérieur, de la Recherche et de l'Innovation.

## Reporting summary
Further information on research design is available in the Nature Portfolio Reporting Summary linked to this article.

## Data availability
Reconstructed tomograms used as representative have been deposited in the EMDB under the following accession codes: EMD-52655; EMD-52667; EMD-52668; EMD-52669; EMD-52670; EMD-52671; EMD-52673; EMD-52674; EMD-52675; EMD-52676; EMD-52677; EMD-52678; EMD-52679; EMD-52680; EMD-52682; EMD-52683; EMD-52684; EMD-52685; EMD-52687; EMD-52688; EMD-52689; EMD-52690; EMD-52691; EMD-52692; EMD-52693; EMD-52694; EMD-52695; EMD-52696; EMD-52697; EMD-52698; EMD-52699; EMD-52700; EMD-52701; EMD-52702; EMD-52703; EMD-52856; EMD-54603; EMD-54605; EMD-54607; EMD-54608. The EMDB accession codes for reconstructed tomograms were indicated in the figures which include their corresponding tomographic slices. The mass spectrometry data have been deposited in the ProteomeXchange Consortium via the PRIDE[61] partner repository under the accession code PXD060041. The characteristics of all the bacteria included in the cryo-EM/cryo-ET dataset and the immunogold labeling dataset are included in Supplementary Data 1 and Supplementary Data 2, respectively. The remaining data, including cryo-EM, cryo-ET and confocal microscopy images, and newly generated materials are available upon request to the corresponding author. Source data are provided with this paper.

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

## Acknowledgements

We are grateful to Professor Yoshinori Umesaki and Dr. Tatsuichiro Shima from Yakult, Japan, for generously sharing with us rat-SFB under monocolonization conditions. We acknowledge Dr. Dan Littman (NYU) for generously sharing the anti-P3340 (Th17Ag) rabbit polyclonal antibody. We are thankful for the technical support provided by the members of the Center for Animal Resources and Research of the Institut Pasteur (Martine Jacob, Eddie Maranghi, Thierry Angelique, Jérôme Toutain and Marvin Pery) and SFR Necker (Emilie Panafieu, Amaury Gensou and Cristian Dicu). We thank Dr. Sylvie Rabot (INRAE) for providing germ-free rats. We acknowledge the SFR Necker Histology Platform, especially Damien Conrozier, for the technical support provided in the preparation of rodent intestinal cryosections. We thank Dr. François Bontems for the helpful discussions. We thank Gaëlle Chauveau-Le Friec for her technical work. The lab of PS is supported by INSERM, CNRS, Université Paris Cité. A.R.C. was supported by a PhD fellowship from BioSPC, Université Paris Cité. This work was funded through the ERC grant NICHEADAPT (866222), the Bill and Melinda Gates Foundation Grand Challenge grant (OPP1141322), and the Bettencourt Foundation Coups d'élan prize awarded to P.S. The work conducted at University of Zurich was funded through the Swiss National Foundation grant (320030L-227822) awarded to O.M. Open Access enabled by the ERC CoG NICHEADAPT (866222) awarded to P.S. The authors thank the center of microscopy and image analysis (ZMB) at the University of Zurich. The HDX-MS platform was funded by the CACSICE Equipex ANR-11-EQPX-0008. We are also grateful for support for Ultrastructural BioImaging Core Facility equipment from the GIS-IBISA, the DIM One Health, the French government (Agence Nationale de la Recherche) Investissements d'Avenir France BioImaging (FBI, ANR-10-INSB-04-01) and Investissement d'Avenir Laboratoire d'Excellence "Integrative Biology of Emerging Infectious Diseases" (ANR-10-LABX-62-IBEID). We also acknowledge the cryo-ET expertise and assistance of the Institut Pasteur's NanoImaging Core facility, created and supported by a PIA grant (EquipEx CACSICE: ANR-11-EQPX-0008).

## Author contributions

A.R.C. and P.S. were responsible for experimental design. SFB purification, immunofluorescence, immunogold labeling, and confocal microscopy imaging by A.R.C. Cryo-EM/ET sample preparation by A.R.C., T.H.T., G.P.A., and A.D.G., and data acquisition by A.R.C., B.H.W., G.P.A., A.D.G., and J.M.W. Cryo-EM/ET data analysis by A.R.C., B.H.W., A.G., A.S.R., A.D.G., O.M., and P.S. SFB protein expression and purification by A.R.C. and A.L. Germ free animal maintenance by M.B. and A.R.C. VHH production and selection by G.A., P.L., and A.R.C. VHH epitope mapping by HDX-MS by S.B. Funding acquisition, project conception and supervision by P.S. Manuscript writing by A.R.C. and P.S.

## Competing interests

The authors declare no competing interests.
