## [Transparent Peer Review file · Nature Communications]

Segmented filamentous bacteria undergo a structural transition at their adhesive tip during unicellular to filament development

Corresponding Author: Dr Pamela Schnupf

Version 0:

Reviewer comments:

Reviewer #1

(Remarks to the Author)

SFB have major immunostimulatory properties that are dependent on adhesion to host cells and SFB surface structures are likely crucial in host interaction. This study is important because it identifies a putatively novel S-layer and reveals a developmental transition of the SFB tip surface including the transient appearance of structures (disordered hair-like structures) consistent in location and timing with being involved in host cell attachment.

Although there is no molecular characterisation of the identified surface (or intracellular) structures, I fully endorse the publication of this manuscript because the cryo-ET images are first-class despite SFB being an extremely challenging study system (use of monoclonalised mice is necessary for bacterial propagation, a lack of availability of large quantities of intracellular offsprings, lack of genetic tools, etc). I should, however, point out that cryogenic electro-tomography lies outside my field of expertise.

I congratulate the authors for stepping outside the realm of classic (model) organisms, the solid cryogenic-ET and I only have edits to hopefully improve the readability and the clarity of the manuscript:

line 22: specifically localised instead of uniquely.

line 26: Maybe better: d Moreover, although the major Th17 antigen is ubiquitous cell wall protein, this was only expose on the filament tip, underscoring its unique properties.

line 60: please clarify what you. mean by near native conditions.

line 64: in line 54 and 55 the authors write that typical surface layer proteins are absent but then here, in line 64, the authors are sure that what they see is an S layer: maybe better "putative S layer"? Or can the authors be 100% sure that it is an S layer?

line 69: There are so many abbreviations that sometimes the reading is super hard: maybe avoid abbreviations in the titles? And replace IO with intracellular offspring in this line?

line 77: it is probably going to be hard to change this but "back" sounds really strange to me! I had preferred to call proximal tip the tip and distal tip the back. I am curious what the other reviewers will say!

line 113: please clarify "mainly", if possible.

lines 121-123: was it always the case that when vesicles were present at the tip or when the cytoplasmic membrane did not fully extend (by the way, extend rather than extended) into the tip track were visible? Or how often was this the case?

Figure 2 legend: maybe specify that Fig.2 deals with the SFB tip in the title of the legend?

line 131: what do you mean by challenging to track?

line 170: images instead of imaged.

line 177: I guess you can delete "the presence of".

line 286: sorry if I missed it: the small subset which was labelled, which stage was it?

line 306 and elsewhere: I would refrain from abbreviating disordered hair like structures. As said, there are already too many abbreviations.

line 326: please clarify/rephrase "bacteria with similar cell ends".

line 327: why polar tip? I would delete polar.

line 330: I believe the acronym EA1 has not been used before, if so, please explain.
lines 333 - 342: I would refrain from using the abbreviation ordHLS.
line 345: better: decrease of iOS with fully developed (double negation is hard to read).
line 371: better: only a quarter of non-permeabilised IOs showed labelling.
line 374: delete generally.

Reviewer #2

(Remarks to the Author)
Significance:

1. The novelty of this study is the observations of the S-layer, hair-like, and other structural features and their associated developmental phases in the tip and back regions of the SFB from both mouse and rat species. The authors have done a thorough morphological analysis and surface/functional characterization to provide insights into the SFB life cycle. While the presented work offers some critical insights into the unknown structural land of SFB, more analyses and further work will be needed to support the main claim that the morphological changes including the appearance and disappearance of the S-layer and disordered and ordered hair-like structures, are critical for the conserved developmental transition of the SFB tip and involvement in host cell attachment.
2. The observation of IO-specific S-layer and its transition is one of the main points throughout the manuscript. The claim that the replacement of the S-layer surface by disordered hair-like structures, is primarily supported by morphological observations. The S-layer assumes a hexagonal-like, seemingly 6-fold symmetry, which is very interesting. The sample availability and enrichment have been a bottleneck for more detailed structural analysis, as stated by the authors as well. While I fully understand the challenge, it would still be very valuable to further characterize the data by doing some subtomogram averaging and FFT lattice characterization. The existing datasets were collected at pixel sizes of 1.5~2 Å range. I wonder if the authors tried to analyze the pattern via STA? Is it possible to have enough particles using the existing tomograms from one of those pixel sizes for STA? The particle picking for regular arranged or lattice-like structures could be done via the geometric picking solution as shown in Qu et al. 2018 PMID: 30478053, Burt et al. 2020, PMID: 32029744. Dynamo could be useful in this case (Castano-Diez et al. 2017. PMID:27288866). That being said, the high-res STA analysis can take a long time, which largely depends on the sample heterogeneity and particle numbers. Thus, even with a structure in a low-res cryo-ET/STA range, the presence or absence of a true hexagonal-like pattern could greatly support the claim of such a unique arrangement. It will also be particularly informative to see this pattern change in broken tips.
3. The S-layer has repetitive subunits, the spacing of which is around 5 nm while the ordered hair-like structure is approximately 8 nm. The authors also saw the broken tip phenotype that displays discontinuities of the S-layers and absence of disordered HLS. Does the broken S-layer also have the regular 6-fold symmetry or hexagonal arrangement? The region of 1-4 labeling is missing in the Extended Data Fig. 1 e, f, where I assume the close-up/zoomed-in views correspond to the larger field of view tomogram slice view in e and f? More analysis, such as FFT to see the pattern arrangement or STA if possible, even low resolution, can greatly help understand the S layer to dis transition, which is one of the main observations in the manuscript.
4. Various new definitions to describe the SFB ultrastructure are introduced, including tracks and plate-like structures etc. While the study is focusing on the tip region, it is unclear how these structures are related to the understanding of the tip development stages. Do they change, e.g. more or less, longer or shorter, in the five or three main stages? While the full length is hard to determine, is there a thickness difference between tracks, filaments, and the plate-like structures? It is interesting to see the chemosensory array, a well-studied structural layer in several bacteria species, in the mSFB and rSFB. Could the authors see the chemosensory array lattice array oriented perpendicular to the projecting beam in their tomograms? Does the chemosensory array undergo morphological changes, such as an increased or decreased appearance, when the S-layer is being replaced by the HLL? It will be valuable to have a relatively comprehensive understanding of the morphological changes upon the developmental transitions by incorporating these structures that are not strictly located in the tip region. Model illustrations of the SFB with these structural features that correspond to the main developmental stages e.g. stage 1, 3, 4/5 should do the job.
5. The authors claim that the observed eVs are SFB-derived. While the eVs do appear in the tip region as shown in Fig. 1, 2, the sample went through purification prior to being put on the grid. It is likely that some eVs from damaged cells or debris could get attached to the SFB during the purification process, which is very common. To prove its SFB origin or support this claim, labeling or live-cell imaging, or observation of these eVs from the native cell cryo-ET where the SFB naturally attached to the surface of the co-cultured host cells, is needed.
6. It is common to see nonspecific binding in gold immunolabeling. In some cases, old gold fiducials used for alignment can preferentially bind to the surface proteins, leading to "labeling-like" effect. It is important to provide a control image to show the gold immunolabeling specificity.
7. The authors hypothesize that the presence of S-layer and HLS are involved in cell wall integrity. While the eV is seen near the broken phenotype, as stated above, it is unclear where the eV comes from. To prove this hypothesis, cell wall integrity needs to be checked for the various developmental stages and see if there is a difference between stage 1 and stage 3 and stage 5. While Th17Ag, a cell wall protein, has been explored in the paper, did the author check other landmarks that prove the cell integrity of the SFB after the purification?

8. Cryo-ET data have been collected at multiple pixel sizes with various defocus and processing. While the method section is comprehensive, it will be good to provide a table that states which pixel size data sets are used for what data analysis e.g. S-layer structure characterizations in Fig. xxx, etc.

Minor:

In Fig. 3 d iii, it is challenging to see the presence of both ordered HLS and disHLS in the current tomogram slice view. Fig 3. B is a 2D projection instead of a tomogram slice.

In Fig 3.f, it is hard to pinpoint each stage and their percentages due to the similar shades of grey. It might be better to use other more distinct colors to represent in the pie chart.

Line 320, the HLL may itself constitute an S-layer since it is not repetitive but also appears to prevent the access of the protein. There is no direct evidence in the manuscript to support this claim.

Line 735, please state the original pixel size and binned pixel size after the bin4 has been performed.

Reviewer #3

(Remarks to the Author)

The manuscript by Gruz and colleagues presents a high-resolution study of segmented filamentous bacteria - an important gut commensal microbe known for its potent immunostimulatory effects using state-of-the-art cryo-electron microscopy and cryo-electron tomography (cryo-EM/ET). The authors capture high-resolution images of SFB at different developmental stages, from short unicellular intracellular offsprings, IO, to long filamentous cells. The authors discover a progression of surface structures, an S-layer transforming into an entirely new, "hair-like" layer, identify novel "hair-like structures" potentially involved in host-cell attachment, and demonstrate differential surface accessibility of the major Th17 antigen at the tip of filamentous forms.

The cryo-EM/ET data is of high quality, the tomograms look excellent, and are analyzed rigorously with respect to the developmental stages. The manuscript is well-written and clear. It provides novel insights into the development and provides food for thought for future research.

The major drawback of the manuscript is the unknown molecular identity of the hair-like structures. This is a very important limitation of the study, and as it is, the manuscript is very descriptive. It may be possible to identify the molecules by pulldowns of Th17, proteomics, bioinformatics, subvolume averaging, or their combination.

Furthermore, it would be interesting to describe the mutual arrangement of the hair-like molecules throughout the developmental process, at least where they are ordered. This could be another important step for the identification of the molecules. Even though the cryo-ET dataset is of moderate size, there should be enough particles for subvolume averaging, and even at moderate resolution, it can provide useful insights.

Version 1:

Reviewer comments:

Reviewer #2

(Remarks to the Author)

I appreciate the authors took time to address the questions brought up, particularly analyzing the OI tip with STA. I commend the authors for the time and work that went into it. I have the following questions and hope to further improve the manuscript.

1) Line 108: Here, the symmetry observation is being described and discussed. I agree that the claim would be "potential" 6-fold symmetry. If the author would like to mention the performance of STA, it would be necessary/important to include the STA results as part of the Supplemental/Extended Fig. I think in this case, the lack of a clear definition of 6-fold symmetry organization could result from the missing wedge artifact despite the missing wedge compensation in PEET or Dynamo. I wonder if the author has tried to limit the angular search range and translational search range as well after all subtomogram were aligned perpendicular to the beam direction. I have seen lattice structures suffer from the missing wedge effect (despite correction) during the alignment. As a result, the wrong alignment led to misleading 6-fold symmetry. Two approaches could be helpful. One would be to align the subtomograms towards a fixed direction, e.g. Revision Fig. 1e, where the membrane is centered and perpendicular to the Y. Then apply a limited search angle/translation so that the aligned/guided positions are not lost during the subsequent alignment in PEET or Dynamo. The other one would be to provide some guidance for the angular search or predetermined orientation in Relion 3D refinement.

The subsequent analysis could require more time. I would suggest removing the sentence "nevertheless, xxxx (data not shown)".

- 2) Line 108, "a more relaxed S-layer". Is there any evidence or measurement, like curvature measurements, to prove the claim?
- 3) Fig. 6, the HLS is too small to see in z (white arrowhead), the HLS will need to be zoomed in/enlarged. In f, the merged and Alexa-658 are mislabeled/backwards. h, the residue font size is too small to read.
- 4) Could the authors comment on why Th17Ag shows the labeling is more predominant in the late-stage IOs compared to the early stage, where the S-layer is present? Also, is Th17Ag present in both ordered and disordered HLS?
- 5) Line 439, it might be good to specify the pixel size for the lower mag because too small a mag (big pixel size) could lead to loss of important structural details that are required for further analysis. A bigger pixel size (4~5 Å/pixel) or montage cryoET (Reference) could help overcome the field of view limitation.
- 6) Line 448, I would suggest removing "the extraction and purification of the S-layer... may overcome these limitations". Do the authors mean purifying the S-layer stems followed by reconstitution, or just purifying and putting them under the microscope? Purification and extraction may break the samples. To further study the organization of the S-layer stems below, a native, in-situ approach, as done here still remains the best. I would suggest removing this comment.
- 7) Line 444, no evidence for "twisted lattice" was provided here. I would think the tip curvature and the sample size would be the limiting factors for higher resolution STA analysis.
- 8) In Fig5 a-d, most of the I or O-HLL don't seem to go together. e.g. in all of them, I barely see a uniform coating of I-HLL. Is it because they appear on different tomographic slices? Could summing a few slices in Slicer/IMOD help to display them both at the same time? They seem to share a very similar spacing. The schematic view (c) shows O and I-HLL appear at the same time with almost identical spacing. If the organization of I/O-HLL in the slice view is not due to the different Z planes, then c needs to be modified.

Reviewer #3

(Remarks to the Author)

I thank the authors for their thorough and detailed response to the initial reviews. The manuscript has been improved by the revisions, and the additional analyses and clarifications strengthened the work. The authors have addressed my previous comments satisfactorily. They have acknowledged the limitations regarding the molecular identity of the newly described hair-like structures and have convincingly argued in their rebuttal that the identification of these components is fraught with significant technical challenges inherent to the SFB system. I appreciate the addition of new supplementary figures showing the top-down views of the disordered and ordered hair-like structures, which support their assertion that subtomogram averaging is not currently feasible due to the flexibility and lack of a clear, repetitive lattice. Despite these improvements, the central and significant gap of the study remains the unknown molecular identity of the S-layer, the disordered hair-like structures, and the hair-like layer. This lack of molecular information necessarily renders the manuscript highly descriptive in nature. While the morphological characterization is of very high quality and provides novel insights into the SFB life cycle, the absence of molecular data is a major limitation. The manuscript does not provide the molecular link, but it is correct and provides a foundation for future molecular studies.

Point-by-point response to reviewers' comments for the manuscript NCOMMS-25-16988-T

Text in black – reviewer comments

Text in orange – author response

Text in green – changes in the manuscript

We thank the reviewers for their careful review of our work and for the constructive feedback, which has substantially improved this manuscript. Our point-by-point response is below.

REVIEWER COMMENTS

Reviewer #1 (Remarks to the Author):

SFB have major immunostimulatory properties that are dependent on adhesion to host cells and SFB surface structures are likely crucial in host interaction. This study is important because it identifies a putatively novel S-layer and reveals a developmental transition of the SFB tip surface including the transient appearance of structures (disordered hair-like structures) consistent in location and timing with being involved in host cell attachment.

Although there is no molecular characterisation of the identified surface (or intracellular) structures, I fully endorse the publication of this manuscript because the cryo-ET images are first-class despite SFB being an extremely challenging study system (use of monoclonised mice is necessary for bacterial propagation, a lack of availability of large quantities of intracellular offsprings, lack of genetic tools, etc). I should, however, point out that cryogenic electro-tomography lies outside my field of expertise.

I congratulate the authors for stepping outside the realm of classic (model) organisms, the solid cryogenic-ET and I only have edits to hopefully improve the readability and the clarity of the manuscript:

We thank the reviewer for the encouraging comments and for their acknowledgement of the challenges of working with Segmented Filamentous Bacteria (SFB).

line 22: specifically localised instead of uniquely.

Has been changed accordingly.

Line 23: IOs were surrounded by a repetitive surface (S)-layer that became replaced by disordered hair-like structures specifically at the IO tip.

line 26: Maybe better: d Moreover, although the major Th17 antigen is ubiquitous cell wall protein, this was only expose on the filament tip, underscoring its unique properties.

Has been changed to:

Line 28: Moreover, the major Th17 antigen, which is a ubiquitous cell wall protein, was only exposed on the filament tip, underscoring the unique properties of the tip structure.

line 60: please clarify what you. mean by near native conditions.

Has been changed to:

Line 63: To obtain a more detailed view of the SFB surface, we imaged purified IOs and filaments from mouse- and rat-SFB in a hydrated state using a combination of cryogenic electron microscopy (cryo-EM) and tomography (cryo-ET).

line 64: in line 54 and 55 the authors write that typical surface layer proteins are absent but then here, in line 64, the authors are sure that what they see is an S layer: maybe better "putative S layer"? Or can the authors be 100% sure that it is an S layer?

While typical surface layer proteins and cell wall anchoring domains are not encoded in the SFB genome, the absence of typical S-layer homology (SLH) and other typical S-layer protein (SLP) cell-wall anchoring domains is not unique to SFB and has been reported for SLP from other monoderm bacteria such as *Lactobacillus acidophilus*¹ and *Lactobacillus brevis*². Instead, a domain of unknown function (DUF4214) is present in 33 proteins of mouse-SFB-NL including the cell-wall protein Th17Ag and other predicted surface proteins and it has been hypothesized to play a role in cell wall anchoring. The introduction has been modified as follows to clarify this point:

Line 55: The typical cell wall binding motifs and domains present in most Clostridia are largely absent^{19–22}. This includes surface (S-) layer homology (SLH) domains characteristic of some, but not all, surface layer (S-layer) proteins from monoderm bacteria²³. Instead, likely surface-anchoring domains in SFB include a domain of unknown function (DUF4214) identified in multiple predicted surface proteins^{19–22}.

Regarding the reviewer's suggestion to use the nomenclature "putative S-layer" instead of "S-layer", while it is true that the molecular identity of its building blocks remains unknown, the surface layer we identified in SFB IOs resembles other bacterial S-layers and it possesses all the remaining characteristics of bacterial S-layers. It is a repetitive outermost layer exterior to the cell wall, possess the typical center-to-center S-layer spacings, has the electron density compatible with a proteinaceous structure and is morphologically similar to other bacterial S-layers. Thus, we would prefer to maintain the S-layer nomenclature but clarify why we are referring to this layer as an S-layer. The manuscript has been modified as follows:

Line 113: Together, these data reveal the presence of a repetitive outermost layer with the typical S-layer center-to-center spacings that surrounds IOs and is conserved in SFB from various hosts.

line 69: There are so many abbreviations that sometimes the reading is super hard: maybe avoid abbreviations in the titles? And replace IO with intracellular offspring in this line?

We thank the reviewer for this suggestion to improve the manuscript readability. We removed abbreviations from the titles (with the exception of SFB) and have reduced the abbreviations for hair-like structures (HLS).

Line 71: Unicellular SFB are surrounded by an S-layer

Line 306: The major Th17 antigen is accessible at the tip of SFB filaments

line 77: it is probably going to be hard to change this but "back" sounds really strange to me! I had preferred to call proximal tip the tip and distal tip the back. I am curious what the other reviewers will say!

We acknowledge the reviewer's comment on the terminology used for the two SFB poles. Given that "tip" has already been used in the literature to designate the structure of the teardrop-shaped SFB that attach to the intestinal epithelium³, we propose to alter the designation of "back" to "distal end". This has been changed throughout the text and manuscript figures.

line 113: please clarify "mainly", if possible.

The characterization of most intracellular features identified required the use of tomograms and our Cryo-ET dataset mainly targeted the SFB tip and not the distal end, due to the sample thickness limits inherent of cryo-ET (i.e. while we have 66 tomograms that allow us to visualize the SFB tip, we only have 3 tomograms for the distal end of mouse-SFB grown in mice in our dataset). Thus, we focused our analysis on the intracellular features identified at the SFB tip and proximal cell body region.

Nevertheless, we have also screened both tomograms and projection images of the distal end of mouse-SFB for the presence of tracks. The same screening was performed for both plate-like structures and filaments using tomograms and the data was included in new columns in **Supplementary Table 1**. Examples of the tracks and plate-like structures identified at the SFB distal end are shown in **Supplementary Fig. 3a/b**.

The manuscript has been changed to:

Line 118: These structures were characterized at the SFB tip and proximal cell body (Fig. 2a-f, Supplementary Fig. 2a-e) although some of these features were also found at the SFB distal end (Supplementary Fig. 3a/b, Supplementary Table 1).

Line 135: Given that our cryo-ET dataset mainly focused on the tip structure, these intracellular features were only characterized in this region even though they could also be found at the distal end (Supplementary Fig. 3a, Supplementary Table 1).

Line 173: Conversely, plate-like structures were less common (Supplementary Table 1) and were found in the centre of the IOs cell body, but also at the SFB distal end (Supplementary Fig. 3b), either isolated or in groups of 2-3 plate-like structures, showing a maximum of 200 nm in length when their full length could be measured (Supplementary Fig. 2a(ii),b(iv)).

lines 121-123: was it always the case that when vesicles were present at the tip or when the cytoplasmic membrane did not fully extend (by the way, extend rather than extended) into the tip track were visible? Or how often was this the case?

To answer to the reviewer's question, we inspected again our dataset and performed a more detailed analysis of the tracks identified at the SFB tip. We now show a significant association between the presence of tracks and both membrane retraction and intracellular vesicles. This analysis is shown in **Supplementary Fig. 5d** and the corresponding information was added to new columns in **Supplementary Table 1**. We have also included examples of the potentially different types of tracks that we see at the filament tip (**Supplementary Fig. 5a-c**) and show that the presence of these features does not seem to be restricted to the SFB tip (**Supplementary Fig. 3a, Supplementary Table 1**).

The text altered in the manuscript is below.

Line 131: We also identified electron-dense lines parallel to the cell wall at the SFB tip, here designated as tracks (Fig. 2a, Supplementary Fig. 2a/b(v),c(vi) and d(vii)). Tracks were commonly identified at the mouse-SFB tip but rarely at the rat-SFB tip (Supplementary Table 1, Supplementary Fig. 2d(vii)), possibly due to the shorter tip length of rat-SFB compared to that of mouse-SFB (Supplementary Fig. 4). Given that our cryo-ET dataset mainly focused on the tip structure, these intracellular features were only characterized in this region even though they could also be found at the distal end (Supplementary Fig. 3a, Supplementary Table 1).

Tracks were designated as 'short' (Fig. 2a, Supplementary Fig. 2a/b(v),c(vi) and d(vii)) when their distance could be measured and as 'long' when we could not fully visualize them in the field of view imaged (Supplementary Fig. 5a). While these features were commonly present on both sides of the tip, we also observed them only on one side (Supplementary Fig. 5b). Short and long tracks could be found in the same bacterium (Supplementary Fig. 5c). Tracks were only identified in SFB that showed evidence of cytoplasmic membrane retraction (Supplementary Fig. 5d(i)) and were present in the majority of SFB also harboring intracellular vesicles (61%, $n = 18$) (Supplementary Fig. 5d(ii)). The length and thickness of short tracks was on average 126 ± 61 nm ($n = 17$) and 55 ± 30 nm ($n = 11$), respectively (Supplementary Fig. 5e(i/ii)).

The length and thickness could not be estimated for long tracks due to the limited region imaged (Supplementary Fig. 5e(i/ii)). Nevertheless, given their similar location and width (Supplementary Fig. 5e(iii)), we hypothesize that both short and long tracks may play a role in maintaining the SFB shape or in anchoring the cytoplasmic membrane within the elongated tip and beyond.

Figure 2 legend: maybe specify that Fig.2 deals with the SFB tip in the title of the legend?

The Fig. 2 and Supplementary Fig. 2 legend titles have been changed to:

Fig. 2. Description of intracellular and surface SFB features identified at the SFB tip and proximal region of the cell body.

Supplementary Fig. 2. Additional examples of the intracellular and extracellular features identified at or near the SFB tip and proximal region of the cell body of SFB; Fig. 2.

line 131: what do you mean by challenging to track?

We clarified this sentence in the manuscript. Has been changed to:

Line 153: One type consisted of multiple filaments clustered together with a thickness of approximately only 26 ± 12 nm ($n = 23$), which made it challenging to follow them in every tomographic slice (Fig. 2c,e/f(iv)), while the other filaments resembled more plate-like structures with a thickness of approximately 63 ± 20 nm ($n = 6$) (Fig. 2d,e/f(v)).

line 170: images instead of imaged.

Has been changed to:

Line 195: SFB were grouped into stages based on the surface structures identified at the tip.

line 177: I guess you can delete "the presence of".

Has been deleted:

Line 203: In 20% of the IOs below $3.2 \mu\text{m}$, discontinuities were observed in the bacterial S-layer without disordered hair-like structures, leading to the 'broken tip' phenotype and preventing us from identifying the tip stage (Undefined: UD, Fig. 3f, Supplementary Fig. 8a).

line 286: sorry if I missed it: the small subset which was labelled, which stage was it?

We thank the reviewer for raising this important point, it was not previously mentioned in the manuscript. A new Supplementary Table (**Supplementary Table 3**) was prepared, which includes the information on the tip stage of the SFB that were or not considered labelled after immunogold labelling. Additionally, we have also attempted to determine if a particular IO stage was more enriched in the small subset of IOs labelled. The results of this analysis have been added to the manuscript and are included below:

Line 323: While labelling was detected across the different IO stages (Supplementary Table 3), late-stage IOs (stages 4 and 5) were present at a significantly higher proportion in the labelled population (23%, $n = 13$) compared to the unlabeled population (0%, $n = 26$) (Supplementary Fig. 15a(i-iii)). The same trend was identified for stage 3 IOs among early SFB stages (stages 1 to 3), although this difference was not significant (60% of labelled IOs $n = 5$ vs 15% of unlabeled IOs $n = 13$) (Supplementary Fig. 15a(iv)).

Line 349: For IOs, as only a small subset, enriched in late-stage IOs, was labelled, we hypothesize that the increased accessibility may be related to the transition of the S-layer to the HLL rather than a general greater permeability of the S-layer.

line 306 and elsewhere: I would refrain from abbreviating disordered hair like structures. As said, there are already too many abbreviations.

We removed abbreviations of disHLS and ordHLS throughout the text. We have only kept these abbreviations in the figures and supplementary movies due to space restraints. The abbreviations for HLS were kept in the discussion for readability although we redefine these terms at the beginning of this section.

line 326: please clarify/rephrase "bacteria with similar cell ends".

Has been changed to:

Line 389: S-layers can play a role in host adhesion in bacteria with symmetric poles³¹.

line 327: why polar tip? I would delete polar.

Has been deleted:

Line 389: However, since in SFB it is the tip that mediates epithelial cell attachment, it is unlikely that the S-layer or the HLL are involved in adhesion but rather may function in bacterial protection and/or maintenance of cell wall integrity.

line 330: I believe the acronym EA1 has not been used before, if so, please explain.

Has been clarified:

Line 384: The potential replacement of one surface layer with another is reminiscent of the transition previously described for *Bacillus anthracis*³⁹. In *B. anthracis*, an exponential phase-associated S-layer composed of the surface array protein (Sap) is replaced by a stationary phase-associated S-layer composed of the extracellular antigen 1 (EA1).

lines 333 - 342: I would refrain from using the abbreviation ordHLS.

We removed abbreviations of disHLS and ordHLS throughout the text. We have only kept these abbreviations in the figures and supplementary movies due to space limitations.

line 345: better: decrease of iOS with fully developed (double negation is hard to read).

We agree with the reviewer's comment regarding the readability of this sentence. Even though there is a decrease in the proportion of IOs with fully developed hair-like structures (30% of Stage 3 mouse-SFB grown in mice in comparison to 14% of Stage 3 mouse-SFB grown in rats, Fig. 7m), this increase is not statistically significant. However, the increase of Stage 2 mouse-SFB from 32% in mice to 61% in rats is statistically significant. To avoid commenting on non-significant differences we propose to keep the sentence below in the manuscript:

Line 416: Furthermore, their emergence prior to bacterial elongation and the increase in the proportion of IOs without fully developed disordered HLS in a heterologous host, where attachment is not supported, further support this hypothesis.

line 371: better: only a quarter of non-permeabilised IOs showed labelling.

Has been changed to:

Line 397: However, only approximately a quarter of non-permeabilized IOs showed labelling and labelling was not restricted to a specific region of the bacterial surface.

line 374: delete generally.

Has been deleted:

Line 399: These results suggest that the HLL in SFB filaments prevents access to this potential cell wall-anchored protein while the S-layer appears to have a similar function for IOs.

Reviewer #2 (Remarks to the Author):

Significance:

1. The novelty of this study is the observations of the S-layer, hair-like, and other structural features and their associated developmental phases in the tip and back regions of the SFB from both mouse and rat species. The authors have done a thorough morphological analysis and surface/functional characterization to provide insights into the SFB life cycle. While the presented work offers some critical insights into the unknown structural land of SFB, more analyses and further work will be needed to support the main claim that the morphological changes including the appearance and disappearance of the S-layer and disordered and ordered hair-like structures, are critical for the conserved developmental transition of the SFB tip and involvement in host cell attachment.

We thank the reviewer for acknowledging the novelty of our morphological characterization of the SFB surface. While we describe the presence of morphologically-distinct features at the surface of SFB from distinct developmental stages, we had no intention to claim these changes are critical for the SFB developmental transition and host cell attachment.

To address this, we would have to identify the molecular composition of the identified structures, to perform genomic deletions of the corresponding genes and to assess the ability of SFB mutants to undergo the characteristic unicellular to filament transition and attach to host cells. Given the lack of available tools to perform these experiments, for the moment we can only provide a descriptive analysis and claim that the S-layer we identified was only observed in unicellular SFB while the HLL was characteristic of longer SFB and, thus, associated with later developmental stages.

Similarly, we cannot currently demonstrate the potential role of disordered hair-like structures in SFB attachment. We hypothesize that these structures may be involved in this close interaction due to the following observations:

- (1) disordered hair-like structures are localized specifically at the SFB adhesive tip;
- (2) disordered hair-like structures are found in unicellular bacteria, appearing prior to bacterial elongation and, thus, prior to bacterial outgrowth;
- (3) the proportion of unicellular bacteria without fully developed hair-like structures (Stage 2 SFB) is statistically higher when mouse-SFB is grown in rats (a setting in which attachment does not occur) in comparison to when these bacteria are grown in mice (a setting in which attachment occurs), suggesting a potential delay in the development of these features in the absence of attachment.

To clarify these points in the manuscript we performed the following modifications:

Line 65: We identified a number of intracellular structures but mainly focused on the SFB surface at the bacterial tip for which we describe a morphological transition during the outgrowth of unicellular IOs into filaments.

Line 67: This transition includes the replacement of an IO-specific S-layer by a repetitive hair-like layer, the appearance of tip structures, and a unique accessibility of the major Th17 antigen, a ubiquitous cell wall protein, at the filament tip.

Line 247: Given the location of the disordered hair-like structures and their presence in unicellular SFB, we hypothesize that these structures may play a role in SFB attachment to the host.

2. The observation of IO-specific S-layer and its transition is one of the main points throughout the manuscript. The claim that the replacement of the S-layer surface by disordered hair-like structures, is primarily supported by morphological observations. The S-layer assumes a hexagonal-like, seemingly 6-fold symmetry, which is very interesting. The sample availability and enrichment have been a bottleneck for more detailed structural analysis, as stated by the authors as well. While I fully understand the challenge, it would still be very valuable to further characterize the data by doing some subtomogram averaging and FFT lattice characterization. The existing datasets were collected at pixel sizes of 1.5–2 Å range. I wonder if the authors tried to analyze the pattern via STA? Is it possible to have enough particles using the existing tomograms from one of those pixel sizes for STA? The particle picking for regular arranged or lattice-like structures could be done via the geometric picking solution as shown in Qu et al. 2018 PMID: 30478053, Burt et al. 2020, PMID: 32029744. Dynamo could be useful in this case (Castano-Diez et al. 2017. PMID:27288866). That being said, the high-res STA analysis can take a long time, which largely depends on the sample heterogeneity and particle numbers. Thus, even with a structure in a low-res cryo-ET/STA range, the presence or absence of a true hexagonal-like pattern could greatly support the claim of such a unique arrangement. It will also be particularly informative to see this pattern change in broken tips.

We have conducted subtomogram averaging (STA) of the surface structures observed in both mouse- and rat-SFB, as suggested by the reviewer. To this end, we followed an approach similar to the one used by Briggs and colleagues for viruses⁴.

Initially, we tested the strategy on a subset of 5 tomograms of the mouse-SFB tip. Tilt series were acquired at 2.21 Å/pix with a total dose of 140 e⁻Å⁻² and were reconstructed as described in the Methods section of the manuscript. CTF estimation and correction by phase-flipping followed by dose filtration were performed following the standard IMOD 4.11 workflow.

The tomograms were reconstructed and imported into a Dynamo catalogue⁵. Then, within Dynamo, the surface of the mouse-SFB tip was traced using the Membrane model approach, with the final cropping mesh sampled to achieve an average picking distance around 7 nm, slightly below the spacing of the stems observed in the tomograms. Then, the coordinates and the associated normal angles were extracted (**Revision Fig. 1a/b**, 8480 particles total).

At the coordinates picked in Dynamo, 3D-CTF-corrected and dose-filtered subtomograms with a box size of 180 px (**Revision Fig. 1c**, 403 nm³, approximately 5 repeats) were reconstructed using IMOD. Subtomogram contrast was inverted, and the intensity was normalized prior moving to the RELION software package. For each tomogram, a wedge mask was reconstructed using the approach outlined by Bharat & Scheres⁶, containing the tilt angle information. Additionally, we converted angular information from the surface picking in Dynamo to RELION-compatible Euler angles using dynamo2m⁷.

We continued processing our data in RELION. In parallel to working with 3D subtomograms, we also calculated the projections of the subtomograms along the Z axis. 2D classification on these projection images recovered the expected pattern of globular S-layer densities (stems) centered below a “zig-zagged” layer that corresponded to the crescent-like elements seen in the side views of tomograms (**Revision Fig. 1d**).

Initial attempts of aligning the subtomograms against an unsymmetrized template resulted in a low-resolution 3D map which matched the overall tomographic appearance of the surface layer, revealing the globular densities centered below a “zig-zagged” layer. The other cross-section, however, revealed the strong curvature of the tip (**Revision Fig. 1e**). Inspection of the angular distribution of particles confirmed that all subtomograms were aligned perpendicular to the beam direction, resulting in a highly anisotropic structure and resolution, mostly lacking information along the Z axis, with a missing cone of information. To overcome this preferred orientation, we tried a symmetry scan, in which we applied a variety of C-group symmetries (C1 – C9, C360) to an unsymmetrized template. The symmetrized maps were subsequently used as a template for a Refine3D job in the RELION software without imposing additional symmetry. The densities of the output maps consistently showed stems arranged in offset lines, but did not yield any conclusive indication of the underlying symmetry (**Revision Fig. 1f**).

A potential hexagonal pattern was observed in the top view of the tomograms shown in the **Supplementary Fig. 1 e/f** and a clear hexagonal pattern was reported for a previously published structure of the S-layer of the bacterium *Caulobacter crescentus*⁸, which possesses a stalk resembling the SFB tip. Thus, we continued with the particles aligned to a six-fold symmetrized reference. 3D classification initially revealed a subset (4273 particles, 50%) of particles resolving into a hexagonal arrangement of the stems (**Revision Fig. 1g**) with an appearance similar to that of the *C. crescentus* S-layer. However, further gold-standard refinement of this subset failed to converge without symmetry imposition, and despite many attempts, no such subset could be identified without providing a C6-symmetrized template. We thus concluded that the hexagonal structure we obtained was suffering from heavy template bias.

As we could not overcome the preferred orientation caused by the strong curvature of the mouse-SFB tip without symmetrization, and we could not identify any symmetry *de novo*, we were not able to pursue STA of the mouse-SFB S-layer. A test of the rat-SFB S-layer revealed the same difficulties.

In conclusion, we could not further elucidate the surface arrangement of either mouse- or rat-SFB by STA, in spite of a suitable dataset and a considerable amount of time invested. In our opinion, there are four major hurdles limiting the feasibility of STA on this interesting target:

1. We are lacking a large enough dataset with a consistent curvature and quality, due to the high heterogeneity of the SFB tip regions and current inability to culture these poorly studied commensals in the absence of host cells
2. In order to determine the symmetry *de novo*, we would require either:
 - a. A planar image of the S-layer to identify the symmetry directly using the power spectrum. This was not possible with the highly curved tip structure
 - b. A clear high-contrast pattern on the cell surface, as seen for example on *C. crescentus*⁹. The repeat distance within the S-layer of the SFB however, is much smaller than the repeat of S-layers that previously resolved by STA (20 - 22 nm^{8,9}). The smaller repeat distance would mean that a high resolution would be needed to determine the overall symmetry
3. The curvature of the surface differs along the tip, and the symmetry of the S-layer may be distorted in the highly curved regions, to accommodate these differences. Thus, the local symmetry may be obscured by the distortions in the global symmetry.
4. Overall, the changes in the curvature at the tip region make the analysis very challenging. This is a major difference from the analysis by Briggs and colleagues for viruses⁴.

Revision Fig. 1. Strategy used to perform subtomogram averaging on the SFB S-layer. **a**, Tomographic slice from a representative mouse-SFB IO tomogram (EMD-52668) used for tip surface tracing using the Membrane model approach (Dynamo). **b**, Corresponding coordinates and associated normal angles extracted from the Membrane model approach (Dynamo). **c**, Schematic representation of the box size used for subtomogram reconstruction using IMOD. **d**, Projection image of 3D subtomograms along the z axis. **e**, Orthoslices of the 3D reconstruction that resulted from subtomogram alignment against an unsymmetrized template. **f**, Input and output maps from a symmetry scan in which C-group symmetries were applied to an unsymmetrized template. **g**, Side (top panel) and top view (bottom panel) of the 3D reconstruction obtained when particles were aligned to a six-fold symmetrized reference.

Since we cannot provide a definitive identification of the S-layer symmetry and are hesitating saying that it is a 6-fold. To make this clear, we added the top view of a mouse-SFB tip showing evidence for a potential twisted lattice and for the apparent organization of the S-layer subunits in rows to a main figure (**Fig. 1I**). We have also modified the text in the manuscript to convey the limitations of our current description. The text has been changed to:

Line 107: A potential 6-fold symmetry could be seen in the rat IO with a 'broken tip' phenotype, possibly due to a more relaxed S-layer (Supplementary Fig. 1f). Nevertheless, subtomogram averaging of the S-layer globular densities did not allow de novo identification of the lattice symmetry (data not shown). Identification of the S-layer symmetry will require a more detailed analysis beyond what whole SFB cryo-ET can provide.

Line 441: The characterization of the SFB S-layer symmetry was also not fully conclusive. In-depth in situ symmetry identification could be performed for the S-layer covering the long stalk of *C. crescentus*²⁸, which is morphologically similar to the SFB tip. However, the strong SFB tip curvature, twisted lattice, and smaller center-to-center spacing of the SFB S-layer (~8 nm) in comparison to the S-layer from *C. crescentus* (20-22 nm^{28,45}) may have contributed to the difficulties encountered in the identification of the lattice symmetry for SFB.

3. The S-layer has repetitive subunits, the spacing of which is around 5 nm while the ordered hair-like structure is approximately 8 nm. The authors also saw the broken tip phenotype that displays discontinuities of the S-layers and absence of disordered HLS. Does the broken S-layer also have the regular 6-fold symmetry or hexagonal arrangement?

The region of 1-4 labeling is missing in the Supplementary Fig. 1e, f, where I assume the close-up/zoomed-in views correspond to the larger field of view tomogram slice view in e and f? More analysis, such as FFT to see the pattern arrangement or STA if possible, even low resolution, can greatly help understand the S layer to dis transition, which is one of the main observations in the manuscript.

A possible hexagonal arrangement can be seen in the top view of an S-layer-containing region of a broken rat-SFB tip. This is shown in the tomographic slice and corresponding zoomed-in views included in **Supplementary Fig. 1f**. However, this apparent pattern was not clear for other broken rat-SFB tips (15 tomograms) or mouse-SFB tips (8 tomograms) from our dataset. Performing subtomogram averaging in the S-layer containing regions of the SFB that possess a broken tip will be challenging given that these regions are very limited and, in most cases, included in the bacterial cell body, thus suffering from increased thickness. The thickness would compromise the subtomogram alignment and is likely to yield inconclusive results given that our attempts to identify the symmetry of the intact S-layer have not been successful (**Revision Fig. 1**).

Moreover, to gain further insights into the transition from the S-layer to disordered hair-like structures, it is interesting to analyze the arrangement of the broken tips, as suggested by the reviewer. We carefully screened our dataset and prepared tomographic slices from the top view of a representative rat-SFB possessing a broken tip. This is included in the **Revision Fig. 2**. The regions of the broken tips that no longer possesses an S-layer do not appear ordered. Thus, performing subtomogram averaging in these regions would require a larger dataset to ensure that we would have enough subtomograms to compensate

for the additional flexibility of this layer. Nevertheless, the high curvature of the tip end would pose the same challenges faced when attempting to perform subtomogram averaging for the SFB S-layer.

Revision Fig. 2. Top view of a rat-SFB IO showing a 'broken tip' region where the S-layer is absent. Representative tomographic slice of a rat-SFB (rSFB) IO tip tomogram (EMD-52689). An S-layer discontinuity at the SFB tip resulting in a 'broken tip' phenotype is shown by a white arrow head. A close-up (z) of the region delimited by a white dashed line was included next to panel a. **Scale bars:** a: 50 nm; a(z): 20 nm.

Regarding the labelling of the regions z 1-4 in **Supplementary Fig. 1e/f**, we have clarified the figure legend and removed one of the zoomed panels to increase clarity. The new figure legend is below:

Supplementary Fig. 1. Top view of the S-layer arrangement at the SFB tip. a-c, Top view of the organization of S-layer subunits in rows seen in tomograms from the tip of (a,b) mouse-SFB (EMD-52685, EMD-52687) and (c,d) rat-SFB IOs (EMD-52688, EMD-52684). Close-ups (z) of the regions delimited by a white dashed line were included below each panel. Red arrow heads show S-layer subunits arranged in rows. e,f, Top view of tomograms which include the tip of (e) mouse-SFB (EMD-52655) and (f) rat-SFB IOs (EMD-52689), showing a potential six-fold symmetry of the S-layer subunits. Close-ups (z) of the regions delimited by a white dashed line were included below or next to each panel. z2 panels were duplicated to highlight example regions of a potential 6-fold symmetry (red dashed hexagons). S-layer discontinuities at the SFB tip resulting in a 'broken tip' phenotype are shown by white arrow heads. mSFB: mouse-SFB, rSFB: rat-SFB. Scale bars: a-f: 50 nm; a-f(z): 20 nm.

4. Various new definitions to describe the SFB ultrastructure are introduced, including tracks and plate-like structures etc. While the study is focusing on the tip region, it is unclear how these structures are related to the understanding of the tip development stages. Do they change, e.g. more or less, longer or shorter, in the five or three main stages? While the full length is hard to determine, is there a thickness difference between tracks, filaments, and the plate-like structures?

We thank the reviewer for highlighting this important point. We have carefully re-checked our dataset and performed multiple analyses to explore potential changes in both intracellular and extracellular features identified according to the SFB stage.

Since tracks could be seen in both tomograms and projection images and the field of view of our tomograms was sufficient to assess their presence, we were able to perform a detailed analysis which is now included in **Supplementary Fig. 5**. This includes an analysis of the presence of tracks in the different SFB tip stages and the comparison between the width of the potentially different types of tracks observed. The corresponding text is below.

Line 131: We also identified electron-dense lines parallel to the cell wall at the SFB tip, here designated as tracks (Fig. 2a, Supplementary Fig. 2a/b(v),c(vi) and d(vii)). Tracks were commonly identified at the mouse-SFB tip but rarely at the rat-SFB tip (Supplementary Table 1, Supplementary Fig. 2d(vii)), possibly due to the shorter tip length of rat-SFB compared to that of mouse-SFB (Supplementary Fig. 4). Given that our cryo-ET dataset mainly focused on the tip structure, these intracellular features were only characterized in this region even though they could also be found at the distal end (Supplementary Fig. 3a, Supplementary Table 1).

Tracks were designated as 'short' (Fig. 2a, Supplementary Fig. 2a/b(v),c(vi) and d(vii)) when their distance could be measured and as 'long' when we could not fully visualize them in the field of view imaged (Supplementary Fig. 5a). While these features were commonly present on both sides of the tip, we also observed them only on one side (Supplementary Fig. 5b). Short and long tracks could be found in the same bacterium (Supplementary Fig. 5c). Tracks were only identified in SFB that showed evidence of cytoplasmic membrane retraction (Supplementary Fig. 5d(i)) and were present in the majority of SFB also harboring intracellular vesicles (61%, $n = 18$) (Supplementary Fig. 5d(ii)). The length and thickness of short tracks was on average 126 ± 61 nm ($n = 17$) and 55 ± 30 nm ($n = 11$), respectively (Supplementary Fig. 5e(i/ii)).

The length and thickness could not be estimated for long tracks due to the limited region imaged (Supplementary Fig. 5e(i/ii)). Nevertheless, given their similar location and width (Supplementary Fig. 5e(iii)), we hypothesize that both short and long tracks may play a role in maintaining the SFB shape or in anchoring the cytoplasmic membrane within the elongated tip and beyond.

Given that tomograms were required to distinguish intracellular filaments and plate-like structures, we carefully went through our dataset and were able to add filaments from 5 other mouse-SFB to our analysis (**Supplementary Table 2**). We compared the thickness and width of intracellular filaments and plate-like structures and included the results in the **Supplementary Fig. 3c**. Comparisons were not made between the thickness and width of tracks and filaments/plate-like structures given that these structures were not found in the same region of the cell body. Tracks were found exterior to the cytoplasmic membrane while filaments and plate-like structures were found inside the bacterial cytosol.

The corresponding changes to the manuscript are below.

Line 153: One type consisted of multiple filaments clustered together with a thickness of approximately only 26 ± 12 nm ($n = 23$), which made it challenging to follow them in every tomographic slice (Fig. 2c,e/f(iv)), while the other filaments resembled more plate-like structures with a thickness of approximately 63 ± 20 nm ($n = 6$) (Fig. 2d,e/f(v)). The difference between the estimated thickness of these two intracellular features was statistically significant (Supplementary Fig. 3c(i), $p = 0.0110$). The width of individual clustered filaments and plate-like structures was also significantly different (Supplementary Fig. 3c(ii), $p = 0.0055$).

We would also be interested in assessing a potential association between the SFB tip stage and the presence of intracellular filaments and plate-like structures. However, we believe it requires a large dataset acquired using the proper conditions to characterize cytosolic features, namely using the FIB milling approach. The thickness of our samples and limited field of view prevented us from exploring a potential association with the SFB tip stages. In our dataset we identified intracellular filaments and plate-like structures near the SFB tip in eleven and three mouse-SFB IOs, respectively (**Supplementary Table 1, Revision Fig. 3**). Intracellular filaments were mainly found in early SFB stages, being detected in four IOs from Stages 1-3 and one IO from Stages 4-5, while plate-like structures were only found in early SFB stages. However, we currently have a small dataset to assess a potential stage association. Furthermore, given the presence of these features in the bacterial cell body, we might have missed their presence in SFB filaments given the thickness and reduced field of view of our tomograms.

Revision Fig. 3. Current dataset to analyze intracellular filaments and plate-like structures. a/b, Size and stage of SFB IOs in which (a) intracellular filaments and (b) plate-like structures were identified.

Given this, we cannot currently assess a potential association between these features and the identified SFB stages. We modified the manuscript to convey the limitations we are currently facing to further analyze cytosolic features. The corresponding changes to the manuscript are below.

Line 258: However, the limited field of view of our tomograms prevents us from correctly assessing a potential stage-association not only for the chemosensory array but also for other cytosolic features including filaments and plate-like structures, which, however, were also only found in SFB IOs (Supplementary Table 1).

Line 438: Limitations of our work are related to the thickness of our samples and the size of the field of view as these prevented a more detailed characterization of the intracellular features identified and their association with the five SFB stages described. Future studies using cryo-FIB⁴⁴ and a lower magnification could overcome this limitation.

5. It is interesting to see the chemosensory array, a well-studied structural layer in several bacteria species, in the mSFB and rSFB. Could the authors see the chemosensory array lattice array oriented perpendicular to the projecting beam in their tomograms? Does the chemosensory array undergo morphological changes, such as an increased or decreased appearance, when the S-layer is being replaced by the HLL? It will be valuable to have a relatively comprehensive understanding of the morphological changes upon the developmental transitions by incorporating these structures that are not strictly located in the tip region. Model illustrations of the SFB with these structural features that correspond to the main developmental stages e.g. stage 1, 3, 4/5 should do the job.

We have carefully examined the tomograms in which a chemosensory array was identified but unfortunately, we could not see the typical lattice arrangement characterized for other bacterial species¹⁰ oriented perpendicularly to the projecting beam. The position of the chemosensory array in the IOs cell body, a region of increased thickness in comparison to the SFB tip, may pose a challenge for the visualization of this pattern. Future studies using cryo-FIB may allow a better characterization of this feature.

As suggested by the reviewer, we have analyzed the presence of the chemosensory array considering the SFB developmental stage. In our current dataset, we identified a chemosensory array in four tomograms from mouse-SFB and two tomograms from rat-SFB (**Supplementary Table 1**).

Unfortunately, the specific SFB stage could only be identified for one of these tomograms (Stage 2 rat-SFB IO) given that the SFB tip end was not in the field of view for 3 tomograms (tip stage NA) and that the

remaining SFB showed a 'broken tip' phenotype (tip stage UD). Nevertheless, all the chemosensory array-containing SFB were IOs. These were below 2.5 μm in length and were surrounded by an S-layer except for one mouse-SFB IO that was 4 μm in length and possessed an HLL. Thus, it appears that the chemosensory array is mainly associated with early SFB stages.

During the re-analysis of our dataset, we have also noticed that for some IOs a potential chemosensory array was found far from the tip end and at the edge of the field of view (**Revision Fig. 4**). Given this, it is likely that most of the tomograms acquired would not allow the visualization of the chemosensory array even if it was present in the bacteria imaged. Thus, to adequately characterize the association of the chemosensory array with the identified SFB stages, new tomograms would have to be acquired at a lower magnification to enable visualization of both the tip end and a large part of the cell body enabling detection of the chemosensory array.

Revision Fig. 4. Example of a potential chemosensory array located far from the tip end. **a**, Representative tomographic slice from the mouse-SFB IO tomogram EMD-52669 (Supplementary movie 4) showing the location in the bacterial cell body of a potential chemosensory array and **(z)** respective close-up. Given the small region seen from the potential chemosensory array, this bacterium was classified as NA regarding the presence of this intracellular feature. **Scale bars:** a: 50 nm; z: 20 nm.

To clarify that with the current dataset we are unable to claim that a chemosensory array was absent from the bacteria imaged, we have replaced “No” by “NA” in the column relative to the presence of a chemosensory array in **Supplementary Table 1**. The same nomenclature was adapted for other intracellular features given their location in the bacterial cell body and not in the SFB tip.

The text was also modified as follows to highlight the limitation of the current dataset:

Line 438: Limitations of our work are related to the thickness of our samples and the size of the field of view as these prevented a more detailed characterization of the intracellular features identified and their association with the five SFB stages described. Future studies using cryo-FIB⁴⁴ and a lower magnification could overcome this limitation.

Given that the chemosensory array is expected to be involved in bacterial motility, we have also attempted to verify if a correlation between the presence of this feature and flagella could be established. In the current data set, only one rat-SFB IO was found to have both flagella and a chemosensory array (**Supplementary**

Table 1). Future studies with the modifications suggested above should be able to address if a possible association could be established between the presence of these two features.

Nevertheless, unlike chemosensory arrays, flagella could be seen not only in tomograms but also in the low-magnification projection images acquired to measure the bacterial length. Thus, we had a more comprehensive dataset and were able to assess the presence of flagella in the identified stages. The results from this analysis were included in the **Supplementary Fig. 11** and the corresponding text added to the manuscript is below.

Line 250: We have also explored a potential association between the different mouse-SFB stages and the intracellular and extracellular features identified. All flagellated IOs belonged to an early developmental stage (stages 1 to 3) and the proportion of early- and late-stages (stages 4 and 5) in flagellated and non-flagellated IOs was significantly different (Supplementary Fig. 11a/b). Furthermore, stage 1 mouse-SFB comprised 37% of the flagellated IOs (n = 11) and only 12% of the non-flagellated IOs (n = 74). However, this change in stage proportions was not statistically significant (Supplementary Fig. 11a/b). Similarly, the chemosensory array was only found in SFB IOs and may be characteristic of early SFB stages since it was identified in five S-layer-covered IOs and in only one HLL-covered IO (Supplementary Table 1). However, the limited field of view of our tomograms prevents us from correctly assessing a potential stage-association not only for the chemosensory array but also for other cytosolic features including filaments and plate-like structures, which, however, were also only found in SFB IOs (Supplementary Table 1).

5. The authors claim that the observed eVs are SFB-derived. While the eVs do appear in the tip region as shown in Fig. 1, 2, the sample went through purification prior to being put on the grid. It is likely that some eVs from damaged cells or debris could get attached to the SFB during the purification process, which is very common. To prove its SFB origin or support this claim, labeling or live-cell imaging, or observation of these eVs from the native cell cryo-ET where the SFB naturally attached to the surface of the co-cultured host cells, is needed.

The reviewer is correct, with the current data we cannot demonstrate that the extracellular vesicles observed are SFB-derived. The evidence that we currently have to support this hypothesis comes from the fact that eVs are identified near SFB even in wide fields of view that do not show clear evidence of cellular debris around the bacteria imaged (**Revision Fig. 5a**). Furthermore, we showed that SFB are able to produce vesicles (intracellular vesicles, **Fig. 2a,f/g(ii)**, **Supplementary Fig. 2a(v),b(iii),c/d(vi)**) and Smith *et al.* showed the presence of a vesicular coating surrounding unicellular SFB in their natural setting, attached to the intestinal epithelium of a calf¹¹ (**Revision Fig. 5b**). Nevertheless, the presence of these vesicles near SFB when these bacteria are attached to intestinal epithelial cells is also not sufficient to prove our hypothesis. Immunogold labelling experiments would be required, as suggested by the reviewer.

Cell wall proteins have been identified in the eVs of monoderm bacteria such as *Staphylococcus aureus*¹². Since we had performed immunogold labelling on purified SFB samples using a rabbit polyclonal antibody and a VHH-FC against the cell-wall protein Th17Ag, we carefully analyzed our cryo-EM dataset to see if we could detect co-localization between the Th17Ag and the eVs imaged. Purified SFB samples fixed with 4% PFA were screened for the presence of decorated eVs. Regardless of the antibody used, the majority of the eVs were not labelled, even when located near the tip of SFB IOs in which there was clear labelling of the SFB surface (**Revision Fig. 5c**). If we consider co-localization with over 2 gold particles a positive result (above the background for the negative control), labelling was detected for 15% (n = 61) and 10% (n = 10) of the eVs imaged after immunogold labelling with the rabbit polyclonal and the VHH-FC, respectively (**Revision Fig. 5d(i)**). However, if labelling is only considered after co-localization with at least 5 gold particles, the percentage of labelled eVs drops to 3% and 0%, respectively (**Revision Fig. 5d(ii)**). Similar results were obtained when we used a polyclonal antibody against another protein predicted to be cell wall anchored, the mouse-SFB-NL homolog of a minor Th17 antigen identified in mouse-SFB-NYU (3240¹³) (**Revision Fig. 5d**). Thus, we could not demonstrate co-localization of these two SFB proteins with the eVs

imaged. Nevertheless, these results do not exclude the possibility that the eVs are SFB-derived since other SFB proteins may compose the eV coating layer.

REDACTED

Revision Fig. 5. Characterization of the eVs identified near SFB IOs. **a**, Projection image showing a large field of view with only one eV near the SFB tip. A close-up of the region where the eV was identified is shown in **(z)**. **b**, Projection image showing 3 unlabeled decorated extracellular vesicles (eVs) found near the tip of a mouse-SFB IO stained using immunogold labelling with a rabbit polyclonal antibody against the Th17Ag. Purified SFB samples fixed in 4% PFA were used for immunogold labelling with gold-conjugated Protein A. eVs are shown by white arrow heads (*a/b*). Projection images were acquired with a Tecnai F20 electron microscope equipped with a Falcon 2 camera. **c**, Transmission electron microscopy images of SFB IOs attached to intestinal epithelial cells of a calf from Smith *et al.*¹¹. The authors use arrow head heads to indicate the presence of a vesicular coating surrounding SFB IOs. **d**, Proportions of decorated eVs considered labelled by immunogold labelling with either a rabbit polyclonal antibody against the Th17Ag, a VHH-Fc anti-Th17Ag or a rabbit polyclonal antibody against AID45222, the mouse-SFB-NL homolog of a minor Th17 antigen identified in mouse-SFB-NYU (3240¹³). Proportions are shown for vesicles considered labelled when colocalized with **(d(i))** at least 2 gold particles or **(d(ii))** at least 5 gold particles. **Scale bars**: a: 1000 nm; z: 100 nm.

Currently, it would not be possible to demonstrate the origin of these vesicles since we do not know their composition. Thus, we removed the panels corresponding to these features from **Fig. 2** and, similarly, the original **Supplementary movie 5**. However, since eVs, regardless of their origin, were identified in multiple tomograms (**Supplementary Table 1**) we feel uncomfortable in not reporting their presence. Thus, we kept the description of the types of eVs observed in **Supplementary Fig. 2**. To clarify that we cannot demonstrate their SFB-derived origin, the text has been changed as follows:

Line 181: In our samples, eVs were found in close proximity (**Supplementary Fig. 2a(iv),b(iii)**) and in direct contact with the SFB tip (**Supplementary Fig. 2e(iii)**). Additionally, both undecorated and decorated vesicles were identified near SFB that possess both intact and 'broken tip' phenotypes (**Supplementary Fig. 2a(iv), b(iii)**, **Supplementary Table 1**). Overall, the average diameter of eVs was 85 ± 40 nm ($n = 15$) in mouse-SFB and 103 ± 54 nm ($n = 26$) in rat-SFB samples (**Supplementary Table 2**), placing them within the range of Gram-positive eVs³⁷. While there is some evidence that similar vesicles can also be found in vivo, on SFB IOs attached to the epithelium of cows³⁸, we are currently unable to determine if the eVs are SFB-derived since their molecular composition remains unknown.

6. It is common to see nonspecific binding in gold immunolabeling. In some cases, old gold fiducials used for alignment can preferentially bind to the surface proteins, leading to "labeling-like" effect. It is important to provide a control image to show the gold immunolabeling specificity.

We agree with the reviewer. Control SFB that are incubated only with Protein-A-gold and not with rabbit polyclonal/VHH-FC anti-Th17Ag are crucial for excluding potential non-specific binding of Protein-A-gold to surface proteins. The original **Supplementary Fig. 10** included these negative controls for SFB IOs and we have now added control images also for SFB filaments (**Supplementary Fig. 13**). Furthermore, given that Protein-A conjugated with gold particles was also used as fiducials for alignment, all the tomograms deposited in the EMD (EMD-52655; EMD-52667 to EMD-52670; EMD-52671 to EMD-52680; EMD-52682 to EMD-52685; EMD-52687 to EMD-52703; EMD-52856; EMD-54603; EMD-54605; EMD-54607; EMD-54608), some of which have been included in **Supplementary Movies 1-9**, also show the lack of preferential binding of Protein-A-gold to SFB surface proteins.

Additionally, we noticed that using the HyD detector while imaging our immunofluorescence-stained SFB in the SP8 confocal microscope would allow us to detect labelling at the SFB filament tip, supporting our findings using immunogold labelling. We included new images showing filament tip labelling in **Fig. 6f** and **Supplementary Fig. 14f**. Similarly, a representative image showing both IOs and filaments from the negative control for the immunofluorescence (no rabbit polyclonal/VHH-FC anti-Th17Ag was added), which was acquired using the HyD detector, was added to **Supplementary Fig. 13 a(iii)**.

7. The authors hypothesize that the presence of S-layer and HLS are involved in cell wall integrity. While the eV is seen near the broken phenotype, as stated above, it is unclear where the eV comes from. To prove this hypothesis, cell wall integrity needs to be checked for the various developmental stages and see if there is a difference between stage 1 and stage 3 and stage 5. While Th17Ag, a cell wall protein, has been explored in the paper, did the author check other landmarks that prove the cell integrity of the SFB after the purification?

Given that the molecular composition of the disordered hair-like structures remains unknown and that there are no differences in size between stage 1 and stage 3 IOs, we are currently unable to compare their cell integrity by other methods than immunogold labelling of cell wall proteins followed by cryo-EM. Furthermore, we are also currently lacking a tool to isolate SFB IOs with a broken tip and even to distinguish them without using electron microscopy techniques.

Nevertheless, as suggested by the reviewer, we performed a classical LIVE/DEAD staining followed by immunofluorescence to assess cell viability after SFB purification (**Revision Fig. 6a-d**). We used propidium iodide and SYTO9 in a 1:1 ratio to stain purified SFB and SFB obtained after short centrifugation steps (1 minute at 70 x g + 5 minutes at 500 x g) to remove the majority of fecal debris and host cells (non-purified SFB). As a control for dead bacteria, we treated purified SFB with 5% SDS for 5 minutes at room temperature. As expected, SDS-treated IOs and filaments stained red or yellow showing evidence of propidium iodide staining and were considered dead (**Revision Fig. 6a**). On the other hand, the majority of the non-SDS-treated SFB IOs (89%, n = 28) and filaments (94%, n = 16) stained green being considered alive (**Revision Fig. 6d**). Nevertheless, the presence of red-stained segments (dead segments) was detected in 24% of the filaments imaged (**Revision Fig. 6b/d**). Staining of non-purified SFB revealed a similar trend given that 44% of the filaments imaged (n = 25) possessed dead segments (**Revision Fig. 6c/d**). Thus, the loss of integrity for these segments seems to have occurred in the mouse gut and not during the SFB purification.

Revision Fig. 6. Assessment of SFB viability after purification. a-c, Staining of (a) purified SFB treated with 5% SDS at room temperature for 5 minutes (DEAD SFB control), (b) purified SFB and (c) non-purified SFB with propidium iodide (PI, red) and SYTO9 (green) from the LIVE/DEAD BacLight Bacterial Viability Kit (L7012, Thermo Fisher Scientific). Images of the same samples showing: (a-c(i)) bright field, (a-c(ii)) SYTO9 (green) and PI signal (red), (a-c(iii)) only SYTO9 signal (grey) and (a-c(iv)) only PI signal (grey) are shown. Bacteria exhibiting red or yellow fluorescence are considered dead. Staining was conducted according to the manufacturer's instructions. PI and SYTO9 were added in a 1:1 proportion and incubated with SFB for 15 minutes at room temperature. Bacteria were fixed with 4% PFA (10 minutes at room temperature) after staining and washes were performed using PBS1X. A single plane and maximum intensity z-stacks are shown for panels a-c(i) and a-c(ii-iv), respectively. Images of SFB IOs not included in the image shown in the main panel were added as complementary panels delimited by a full line. Red arrow heads indicate dead segments present in SFB filaments. d, Proportions of live and dead SFB identified after staining with PI and SYTO9. The proportion of SFB filaments in which both live and dead segments were identified is also shown. **Scale bars:** 5 μm for SFB filament images, 1 μm for close-ups/additional images of SFB IOs.

Nevertheless, these results do not allow us to validate the hypothesis that the S-layer and the HLL are involved in cell integrity and it remains unclear if the HLL might have been compromised in the DEAD segments. To address this hypothesis, we would have to establish the technique of cryo-correlative light and electron microscopy for SFB imaging which, could not be achieved in the time frame of this revision.

We have modified the text to clarify that our hypothesis was only based on the cell wall accessibility experiments performed for Th17Ag labelling. The text is now as follows:

Line 389: S-layers can play a role in host adhesion in bacteria with symmetric poles³¹. However, since in SFB it is the tip that mediates epithelial cell attachment, it is unlikely that the S-layer or the HLL are involved in adhesion but rather may function in bacterial protection and/or maintenance of cell wall integrity. Immunogold labelling of the Th17Ag supports this hypothesis. The Th17Ag was previously found to surround SFB filaments in intestinal sections and to be taken up from the filament tip through the formation of host endocytic vesicles¹⁸. We obtained similar labelling with permeabilization but without permeabilization this protein was surface-exposed only at the filament tip for the majority of the filaments imaged. Similar to filaments, strong labelling was observed in IOs under permeabilization conditions. However, only approximately a quarter of non-permeabilized IOs showed labelling and labelling was not restricted to a specific region of the bacterial surface. These results suggest that the HLL in SFB filaments prevents access to this potential cell wall-anchored protein while the S-layer appears to have a similar function for IOs. We hypothesize that the increased surface exposure in a subset of IOs may be due to changes in cell wall accessibility during the S-layer to HLL transition. These findings provide further evidence of a developmental transition occurring at the SFB tip.

8. Cryo-ET data have been collected at multiple pixel sizes with various defocus and processing. While the method section is comprehensive, it will be good to provide a table that states which pixel size data sets are used for what data analysis e.g. S-layer structure characterizations in Fig. xxx, etc.

The **Supplementary Table 1** has been revised to include the original pixel size of all the tomograms included in our dataset and the pixel size after binning for the tomograms used as representative and for the FFT analysis. We have also added extra columns to indicate which tomograms were used for the analyses shown in Fig. 1k, 3n and 5f.

Minor:

In Fig. 3 d iii, it is challenging to see the presence of both ordered HLS and disHLS in the current tomogram slice view.

The reviewer is correct. Only disordered hair-like structures can be seen in **Fig. 3d(iii)** since the region shown is only the end of the tip of a mouse-SFB filament (i.e. the full tip cannot be seen). In the schematics shown in **Fig. 3d(iv)** we have represented the full tip until the bacterial curvature (the region considered the tip is now better indicated in **Fig. 1a**) to show that Stage 4 SFB have disordered hair-like structures at the tip end but ordered hair-like structures further down if we look at the tip side. We adjusted the schematics shown in **Fig. 3d(iv)** to more accurately reflect the phenotype seen in Stage 4 SFB. In addition, we added a new panel showing the side of the tip of this Stage 4 filament in which ordered hair-like structures can be seen (**Fig. 3h**). Additionally, we added the designation “tip end” to **Supplementary Fig. 6** and prepared tomographic slices from another Stage 4 filament showing the presence of disordered hair-like structures at the tip end and ordered hair-like structures at the tip side. This new panel was added to **Supplementary Fig. 6**.

The corresponding text has been changed accordingly:

Line 222: The HLL was found at the SFB tip end (stage 5, Fig. 3d-e, Supplementary Fig. 6b-d(i-ii)), tip side (stage 4 and 5, Fig. 3h/i, Supplementary Fig. 6b-d(iii)) and distal end (stage 4 and 5, Fig. 3k, Supplementary Fig. 9c-d). In stage 4 SFB, the end of the tip possessed disordered hair-like structures (Fig. 3d), which, however, were significantly smaller ($p < 0.0001$) than those of stage 3 (28 ± 5 nm, $n = 250$, Fig. 3n).

Fig 3. B is a 2D projection instead of a tomogram slice.

The reviewer is correct, Fig. 3b is a 2D projection and we verified that this is correctly indicated in the figure legend. We believe that this projection image is the clearest representative we have in our dataset to show the morphological features characteristic of Stage 2 SFB for mouse-SFB grown in mice.

In Fig 3.f, it is hard to pinpoint each stage and their percentages due to the similar shades of grey. It might be better to use other more distinct colors to represent in the pie chart.

The colors of the pie chart of Fig. 3f were changed following the reviewer's recommendation.

Line 320, the HLL may itself constitute an S-layer since it is not repetitive but also appears to prevent the access of the protein. There is no direct evidence in the manuscript to support this claim.

We agree with the reviewer, we do not provide direct evidence to demonstrate that the HLL is an S-layer and this layer is morphologically different from the bacterial S-layers described in the literature.

We have modified the sentence to:

Line 383: The HLL may constitute an atypical surface layer as it is repetitive and exterior to the cell wall.

We claim that the HLL may be preventing access to the Th17Ag since this antigen was only accessible on the filament tip and not on the remaining cell body and distal end where the HLL was present (**Fig. 6a**) even though it could be accessible in every region of SFB filaments when a denaturation treatment that compromised the integrity of the HLL was performed (**Fig. 6b**).

Line 735, please state the original pixel size and binned pixel size after the bin4 has been performed.

The original pixel size and binned pixel size of every tomogram used for each analysis has been included in **Supplementary Table 1**.

Reviewer #3 (Remarks to the Author):

The manuscript by Gruz and colleagues presents a high-resolution study of segmented filamentous bacteria - an important gut commensal microbe known for its potent immunostimulatory effects using state-of-the-art cryo-electron microscopy and cryo-electron tomography (cryo-EM/ET). The authors capture high-resolution images of SFB at different developmental stages, from short unicellular intracellular offsprings, IO, to long filamentous cells. The authors discover a progression of surface structures, an S-layer transforming into an entirely new, "hair-like" layer, identify novel "hair-like structures" potentially involved in host-cell attachment, and demonstrate differential surface accessibility of the major Th17 antigen at the tip of filamentous forms.

The cryo-EM/ET data is of high quality, the tomograms look excellent, and are analyzed rigorously with respect to the developmental stages. The manuscript is well-written and clear. It provides novel insights into the development and provides food for thought for future research.

We thank the reviewer for the encouraging comments and for highlighting the study's strengths and relevance to the scientific community.

The major drawback of the manuscript is the unknown molecular identity of the hair-like structures. This is a very important limitation of the study, and as it is, the manuscript is very descriptive. It may be possible to identify the molecules by pulldowns of Th17, proteomics, bioinformatics, subvolume averaging, or their combination.

We agree with the reviewer that our study presents a morphological characterization of novel structures that would benefit from future characterization and additional studies. Moreover, identifying the molecular composition of these novel surface features, and in particular of the disordered hair-like structures, would be important to better understand SFB life-cycle and the intimate host-bacteria interaction. We have used multiple approaches (including proteomics, transcriptomics, bioinformatics analysis and localization studies of candidate proteins), some of which have also been suggested by the reviewer, to achieve this aim and are currently still working towards this goal. However, the execution of such a project using Segmented Filamentous Bacteria is complex due to multiple challenges including the current lack of genetic manipulation tools and the current impossibility to grow these bacteria in the absence of host cells. Thus, we cannot yet provide a conclusive answer despite a large amount of additional work already performed that, however, has for now yielded only negative results.

The use of subvolume averaging to complement other strategies and allow identification of the molecular composition of the disordered hair-like structures has also been suggested by the reviewer. In other instances, this strategy has been successfully used to identify the proteins composing uncharacterized complexes localized at the bacterial surface when coupled with proteomics and AlphaFold predictions of protein structures¹⁴. However, in our case, there is a low confidence in the AlphaFold structural predictions for some of our candidate proteins. Additionally, it is challenging to have each individual hair-like structure as a discrete feature due to their high density, proximity from neighboring structures, and apparent flexibility. We added to the manuscript tomographic slices of the tip's top view from Stage 3 IOs to provide evidence for the disordered structural arrangement and apparent flexibility of these structures (**Supplementary Fig. 7a**).

The following description was added to the manuscript:

Line 197: In the remaining IOs below 3.2 μm , the tip showed varying degrees of hair-like structures with a disordered appearance seen from both side (Fig. 3b/c/f, Supplementary Fig. 6a) and top views (Supplementary Fig. 7a).

These observations together with the high curvature of the SFB tip would make it very challenging to obtain meaningful results from subtomogram averaging of hair-like structures. We expect this to be particularly difficult given that our current attempts to perform this analysis on the well-ordered and repetitive SFB S-layer have not been successful (**Revision Fig. 1**).

The reviewer also suggested to perform pulldowns of the major Th17 antigen (Th17Ag) to identify the molecular composition of the disordered hair-like structures. The results presented in this manuscript (Fig. 6) and a previous study from Ladinsky *et al.*¹⁵ suggest that the Th17Ag is localized in the bacterial cell wall. Given this, it remains unclear if proteins composing the disordered hair-like structures would have a direct interaction with the Th17Ag and, thus, if these would be successfully captured using a pull-down strategy.

As the identification of the molecular composition of the characterized surface structures is still a work in progress, we have included the following sentence at the end of the discussion:

Line 458: Given that the major immunostimulatory properties of SFB are dependent on adhesion to host cells and that surface-exposed supramolecular structures generally play critical roles in host-bacterial interactions, future studies will focus on unravelling the molecular details of the novel surface structures identified to obtain further insights on how the SFB-host cell interaction is established.

Furthermore, it would be interesting to describe the mutual arrangement of the hair-like molecules throughout the developmental process, at least where they are ordered. This could be another important step for the identification of the molecules. Even though the cryo-ET dataset is of moderate size, there should be enough particles for subvolume averaging, and even at moderate resolution, it can provide useful insights.

Similar to what we described for the disordered hair-like structures, there seems to be flexibility in the subunits composing the hair-like layer (HLL). We added to the manuscript tomographic slices tip's top view of Stage 5 SFB to show the arrangement and apparent flexibility of these structures (**Supplementary Fig. 7b/d**). This revealed the absence of a clearly ordered top view of the HLL subunits, with the exception of the potential transitional stage (**Supplementary Fig. 7c**) which was included in **Supplementary Fig. 10b**.

The following description was added to the manuscript:

Line 221: It is composed of an array of ordered hair-like structures whose arrangement is challenging to identify from the tip's top view (Supplementary Fig. 7b).

Line 278: As noted for mouse-SFB, it was challenging to identify a clear arrangement of the HLL subunits from the top view of the rat-SFB tip. An exception was a rare rat-SFB IO showing a potential transitional stage between stages 3 and 4 (Supplementary Fig. 10b) where arrangements resembling rows could be seen (Supplementary Fig. 7c).

Given this and the fact that our current attempts to perform subtomogram averaging on the well-ordered SFB S-layer have not been successful (**Revision Fig. 1**), we believe this strategy will not bring meaningful results towards the HLL characterization. Furthermore, the high density and lower HLL subunit spacings in comparison to the SFB S-layer (~5 nm vs ~8 nm) imply that an even higher resolution would be necessary to resolve the HLL lattice arrangement.

References

1. Smit, E., Oling, F., Demel, R., Martinez, B. & Pouwels, P. H. The S-layer protein of *Lactobacillus acidophilus* ATCC 4356: Identification and characterisation of domains responsible for S-protein assembly and cell wall binding. *J Mol Biol* **305**, 245–257 (2001).
2. Åvall-Jääskeläinen, S. *et al.* Identification and characterization of domains responsible for self-assembly and cell wall binding of the surface layer protein of *Lactobacillus brevis* ATCC 8287. *BMC Microbiol* **8**, (2008).
3. Schnupf, P. *et al.* Growth and host interaction of mouse segmented filamentous bacteria in vitro. *Nature* **520**, 99–103 (2015).
4. Qu, K. *et al.* Structure and architecture of immature and mature murine leukemia virus capsids. *Proc Natl Acad Sci U S A* **115**, E11751–E11760 (2018).
5. Castaño-Díez, D., Kudryashev, M. & Stahlberg, H. Dynamo Catalogue: Geometrical tools and data management for particle picking in subtomogram averaging of cryo-electron tomograms. *J Struct Biol* **197**, 135–144 (2017).
6. Bharat, T. A. M. & Scheres, S. H. W. Resolving macromolecular structures from electron cryo-Tomography data using subtomogram averaging in RELION. *Nat Protoc* **11**, 2054–2065 (2016).
7. Burt, A., Gaifas, L., Dendooven, T. & Gutsche, I. A flexible framework for multi-particle refinement in cryo-electron tomography. *PLoS Biol* **19**, (2021).
8. Bharat, T. A. M. *et al.* Structure of the hexagonal surface layer on *Caulobacter crescentus* cells. *Nat Microbiol* **2**, 1–6 (2017).
9. Gambelli, L. *et al.* Architecture and modular assembly of *Sulfolobus* S-layers revealed by electron cryotomography. (2019) doi:10.1073/pnas.1911262116.
10. Oikonomou, C. M. & Jensen, G. J. The Atlas of Bacterial & Archaeal Cell Structure: an Interactive Open-Access Microbiology Textbook. *J Microbiol Biol Educ* **22**, e00128-21 (2021).
11. Smith, T. M. Segmented Filamentous Bacteria in the Bovine Small Intestine. *J. Comp. Path* **117**, 185–190 (1997).
12. Wang, X., Thompson, C. D., Weidenmaier, C. & Lee, J. C. Release of *Staphylococcus aureus* extracellular vesicles and their application as a vaccine platform. *Nat Commun* **9**, (2018).
13. Yang, Y. *et al.* Focused specificity of intestinal TH17 cells towards commensal bacterial antigens. *Nature* **510**, 152–156 (2014).
14. Jensen, R. K. *et al.* In-cell discovery and characterization of a non-canonical bacterial protein translocation-folding complex. *bioRxiv* (2025) doi:10.1101/2025.04.25.650208.
15. Ladinsky, M. S. *et al.* Endocytosis of commensal antigens by intestinal epithelial cells regulates mucosal T cell homeostasis. *Science* (1979) **363**, eaat4042 (2019).

Point-by-point response to reviewers' comments for the manuscript NCOMMS-25-16988A

Text in black – reviewer comments

Text in orange – author response

Text in green – changes in the manuscript

We thank the reviewers for their feedback on our revised manuscript and provide our point-by-point response below.

REVIEWER COMMENTS

Reviewer #2 (Remarks to the Author):

I appreciate the authors took time to address the questions brought up, particularly analyzing the OI tip with STA. I commend the authors for the time and work that went into it. I have the following questions and hope to further improve the manuscript.

We thank the reviewer for their kind comments and his/her further contribution to improve the manuscript.

1) Line 108: Here, the symmetry observation is being described and discussed. I agree that the claim would be "potential" 6-fold symmetry. If the author would like to mention the performance of STA, it would be necessary/important to include the STA results as part of the Supplemental/Extended Fig. I think in this case, the lack of a clear definition of 6-fold symmetry organization could result from the missing wedge artifact despite the missing wedge compensation in PEET or Dynamo. I wonder if the author has tried to limit the angular search range and translational search range as well after all subtomogram were aligned perpendicular to the beam direction. I have seen lattice structures suffer from the missing wedge effect (despite correction) during the alignment. As a result, the wrong alignment led to misleading 6-fold symmetry. Two approaches could be helpful. One would be to align the subtomograms towards a fixed direction, e.g. Revision Fig. 1e, where the membrane is centered and perpendicular to the Y. Then apply a limited search angle/translation so that the aligned/guided positions are not lost during the subsequent alignment in PEET or Dynamo. The other one would be to provide some guidance for the angular search or predetermined orientation in Relion 3D refinement.

The subsequent analysis could require more time. I would suggest removing the sentence "nevertheless, xxxx (data not shown)".

We thank the reviewer for their guidance on alternative approaches that can be applied to avoid missing wedge artifacts that may prevent the identification of a clear 6-fold symmetry. We have tried both approaches suggested.

During picking, the orientation of each particle relative to the picking surface was determined using the Dynamo workflow. The angular information was converted to Relion convention using the dynamo2m package (<https://github.com/alisterburt/dynamo2m>) and used in Relion as a starting point for classification or refinement. We tried giving this information both as the particle orientation (using the rlnAngle header), or as a constraint on the angle using the rlnAnglePrior header. The latter is not overwritten during refinement, and provides a soft guidance for the alignment. Subsequently, we used the `—sigma_ang` flag to limit the angular search in Relion 3D Refinement to orientations close to those pre-orientations determined during picking.

Neither of these approaches helped us converge on a global S-layer symmetry. In our opinion, the high curvature of the SFB tip provides a strong feature, which enforces alignment of the individual particles in the beam direction, and thus with aligned missing wedges. We tried to mask only the central density during alignment to overcome the curvature of the tip. However, due to the small repeat distance, the masked volume was too small and provided too little density for STA to be successful.

Given the current limitations towards understanding the putative S-layer structure, we accepted the reviewer's suggestion and removed the following sentence from the manuscript: "Nevertheless, subtomogram averaging of the S-layer globular densities did not allow de novo identification of the lattice symmetry (data not shown).".

Line 107: Identification of the S-layer symmetry will require a more detailed analysis beyond what whole SFB cryo-ET can provide.

Additionally, we no longer mention "potential six-fold symmetry" in the manuscript given the limitations faced with the current structural analysis. The text was modified as follows:

Line 100: Additionally, an apparent hexagonal organization was detected (Supplementary Fig. 1e,f) and most clearly seen in a tomogram of a rat-SFB IO with a 'broken tip' phenotype whereby the S-layer showed abrupt discontinuity and was absent at the tip end (Supplementary Fig. 1f).

2) Line 108, "a more relaxed S-layer". Is there any evidence or measurement, like curvature measurements, to prove the claim?

Since in IOs with a 'broken tip' phenotype the restraint to maintain the tip shape would be lost, we hypothesized that this may lead to a more relaxed S-layer lattice. However, the reviewer is correct, we do not have direct measurements that allow us to prove our hypothesis that the potential 6-fold symmetry seen in the rat IO with a 'broken tip' phenotype can be visualized due to a more relaxed S-layer.

Given that this potential 6-fold symmetry was only seen in one bacterium with a 'broken tip' phenotype, curvature measurements would not allow us to confirm this hypothesis. We have removed the sentence "A potential 6-fold symmetry could be seen in the rat IO with a 'broken tip' phenotype, possibly due to a more relaxed S-layer (Supplementary Fig. 1f)." from the manuscript.

Line 107: Identification of the S-layer symmetry will require a more detailed analysis beyond what whole SFB cryo-ET can provide.

3) Fig. 6, the HLS is too small to see in z (white arrowhead), the HLS will need to be zoomed in/enlarged. In f, the merged and Alexa-658 are mislabeled/backwards. h, the residue font size is too small to read.

The projection images included in Fig. 6(a-d) were acquired using a 200 kV Tecnai F20 electron microscope, making it more challenging to prepare high resolution close ups to see the hair-like structures. Thus, we decided to image other grids from this experiment, in which fixed SFB were labelled with a VHH-Fc, using a 300 kV Titan Krios electron microscope. This allowed us to add a new panel to Fig. 6 (panel e) to better show the hair-like structures labelled at the SFB filaments tip (Fig. 6e(i)) and unlabeled at the SFB filaments distal end (Fig. 6e(ii)).

The text in the manuscript was adjusted accordingly.

Line 322: The Th17Ag labelling was seen in regions containing disordered hair-like structures (Fig. 6e(i)) but not the HLL (Fig. 6e(ii)).

We have also placed the panels of Fig. 6f (now Fig. 6g) in the correct position and increased the font size of the amino acid residues numbers of Fig. 6h (now Fig. 6i).

4) Could the authors comment on why Th17Ag shows the labeling is more predominant in the late-stage IOs compared to the early stage, where the S-layer is present? Also, is Th17Ag present in both ordered and disordered HLS?

To better reply to these questions, we imaged the fixed SFB labelled with a VHH-Fc and imaged using a 300 kV Titan Krios electron microscope, as mentioned above, since it allowed us to visualize S-layer subunits and differentiate between Stage 4 and Stage 5 filaments. The additional SFB imaged were added to Supplementary Table 3 and the analyses shown in Fig. 6f and Supplementary Fig. 16 were updated with the new data collected. The manuscript was also updated accordingly.

Also, is Th17Ag present in both ordered and disordered HLS?

In fixed filaments, the Th17Ag is predominantly inaccessible in regions containing ordered HLS as shown by the lack of labelling at the filaments distal end (Fig. 6a/e and Supplementary Fig. 14a). Additionally, the tip-restricted labelling in SFB filaments imaged using the 300 kV Titan Krios electron microscope occurs in Stage 4 filaments, in regions containing the disordered hair-like structures (Fig. 6e, Supplementary Table 3).

However, in fixed IOs a different pattern was observed. Th17Ag labelling can be seen in regions containing ordered HLS (Supplementary Fig. 15 a/b) but is not always present in regions containing disordered HLS (Supplementary Fig. 15c). In addition, it was also found in S-layer-containing regions (Supplementary Fig. 15c).

The manuscript has been modified to clearly present these observations.

Line 319: For filaments, Th17Ag labeling was seen in 82% of the filaments imaged ($n = 17$) and, for 93% of these, was strikingly restricted to the filament tip ($n = 13$) (Fig. 6a/f, Supplementary Table 3). The Th17Ag labelling was seen in regions containing disordered hair-like structures (Fig. 6e(i)) but not the HLL (Fig. 6e(ii)). For IOs, Th17Ag labelling was observed in only 30% of the IOs imaged ($n = 27$) (Fig. 6f) and, in 75% of those, occurred at the tip but also on the remaining cell body, including at the distal end (Fig. 6c). In agreement, IOs showed labelling with the VHH-Fc targeting the Th17Ag using immunofluorescence (Fig. 6g). In contrast with the pattern observed in filaments, labelling of IOs was, albeit weakly, observed in regions containing the HLL (Supplementary Fig. 15a/b) but was not always present in regions including disordered hair-like structures (Supplementary Fig 15c). Additionally, as the Th17Ag was detected in the IOs distal end (Fig. 6c, Supplementary Fig. 14c) and in other regions where an S-layer was present (Supplementary Fig. 15c), it is unlikely that the Th17Ag is a component of the disordered hair-like structures.

Could the authors comment on why Th17Ag shows the labeling is more predominant in the late-stage IOs compared to the early stage, where the S-layer is present?

We hypothesize that late-stage IOs may be more predominantly labelled due to the transition between the S-layer to the HLL, which may facilitate access to the cell wall localized Th17Ag. Late-stage IOs may represent transitional stages of Th17Ag accessibility due to the formation of the HLL that, in filaments, blocks the access to this cell wall protein. We modified the manuscript to better present this hypothesis.

Line 359: For IOs, as only a small subset, enriched in late-stage IOs, was labelled, we hypothesize that there may be intermediate accessibility stages (Supplementary Fig. 15a/b) prior to the impermeability of the HLL seen in filaments.

5) Line 439, it might be good to specify the pixel size for the lower mag because too small a mag (big pixel size) could lead to loss of important structural details that are required for further analysis. A bigger pixel size (4~5 Å/pixel) or montage cryoET (Reference) could help overcome the field of view limitation.

We agree with the reviewer, it would be better to increase the field of view without compromising the resolution of our tomogram. We modified this sentence in the discussion to suggest the use of montage cryo-ET schemes to overcome the field of view limitation instead of using a lower magnification.

Line 448: Future studies using cryo-FIB⁴⁵ and montage cryo-ET schemes⁴⁶ to increase the field of view without compromising tomogram resolution could overcome this limitation.

6) Line 448, I would suggest removing "the extraction and purification of the S-layer... may overcome these limitations". Do the authors mean purifying the S-layer stems followed by reconstitution, or just purifying and putting them under the microscope? Purification and extraction may break the samples. To further study the organization of the S-layer stems below, a native, in-situ approach, as done here still remains the best. I would suggest removing this comment.

Our previous suggestion was to extract the S-layer directly from SFB followed by imaging using cryo-ET, similarly to what was performed for *Caulobacter crescentus*¹. As discussed above, we have already performed multiple attempts to identify the S-layer *in situ*. Thus, we believe that it may be necessary to try a different strategy.

We proposed this approach since we hypothesize that extracting the S-layer from SFB may eliminate the potential restraints necessary to maintain the tip shape and the tip curvature limitation leading to a more relaxed S-layer lattice and to the isolation of lattice sheets that could potentially facilitate the symmetry identification.

However, we agree with the reviewer, this approach does not guarantee the maintenance of the S-layer lattice structure. As suggested by the reviewer, we removed the following sentences: "The extraction and purification of the S-layer from IOs may overcome these limitations. Yet, given the challenges related to the study of SFB, including the use of monoclonized mice for bacterial propagation, a lack of availability of large quantities of IOs, and the lack of knowledge on the proteins that compose the IO-associated S-layer, obtaining S-layer sheets in solution will be challenging."

7) Line 444, no evidence for "twisted lattice" was provided here. I would think the tip curvature and the sample size would be the limiting factors for higher resolution STA analysis.

When analyzing the top view of mouse-SFB IOs tips, the arrangement of the S-layer subunits led to the visual appearance of a "twisted lattice" or "twisted tip". This can be seen in Fig. 1I and Supplementary Fig. 1a. We modified the results section to clarify that this designation was based on the visual appearance of the S-layer subunits arrangement from the top view.

Line 104: In the top views of intact SFB tips, the arrangement of S-layer subunits revealed a twisted appearance at the tip (Supplementary Fig. 1a).

Additionally, we took the reviewers suggestion and removed this descriptive designation from the discussion.

Line 452: However, the strong SFB tip curvature, and smaller center-to-center spacing of the SFB S-layer (~8 nm) in comparison to the S-layer from *C. crescentus* (20-22 nm^{28,47}) may have contributed to the difficulties encountered in the identification of the lattice symmetry for SFB.

8) In Fig5 a-d, most of the I or O-HLL don't seem to go together. e.g. in all of them, I barely see a uniform coating of I-HLL. Is it because they appear on different tomographic slices? Could summing a few slices in Slicer/IMOD help to display them both at the same time? They seem to share a very similar spacing. The schematic view (c) shows O and I-HLL appear at the same time with almost identical spacing. If the organization of I/O-HLL in the slice view is not due to the different Z planes, then c needs to be modified.

We understand the reviewer's concern and acknowledge that it is difficult to see a uniform coating for the I-HLL. As the reviewer pointed out, this occurs since the subunits of the HLL present in the inner (I-HLL) and outer (O-HLL) region of the layer cannot always be seen in the same tomographic slice, as shown in revision Fig. 1a/b. Nevertheless, a more electron dense region ("shade") that includes the inner part of the HLL subunits can be seen in the different tomographic slices (Revision Fig. 1c).

The projection images included in the manuscript (Fig. 5a-d(i)) and in the Revision Fig. 1 were prepared using the Slicer tool in IMOD and 10 tomographic slices were summed. Thus, unfortunately this strategy was not enough to clearly display both I-HLL and O-HLL at the same time for large regions of the bacterial surface.

It is also noteworthy to mention that the projection images included in Fig. 5a-d are from tomograms subjected to binning 2 (pixel size: 4.42 Å or 4.02 Å) since these were used for the FFT analysis presented. Using tomograms subjected to binning 4 and filtered using topaz denoising² helps to better display both I-HLL and O-HLL in the same tomographic slice. This can be seen in the tomographic slices included in Fig. 3h/i (mouse-SFB, pixel size: 8.84 and 7.00, respectively). A projection image where both I-HLL and O-HLL can clearly be seen at the same time is also included in Fig. 4f (rat-SFB). To clarify that I-HLL and O-HLL are challenging to display in the same tomographic slice and point towards other examples in which they can be seen more clearly, we included the following sentence in the manuscript:

Line 294: The I-HLL and O-HLL were often challenging to identify in the same Z plane (Fig. 5a-d). However, we could simultaneously observe hair-like structures both in the exterior and in the interior of the HLL electron dense line for both mouse-SFB (Fig. 3h/i) and rat-SFB (Fig. 4f).

Revision Fig. 1. Visualization of HLL subunits in different tomographic slices. **a-c**, Representative tomographic slices from reconstructed tomogram EMD-52677 at **(a/c)** $z=639$ and at **(b)** $z=621$. **(c)** Identification of the different HLL-related layers shown in **(c(i))** a representative tomographic slice and in **(c(ii))** a schematic. HLL subunits viewed from the outer hair-like layer (O-HLL) and the inner hair-like layer (I-HLL) are shown by red and white arrows, respectively. **Scale bars:** a-c: 20 nm.

Reviewer #3 (Remarks to the Author):

I thank the authors for their thorough and detailed response to the initial reviews. The manuscript has been improved by the revisions, and the additional analyses and clarifications strengthened the work. The authors have addressed my previous comments satisfactorily. They have acknowledged the limitations regarding the molecular identity of the newly described hair-like structures and have convincingly argued in their rebuttal that the identification of these components is fraught with significant technical challenges inherent to the SFB system. I appreciate the addition of new supplementary figures showing the top-down views of the disordered and ordered hair-like structures, which support their assertion that subtomogram averaging is not currently feasible due to the flexibility and lack of a clear, repetitive lattice. Despite these improvements, the central and significant gap of the study remains the unknown molecular identity of the S-layer, the disordered hair-like structures, and the hair-like layer. This lack of molecular information necessarily renders the manuscript highly descriptive in nature. While the morphological characterization is of very high quality and provides novel insights into the SFB life cycle, the absence of molecular data is a major limitation. The manuscript does not provide the molecular link, but it is correct and provides a foundation for future molecular studies.

We thank the reviewer for acknowledging our work towards the improvement of the manuscript, the technical challenges associated with the study of SFB, and the quality of our morphological characterization. We also appreciate the reviewer's interest in the molecular identity of the novel structures here described. We will continue our work in this direction and hope to be able to provide molecular insights into SFB's life cycle in the future.

References

1. Bharat, T. A. M., Kureisaite-Ciziene, D., Hardy, G. G., Yu, E. W., Devant, J. M., Hagen, W. J. H., Brun, Y. V., Briggs, J. A. G. & Löwe, J. Structure of the hexagonal surface layer on *Caulobacter crescentus* cells. *Nat Microbiol* 2, 1–6 (2017).
2. Bepler, T., Kelley, K., Noble, A. J. & Berger, B. Topaz-Denoise: general deep denoising models for cryoEM and cryoET. *Nat Commun* 11, 5208 (2020).